ARTICLES

# Antibiotic resistance genes in the gut microbiota of mothers and linked neonates with or without sepsis from low- and middle-income countries

M. J. Carvalho [1,2,29] ✉, K. Sands[1,3,29], K. Thomson[1,3], E. Portal[1], J. Mathias[1], R. Milton [1,4], D. Gillespie [4], C. Dyer[1,4], C. Akpulu[1,3], I. Boostrom[1], P. Hogan[1], H. Saif[1], A. Ferreira[1,5], M. Nieto [1,3], T. Hender [1], K. Hood[4], R. Andrews [1], W. J. Watkins[1], B. Hassan[1], G. Chan[6,7,8], D. Bekele [7,9], S. Solomon [10], G. Metaferia[10], S. Basu [11], S. Naha[11], A. Sinha[11], P. Chakravorty[12], S. Mukherjee[13], K. Iregbu[14], F. Modibbo[15], S. Uwaezuoke[16], L. Audu [14], C. P. Edwin[17], A. H. Yusuf [18], A. Adeleye[19,20], A. S. Mukkadas[15], R. Zahra[21], H. Shirazi[22], A. Muhammad[21], S. N. Ullah[21], M. H. Jan[21], S. Akif[21], J. B. Mazarati[23], A. Rucogoza[23], L. Gaju[23], S. Mehtar[24,25], A. N. H. Bulabula[25,26], A. Whitelaw[27,28], L. Roberts[27], BARNARDS Group* and  T. R. Walsh [1,3]

Early development of the microbiome has been shown to affect general health and physical development of the infant and, although some studies have been undertaken in high-income countries, there are few studies from low- and middle-income countries. As part of the BARNARDS study, we examined the rectal microbiota of 2,931 neonates (term used up to 60 d) with clinical signs of sepsis and of 15,217 mothers screening for $bla_{CTX-M-15}$, $bla_{NDM}$, $bla_{KPC}$ and $bla_{OXA-48}$-like genes, which were detected in 56.1%, 18.5%, 0% and 4.1% of neonates' rectal swabs and 47.1%, 4.6%, 0% and 1.6% of mothers' rectal swabs, respectively. Carbapenemase-positive bacteria were identified by MALDI-TOF MS and showed a high diversity of bacterial species (57 distinct species/genera) which exhibited resistance to most of the antibiotics tested. *Escherichia coli*, *Klebsiella pneumoniae* and *Enterobacter cloacae/E. cloacae* complex, the most commonly found isolates, were subjected to whole-genome sequencing analysis and revealed close relationships between isolates from different samples, suggesting transmission of bacteria between neonates, and between neonates and mothers. Associations between the carriage of antimicrobial resistance genes (ARGs) and healthcare/environmental factors were identified, and the presence of ARGs was a predictor of neonatal sepsis and adverse birth outcomes.

Classically, antimicrobial resistance (AMR) is perceived as a clinical problem but non-clinical environments (for example, the human gut microbiota) are now increasingly important due to their contribution in disseminating AMR genes (ARGs). Furthermore, ARGs frequently exchange between bacteria within the human microbiota, where the intestinal bacterial community acts as a hub for horizontal gene transfer[1,2]. This is especially concerning for the neonatal population because colonization with

multi-drug-resistant (MDR) bacteria is a precursor to invasive infections such as those leading to sepsis[3,4]. The incidence of neonatal sepsis and related deaths is higher in low- and middle-income countries (LMICs), which are often under-resourced to prevent, identify and treat sepsis[4]. Neonatal gut microbiota development and composition are shaped by the mother's vaginal and rectal microbiotas at birth and, later, by the clinical and community environment[5]. The use of antibiotics, often β-lactams due to availability

[1]Institute of Infection and Immunity, Cardiff University, Cardiff, UK. [2]Institute of Biomedicine, Department of Medical Sciences, University of Aveiro, Aveiro, Portugal. [3]Ineos Oxford Institute of Antimicrobial Research, Department of Zoology, University of Oxford, Oxford, UK. [4]Centre for Trials Research, Cardiff University, Cardiff, UK. [5]Parasites and Microbes Programme, Wellcome Sanger Institute Hinxton, Hinxton, UK. [6]Division of Medical Care, Boston Children's Hospital, Boston, MA, USA. [7]Department of Epidemiology, Harvard T.H. Chan School of Public Health, Boston, MA, USA. [8]Department of Pediatrics, St Paul's Hospital Millennium Medical College, Addis Ababa, Ethiopia. [9]Department of Obstetrics and Gynecology, St Paul's Hospital Millennium Medical College, Addis Ababa, Ethiopia. [10]Department of Microbiology, Immunology and Parasitology, St Paul's Hospital Millennium Medical College, Addis Ababa, Ethiopia. [11]Division of Bacteriology, ICMR-National Institute of Cholera and Enteric Diseases, Kolkata, India. [12]Department of Obstetrics & Gynecology, IPGMER & SSKM Hospital, Kolkata, India. [13]Department of Neonatology, IPGMER & SSKM, Kolkata, India. [14]National Hospital, Abuja, Nigeria. [15]Murtala Muhammad Specialist Hospital, Kano City, Nigeria. [16]Federal Medical Centre Jabi, Abuja, Nigeria. [17]Department of Microbiology, Medway Maritime Hospital NHS Foundation Trust, Gillingham, UK. [18]Aminu Kano Teaching Hospital, Kano, Nigeria. [19]54gene, Lagos, Nigeria. [20]Bayero University, Kano, Nigeria. [21]Department of Microbiology, Quaid-i-Azam University, Islamabad, Pakistan. [22]Pakistan Institute of Medical Sciences, Islamabad, Pakistan. [23]The National Reference Laboratory, Rwanda Biomedical Centre, Kigali, Rwanda. [24]Unit of IPC, Stellenbosch University, Cape Town, South Africa. [25]Infection Control Africa Network, Cape Town, South Africa. [26]Department of Global Health, Stellenbosch University, Cape Town, South Africa. [27]Division of Medical Microbiology, Stellenbosch University, Cape Town, South Africa. [28]National Health Laboratory Service, Tygerberg Hospital, Cape Town, South Africa. [29]These authors contributed equally: M. J. Carvalho, K. Sands. *A list of members and their affiliations appears in the Supplementary Information. ✉e-mail: mjcarvalho@ua.pt

and cost[6], perturbs the gut microbiome and can modulate bacterial populations that have a negative impact on neonatal development. Gibson et al. and other studies from Dantas's group, primarily from high-income countries, have demonstrated that antibiotic therapy in preterm infants can dramatically affect the gut microbiome[7–9].

Large-scale multi-national studies using molecular methods to assess the carriage of ARGs among maternal and neonatal microbiota in LMICs are non-existent. BARNARDS is a network of 12 clinical sites across 7 LMICs in Africa and south Asia aiming to assess the incidence, prevalence, risk factors, bacterial causes and burden of AMR in neonatal sepsis (https://www.ineosoxford.ox.ac.uk/research/barnards). The genomic characterization of BARNARDS' sepsis isolates has already been discussed[10], as well as their resistance profiles to β-lactam and aminoglycoside antibiotics, suggesting that the World Health Organization (WHO) may need to revise their antibiotic guidelines for neonatal sepsis within LMICs, where antibiotic resistance to current therapeutic recommendations is extremely high[6].

In the present study, we characterize the Gram-negative gut microbiota of mothers and septic/non-septic neonates carrying clinically important extended-spectrum β-lactamases (ESBLs) and carbapenemase genes. We investigated statistical associations across maternal, neonatal, living environment and hospital environment domains and carriage of ESBLs and carbapenemase genes. In addition, we determined associations between neonatal/maternal carriage of ARGs, and sociodemographic and clinical environment traits. Furthermore, using whole-genome sequencing (WGS), we characterized common Gram-negative bacteria (GNBs) carrying carbapenemase genes, detailing specific variants and plasmid types across the different study sites.

## Results

**Prevalence of β-lactamase genes among mothers and neonates.**
Overall, BARNARDS recruited 35,040 mothers and their respective neonates ($n = 36,285$). In the present study, 18,148 rectal swabs were analysed to assess the presence of clinically important β-lactamases in the mothers' and neonates' gut microbiota, using the $bla_{CTX-M-15}$-like gene as a marker for the presence of ESBLs and $bla_{NDM}$, $bla_{KPC}$ and $bla_{OXA-48}$-like genes, as markers of carbapenemase genes (Fig. 1).

Among 2,931 neonatal rectal swabs (BRs) analysed (our protocol aimed at collecting rectal swabs from clinically diagnosed septic neonates aged ≥7 d; however, frequently, samples were collected independently of age and all were included in the present study); 626 were from neonates with biological sepsis (BS) and 2,305 were from non-BS (NoBS) cases. The $bla_{CTX-M-15}$, $bla_{NDM}$, $bla_{KPC}$ and $bla_{OXA-48}$-like genes were detected in 56.1% (within 65% of BS and 54% of NoBS), 18.5% (within 24% BS and 17% of NoBS), 0% and 4.1% (within 10% of BS and 2% of NoBS) of BRs, respectively. The prevalence of all genes was higher in south Asian countries (63.0% $bla_{CTX-M-15}$, 34.7% $bla_{NDM}$ and 8.0% $bla_{OXA-48}$-like genes) compared with African countries (49.9% $bla_{CTX-M-15}$, 4.1% $bla_{NDM}$ and 0.6% $bla_{OXA-48}$-like genes). The gene $bla_{KPC}$ was not detected among BRs (Fig. 2a).

From 15,217 mothers' rectal swabs (MRs) analysed, 1,299 were from mothers of neonates with BS, 13,850 from mothers of NoBS neonates and 68 from mothers with a multiple pregnancy, whose neonates had different sepsis outcomes (BSyn). From these, 47.1% (detected within 54.4% BS, 46.4% NoBS and 50% BSyn), 4.6% (detected within 6.93% BS, 4.40% NoBS and 1.47% BSyn), 0.05% (NoBS) and 1.6% (detected within 1.92% BS, 1.57% NoBS and 4.41% BSyn) carried $bla_{CTX-M-15}$, $bla_{NDM}$, $bla_{KPC}$ and $bla_{OXA-48}$-like genes, respectively. A higher prevalence of genes was seen in MRs from south Asian countries (60.7% $bla_{CTX-M-15}$, 8.4% $bla_{NDM}$, 0.1% $bla_{KPC}$ and 2.4% $bla_{OXA-48}$-like genes) compared with African countries (35.7% $bla_{CTX-M-15}$, 1.3% $bla_{NDM}$, 0% $bla_{KPC}$ and 1.0% $bla_{OXA-48}$-like genes; Fig. 2b).

We found the prevalence of $bla_{CTX-M-15}$ among MRs and BRs (Fig. 2ab) to be higher than previously reported[11–17]. The prevalence of $bla_{OXA-48}$-like genes in our African sites was similar to or lower than that of other studies[18–20]. The $bla_{OXA-48}$-like genes are reportedly widespread throughout south Asia[21], but in our study this was observed only in Pakistan among BRs. It is interesting that $bla_{NDM}$ prevalence was higher than previously reported in Pakistan[14,22,23], India[11,24,25] and Bangladesh[26]. Previous reports of $bla_{NDM}$ neonatal carriage in Africa are few and show low-frequency rates among children and pregnant women[12,20,27–29]. Although $bla_{KPC}$ is widely disseminated throughout America and Europe, it is not common in south Asia or Africa[11,12,18,21,24,25], as affirmed by the present study.

We analysed the neonate's age at the time of BR collection against carriage of ARGs and found that, from day 0, ARGs were consistently found among BRs (Fig. 2c), regardless of whether delivery was via caesarean section (CS) or spontaneous vaginal delivery (SVD) and whether or not neonates developed BS (Extended Data Fig. 1a–d). A steady decrease was observed for the prevalence of $bla_{NDM}$ (53.7% to 27.7%) and $bla_{OXA-48}$-like genes (35.4% to 0%) genes among the Asian samples through the first 14 d of life (Fig. 2c), independent of type of delivery or sepsis outcome (Extended Data Fig. 1a,c,d).

There were higher rates of ARG carriage in BS Asian neonates during the first 14 d of life (BS: $bla_{CTX-M-15}$ 80% (121/152), $bla_{NDM}$ 54% (82/152) and $bla_{OXA-48}$-like genes 29% (44/152); NoBS: $bla_{CTX-M-15}$ 57% (250/441), $bla_{NDM}$ 33% (145/441) and $bla_{OXA-48}$-like genes 7% (33/441)), which was also seen for African neonates, although with substantially lower differences (BS: $bla_{CTX-M-15}$ 58% (139/239), $bla_{NDM}$ 5% (12/239) and $bla_{OXA-48}$-like genes 1% (3/239); NoBS: $bla_{CTX-M-15}$ 41% (274/674), $bla_{NDM}$ 3% (21/674) and $bla_{OXA-48}$-like genes 1% (4/674); Extended Data Fig. 1a,b).

Similarly, among neonates born by CS in Asia, the rates of ARG carriage during the first 14 d of life were higher (CS: $bla_{CTX-M-15}$ 69% (178/259), $bla_{NDM}$ 44% (115/259) and $bla_{OXA-48}$-like genes 19% (49/259); SVD: $bla_{CTX-M-15}$ 58% (193/334), $bla_{NDM}$ 34% (112/334) and $bla_{OXA-48}$-like genes 8% (28/334)). This was not seen in neonates from Africa where the carriage of ARGs during the same period was similar for SVD ($bla_{CTX-M-15}$ 46%, 272/595; $bla_{NDM}$ 3%, 20/595; $bla_{OXA-48}$-like genes 1%, 7/595) and CS-delivered babies ($bla_{CTX-M-15}$ 43%, 131/303; $bla_{NDM}$ 4%, 12/303; $bla_{OXA-48}$-like genes 0%, 0/303; Extended Data Fig. 1c,d).

**Bacterial diversity in maternal and neonatal gut microbiota.** In total, 1,072 GNB isolates harbouring carbapenemase genes were recovered (Extended Data Fig. 2). From 412 BRs, we characterized 556 carbapenemase-positive bacteria (CPBs) comprising 33 species/genera with 9 isolates unidentified (Extended Data Fig. 2a). *K. pneumoniae* ($n = 161$), *E. coli* ($n = 132$) and *E. cloacae* complex ($n = 92$) were most common, accounting for 69.6% ($n = 378/543$) and 80.6% ($n = 54/67$) of positive isolates for $bla_{NDM}$ and $bla_{OXA-48}$-like genes. *K. pneumoniae* and *E. coli* were the predominant concomitant carriers of $bla_{NDM}$ and $bla_{OXA-48}$-like genes ($n = 46/54$).

Among 378 MRs, 516 CPBs from 37 distinct species/genera were characterized, with 63 isolates unidentified (Extended Data Fig. 2b). *E. coli* ($n = 132$), *K. pneumoniae* ($n = 50$) and *E. cloacae* complex ($n = 45$) were the most common, altogether accounting for 42.2% ($n = 193/457$) and 58.5% ($n = 48/82$) of positive isolates for $bla_{NDM}$ and $bla_{OXA-48}$-like genes. *K. pneumoniae* and *E. coli* were the major carriers of $bla_{NDM}$ and $bla_{OXA-48}$-like genes concomitantly ($n = 14/21$).

We found a wider CPB species diversity among BRs and MRs than previously described, where most were *K. pneumoniae*, *E. coli* and *E. cloacae* complex[14,28,29]. Evidence shows premature birth dramatically influences species richness and composition in the first months of life, enriching for *E. coli*, *E. cloacae* and *Klebsiella* sp.[30]. The 2,931 samples analysed in the present study were from 2,011

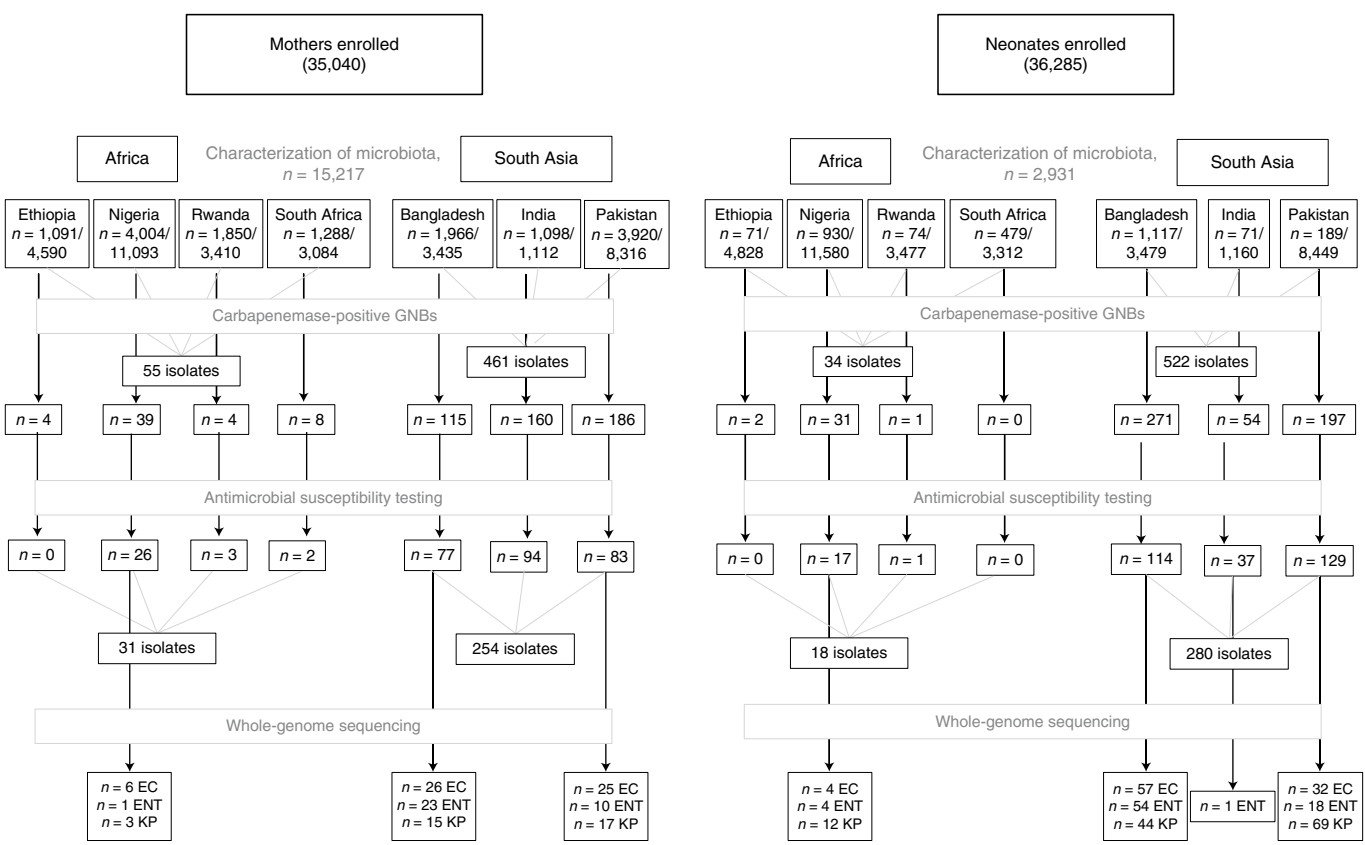

**Fig. 1 | Total number of rectal samples collected from mothers and neonates and characterized.** Diagram detailing the total number of mother and neonate rectal samples collected and screened for the presence of $bla_{CTX-M-15}$, $bla_{NDM}$, $bla_{KPC}$ and $bla_{OXA-48}$-like genes, the number of Gram-negative isolates carrying carbapenemase genes, the number of isolates tested for antibiotic susceptibility and the number of EC (*E. coli*), ENT (*E. cloacae* complex) and KP (*K. pneumoniae*) isolates characterized by WGS and bioinformatics analysis. Isolates for WGS were chosen after culture on VE (vancomycin, ertapenem) agar. Recoverable isolates after −80 °C preservation were selected for gDNA extraction and WGS.

term (69%), 736 (25%) preterm and 147 (5%) post-term neonates (1% clinical data missing).

Antibiotic resistance profiles (Extended Data Fig. 3a) were established for 298 BR and 281 MR CPBs. Resistance rates were especially high for amoxicillin (97%), imipenem and ertapenem (both 91%). The gentamicin resistance rate among BR isolates was higher (84%) than for MR isolates (68%). Although high resistance rates were expected due to our selective culture method, and isolates with intrinsic resistances were recovered[31], these findings contrast previous findings of low AMRs in the south Asian community[32].

**Genomic analysis of *E. coli*, *E. cloacae* and *K. pneumoniae*.** The genomic epidemiology of ARGs from the dominant species among MRs and BRs (*E. coli*, *E. cloacae* complex and *K. pneumoniae*) was characterized by whole-genome sequencing (WGS).

From 265 *E. coli*, 150 isolates were sequenced (93 BRs, 57 MRs; Figs. 1 and 3) showing high genomic diversity, with 44 sequence types (STs) including one previously undefined, ST10987. Isolates were scattered across lineages when globally contextualized across *E. coli* from neonatal, animal, clinical and environmental samples[33–37] (Fig. 3). Of 129 *E. coli*-carrying $bla_{NDM}$ genes, the $bla_{NDM-5}$ was prominent ($n = 69$) across many different STs. $bla_{OXA-181}$ was identified in 25 *E. coli* isolates from 15 STs from Bangladesh, Nigeria and Pakistan. ST405 was frequently isolated from Bangladesh ($n = 24$), with SNP analysis revealing distinct clades for each clinical site (Extended Data Fig. 4). Of the 13 ST405 *E. coli* isolates with $bla_{NDM-5}$ on an IncFII plasmid (88,885 bp; Supplementary Table 1) from BK (8 BRs, 5 MRs; the sites' acronyms are detailed in Methods), 2 isolates were from a mother–neonate pair; however, all ($n = 13$) were

within 6 pairwise SNPs (Extended Data Fig. 4). ST405 *E. coli* clones have been isolated from samples from the same neonates at distinct time points, showing the persistence of this lineage in the microbiota[7]. A cluster of 13 BC BR ST4684 *E. coli* isolates were isolated within a 5-month period in 2016 (Fig. 3), all containing $bla_{NDM-1}$-like genes on an IncX3 plasmid (57,221 bp; Supplementary Table 1). SNP analysis revealed this cluster to be clonal (0 pairwise SNPs). The 25 *E. coli* isolates carried $bla_{OXA-181}$ (BC $n = 2$, NK $n = 4$, PP $n = 19$; 7 BRs, 18 MRs), often on a ColKP3 plasmid ($n = 22$; Supplementary Table 1). Notably, in ST410 from PP BRs, $bla_{NDM-5}$ and $bla_{OXA-181}$ were concomitantly detected, and dual-carbapenemase ARGs were also found in ST410 from global collections from the National Center for Biotechnology Information (NCBI). Of 150 BARNARDS' CBP *E. coli* isolates, 90 carried $bla_{CTX-M-15}$ (Fig. 3) and 4 different *E. coli* harboured *mcr* (*mcr-1*, $n = 2$ and *mcr-9*, $n = 2$).

From 136 isolates identified by MALDI coupled to time-of-flight mass spectrometry (MALDI-TOF MS) as *E. cloacae* complex, 111 (77 BRs, 34 MRs) were sequenced (Figs. 1 and 4), revealing 34 STs including 5 previously undefined (Fig. 4). The *E. hormaechei* STs ST113, ST171 and ST418 were dominant (Fig. 4) and *E. hormaechei* ST418 harbouring $bla_{NDM-1}$-like genes was recovered predominantly from neonates in BC and BK, with SNP analysis indicating genomic variability between 0 and 1,601 pairwise SNPs (Supplementary Table 1 and Extended Data Fig. 5). SNP analysis of 13 ST68 *E. cloacae* isolates from BC (8 MRs, 5 BRs), recovered during July 2016, suggests that these are very closely related (within 4 pairwise SNPs). The $bla_{NDM-1}$-like gene was most common ($n = 98$ (BC $n = 53$, BK $n = 17$, IN $n = 1$, NN $n = 2$, PC $n = 4$, PP $n = 21$; 72 BRs, 26 MRs)) and associated with 27 different STs, often on an IncN2 or IncA/

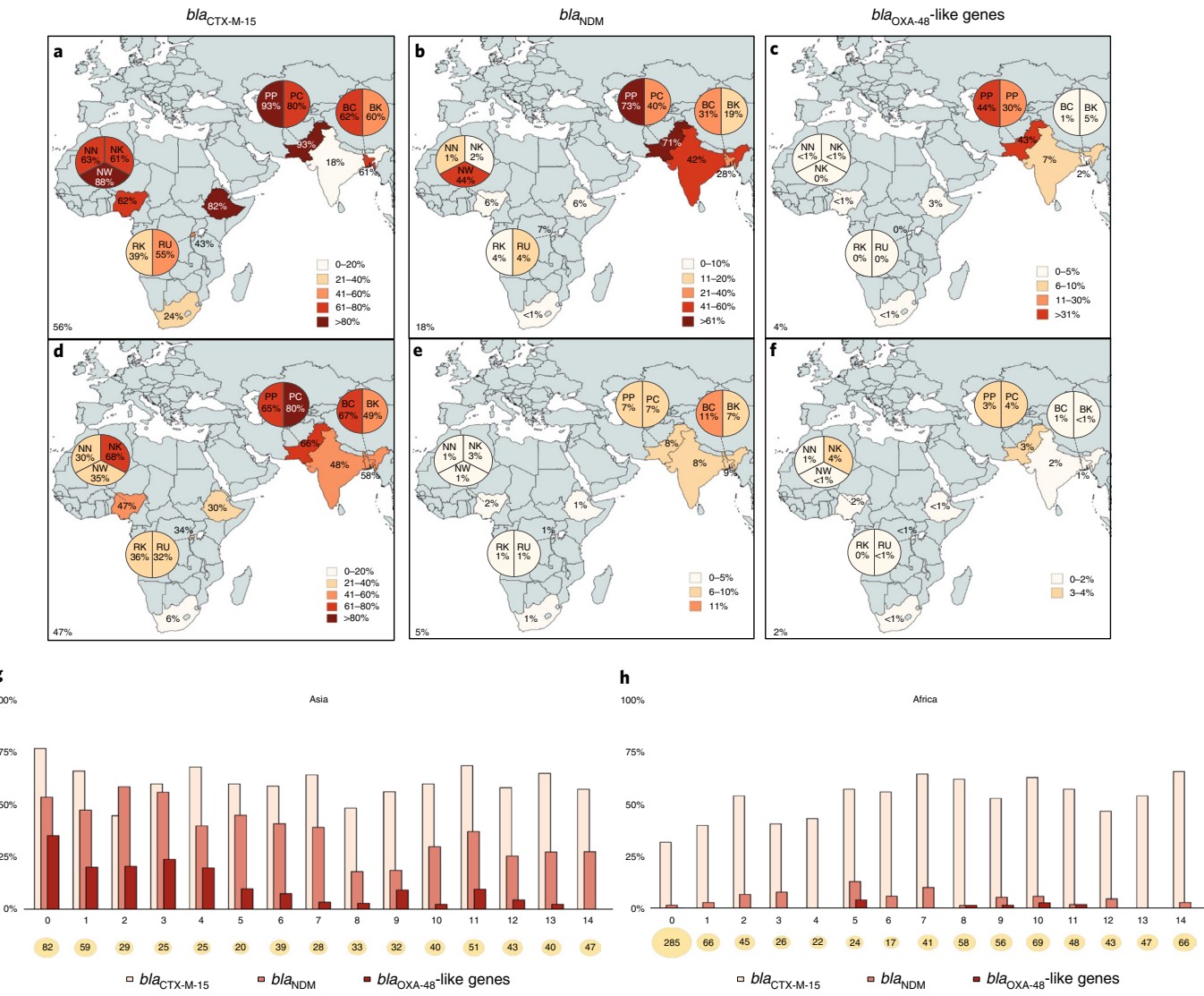

**Fig. 2 | Prevalence of *bla*<sub>CTX-M-15</sub>, *bla*<sub>NDM</sub> and *bla*<sub>OXA-48</sub>-like genes among the rectal swabs of neonates and mothers. a–c**, Prevalence of $bla_{CTX-M-15}$, $bla_{NDM}$ and $bla_{OXA-48}$-like genes among the rectal swabs of neonates. The prevalence of all genes was higher in South-Asian countries compared to African countries, except for $bla_{KPC}$, which was not found among neonates. **d–f**, Prevalence of $bla_{CTX-M-15}$, $bla_{NDM}$ and $bla_{OXA-48}$-like genes among the rectal swabs of mothers. A higher prevalence of genes was seen in MR from South-Asian countries compared to African countries. $bla_{KPC}$ genes were found in three Indian and four Pakistani rectal samples from mothers. The BARNARDS network included the following hospitals: Bangladesh: BC and BK; Ethiopia: ES; India: IN; Nigeria: NN, NW and NK; Pakistan: PP and PC; Rwanda: RU and RK; and South Africa: ZAT. Coloured maps were created using MapChart (https://www.mapchart.net). **g,h**, Carriage of $bla_{CTX-M-15}$, $bla_{NDM}$ and $bla_{OXA-48}$-like genes among neonates' rectal swabs against age of neonates at rectal swab collection per continent: Asia (**g**) and Africa (**h**). The prevalence of each ARG is plotted. The total number of samples collected per day is shown in the circles below the graphs. From day 0, ARGs were detected in the neonatal rectal microbiota. There was a tendency to a decrease in prevalence of $bla_{NDM}$ (53.7% to 27.7%) and $bla_{OXA-48}$-like (35.4% to 0%) genes among the Asian samples through the first 14 d of life.

C2 plasmid (Supplementary Table 1). The $bla_{NDM-5}$-like gene (*n*=6) was largely identified in ST66 isolates from Bangladesh, whereas $bla_{NDM-7}$ *Enterobacter* sp. (*n*=5) was associated with four distinct STs, including previously undefined ST1372 from a Nigerian BR. $bla_{OXA-48}$ variants were not detected in *Enterobacter* spp. In addition, 14 *Enterobacter* spp. isolates concomitantly carried *mcr-9.1* with a variant of $bla_{NDM}$-like gene (PP *n*=9, PC, *n*=4, BC *n*=1).

From 211 isolates from *K. pneumoniae*, 161 were sequenced (125 BRs, 36 MRs; Figs. 1 and 5), including 11 *K. quasipneumoniae* subsp. *quasipneumoniae* and 9 *K. quasipneumoniae* subsp. *similipneumoniae* (Fig. 5). Three PP BR ST15 *K. pneumoniae* isolates possessed the same ST as that of the isolate causing sepsis in the same neonate[10]. We detected 46 STs from which ST11, ST14, ST15 and ST48 were common across the collective phylogeny (Fig. 5). ST11

was predominantly found in Europe carrying $bla_{OXA-48}$ or $bla_{OXA-245}$; however, in our study *n*=6/8 PP ST11 isolates carried $bla_{NDM-7}$ (Extended Data Fig. 6). The $bla_{NDM-1}$ was most frequent (*n*=124; Fig. 5), with 72 from Pakistan of which 4 belonged to previously undefined STs 4980–4983, and in 31 ST15 STs, the $bla_{NDM-1}$ was IncA/C2 or IncN2 plasmid mediated (141,533 bp; Supplementary Table 1). In Bangladesh, 54 *K. pneumoniae* isolates carried $bla_{NDM}$ and, from these, 5 ST14 STs harboured $bla_{NDM-1}$, $bla_{OXA-232}$ and $bla_{CTX-M-15}$. The $bla_{OXA-232}$-like gene was identified on a ColKP3 plasmid in ST14 *K. pneumoniae* (Supplementary Table 1). Of 15 *Klebsiella* spp. isolates from Nigeria, 12 were *K. pneumoniae*, 2 *K. quasipneumoniae* subsp. *similipneumoniae* and 1 *K. quasipneumoniae* subsp. *quasipneumoniae*. Ten STs were detected, including previously undefined ST4979: nine carried $bla_{NDM-1}$ and the remaining six carried $bla_{NDM-7}$.

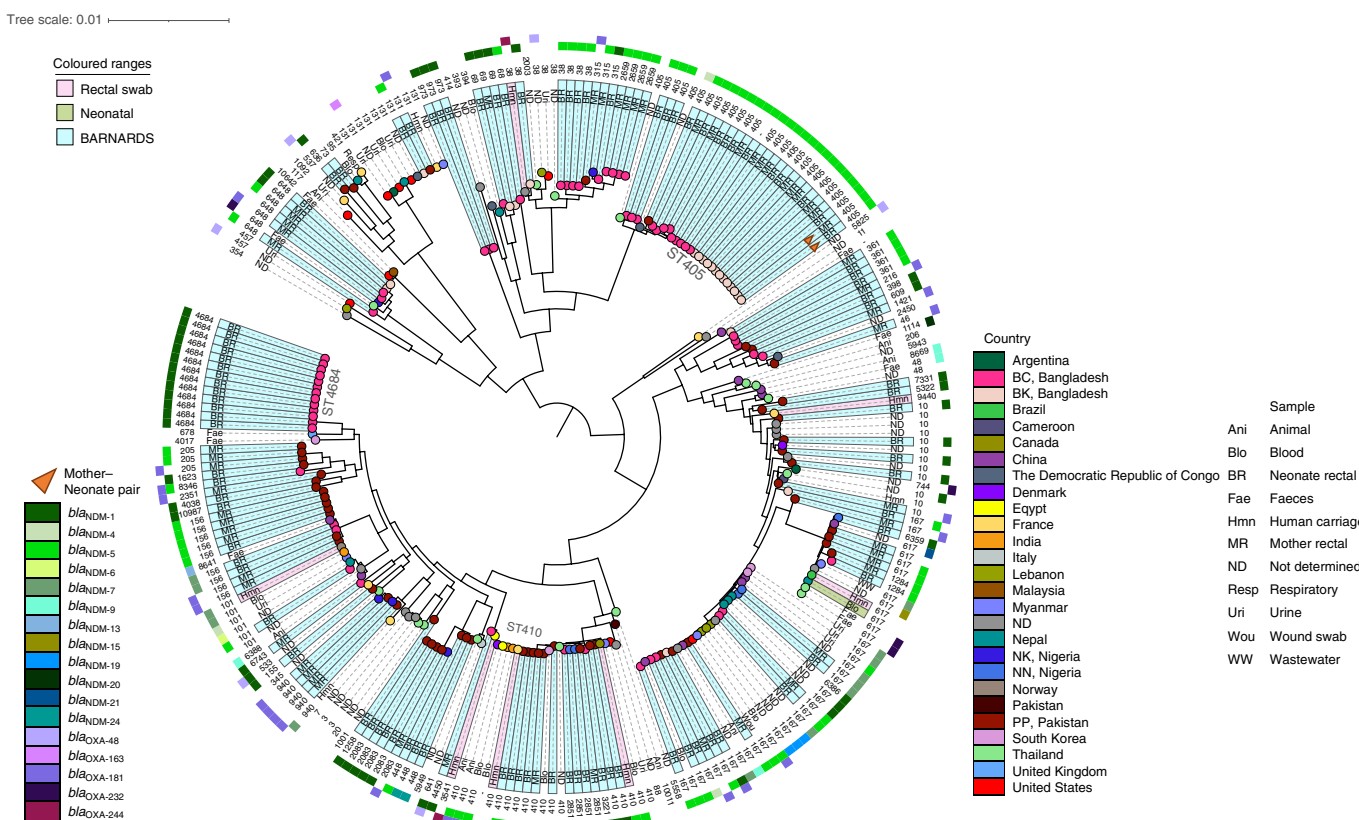

**Fig. 3 | Core genome characterization of *E. coli* isolates.** The phylogenetic tree of 253 *E. coli* genomes, including 150 from BARNARDS and 103 from other studies[11–15], is shown, using Roary (v.3.12.0)[39] and FastTree[40] (v.2.1.11). Isolates are coloured at the endpoint according to country and the outer ring abbreviation is labelled according to the sample source. STs for all isolates are shown in the text after the sample source. The additional two outer rings denote the presence of *bla*_NDM_ and *bla*_OXA-48_-like genes. Clades containing isolates from the present study are highlighted in teal, green clades indicate *E. coli* neonatal sepsis isolates from other studies and pink clades relate to *E. coli* rectal carriage from different studies. Major STs are labelled around the phylogeny and isolates that belong to a mother–neonate pair are denoted by an orange triangle. For site acronyms, see Methods.

**Risk factors for the rectal carriage of β-lactamase genes.** To determine maternal, neonatal, living environment and hospital environment features associated with the carriage of *bla*_CTX-M-15_, *bla*_NDM_ or *bla*_OXA-48_-like genes among the gut microbiota of mothers and neonates, we performed several exploratory univariate and multivariable analyses (Table 1 and Supplementary Table 2). We fitted a multivariable model including WASH (water, sanitation and hygiene)-associated features (Extended Data Fig. 7a), to understand the impact of these indicators in the carriage of the ARGs in the study among MRs. In 2017, 7% (Ethiopia) to 76% (South Africa) of the population in countries of the BARNARDS network used at least basic sanitation services (Supplementary Table 1) and we found that occasional handwashing by the mothers or households supplied with a wastewater network were independent risk factors for carrying *bla*_CTX-M-15_.

Multivariable models showed that occasional handwashing was associated with MR carriage of *bla*_CTX-M-15_ or *bla*_NDM_, whereas frequent handwashing was associated with the carriage of *bla*_OXA-48_-like genes (Table 1 and Supplementary Table 2a). Poor hygiene is a driver for carriage of ARGs[38] and we speculate that deficient hand hygiene, even if frequent, could be associated with the carriage of these ARGs, specially *bla*_CTX-M-15_. We also found that a maternal infection in the 3 months before enrolment in the present study was associated with MR carriage of *bla*_CTX-M-15_ (Table 1 and Supplementary Table 2a). Carriage of ARGs among MRs was associated with the mothers' use of antibiotics in the 3 months before enrolment (Table 1 and Supplementary Table 2a). In similar settings, previous use of antibiotics has been described as a risk factor

for carriage of ESBL producers/MDR isolates[15,38], but this was not supported from findings in other studies[18,29,39]. We did not find an association between neonates' age at time of sampling and carriage of ARGs (Supplementary Table 2b). Previously, increased neonatal age was associated with carriage of ESBL producers[13].

**β-Lactamase gene carriage and birth outcomes.** Our exploratory analysis suggests that planned or emergency CS or premature birth may be associated with the mother's carriage of either *bla*_CTX-M-15_ or *bla*_NDM_. Also, the odds of having preterm premature rupture of membranes (PPROM) were higher for mothers carrying *bla*_CTX-M-15_ or *bla*_NDM_, whereas the odds of having perinatal asphyxia or a breech birth were higher for mothers carrying *bla*_CTX-M-15_ (Supplementary Table 2c).

As the hospital environment has been associated with carriage of ARGs[19,29], we investigated whether neonates born within clinical sites (birth cohort) were more likely to carry β-lactamase genes compared with those born elsewhere (admission cohort, admitted with suspected sepsis). We found that the odds of carrying *bla*_NDM_ were higher for neonates from the birth cohort (Supplementary Table 2d). We did not find significant associations between birth healthcare/environment factors and the carriage of these ARGs among the birth cohort (Extended Data Fig. 7b and Supplementary Table 2a). Univariate analysis including neonates from both cohorts showed that those born by emergency CS were more likely to carry *bla*_CTX-M-15_ in agreement with other studies[13,39] and neonates born after PPROM had higher odds of carrying *bla*_OXA-48_-like genes (Supplementary Table 2e).

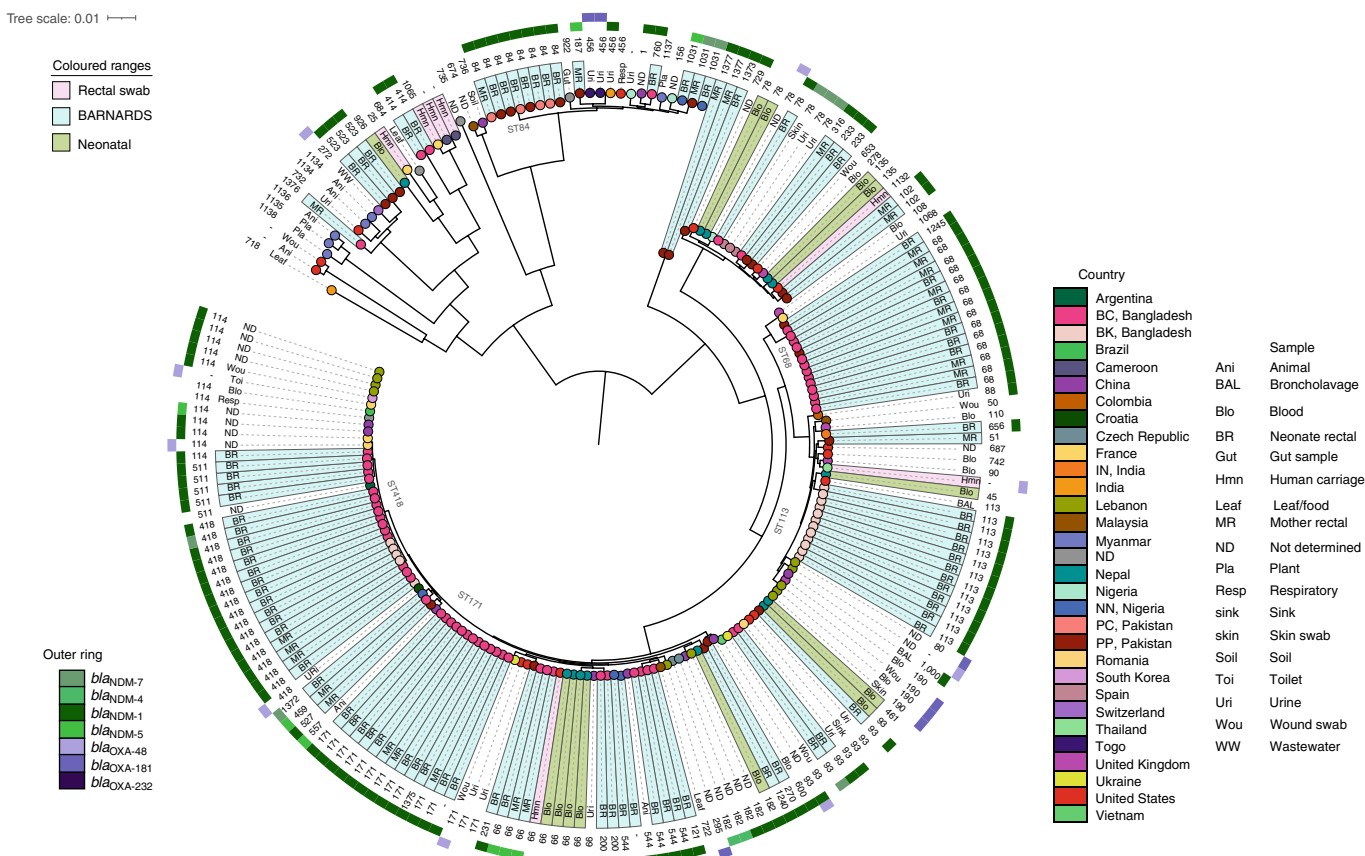

**Fig. 4 | Core genome characterization of *Enterobacter* spp. isolates.** The phylogenetic tree of 209 *Enterobacter* spp. genomes including 111 from BARNARDS and 98 from other studies[12,13,15] is shown, using Roary (v.3.12.0)[55] and FastTree (v.2.1.11)[56]. Isolates are coloured at the endpoint according to country and the outer ring abbreviation is labelled according to the sample source. STs for all isolates are shown in the text after the sample source. The additional two outer rings denote the presence of $bla_{NDM}$ and $bla_{OXA-48}$-like genes. Clades containing isolates from the present study are highlighted in teal, green clades indicate *Enterobacter* spp. neonatal sepsis isolates from other studies and pink *Enterobacter* spp. rectal carriage from different studies. Major STs are labelled around the phylogeny. For site acronyms, see Methods.

**β-Lactamase gene carriage and neonatal sepsis.** We found that colonization of the mother's gut with $bla_{CTX-M-15}$ or $bla_{NDM}$ positive microbiota was associated with the development of BS in the neonate, and this may be due to the mother transmitting MDR pathogens to her neonate during or after labour and birth, potentially leading to neonatal BS[40]. Neonates carrying $bla_{CTX-M-15}$ or $bla_{OXA-48}$-like genes in their microbiota were more likely to have BS compared with non-carriers (Supplementary Table 2f).

## Discussion

In the present study, we report high carriage of $bla_{CTX-M-15}$, $bla_{OXA-48}$-like genes and $bla_{NDM}$ among the rectal microbiota of mothers and neonates with either suspected or confirmed BS. Carriage of genes was higher for neonates compared with mothers, as previously reported[15,17], particularly, for $bla_{NDM}$ in samples from Bangladesh, Nigeria and Pakistan (Fig. 2). We speculate that, because most of these neonates have been administered antibiotics, if presenting with clinical sepsis, antibiotic selection pressure favoured resistant bacteria, as described previously[7,15,18,19]. We highlighted the carriage of ARGs in neonates from the very early hours after birth, irrespective of delivery type or sepsis outcome, which may have been underpinned by antibiotic therapy after acquisition from the mother and/or environment.

Our results further highlight the importance of access to safe water, sanitation and good hygiene to reduce the mortality rate. WASH-related factors might have been associated with the carriage of $bla_{CTX-M-15}$ among MRs, and the carriage of $bla_{CTX-M-15}$ or $bla_{NDM}$

with poor birth outcomes and neonatal sepsis. Similarly, previous maternal infection and use of antibiotics were associated with the carriage of β-lactamase genes among MRs, and further associated with more adverse birth outcomes and neonatal sepsis. In addition, our exploratory analysis suggested that complicated births such as PPROM and clinical interventions such as a CS could be associated with neonatal ARG carriage and neonatal sepsis. We acknowledge that all statistical analyses performed are exploratory and not causal. Other uncharacterized covariates such as medical history and/or socioeconomic factors are also likely to add to the AMR burden and poor health outcomes.

The genomic analysis unveiled the existence of indistinguishable *E. coli* isolates from MRs and BRs, suggesting transmission from mother to neonate during or after labour. Furthermore, *K. pneumoniae*, which was found to be the most common cause of sepsis in neonates enrolled in BARNARDS[10], was also the most prevalent isolate among BRs. SNP analysis revealed three cases where *K. pneumoniae* BRs and sepsis isolates from the same neonate were very closely related, indicating that transmission events either in the clinical setting or in the newborn gut microbiota might have occurred. In addition, SNP analysis of *E. coli*, *E. cloacae* and *K. pneumoniae* genomes from neonates attending the same clinical sites indicated clonal cases. Moreover, ARGs, and in particular $bla_{NDM}$-like genes, were found in different plasmids (IncX3 in Bangladesh; IncN2 or IncA/C2 in Pakistan), emphasizing a diverse dissemination of MDR pathogens harbouring ARGs. The identification of ARGs in the microbiota of neonates from the first hours of life indicates

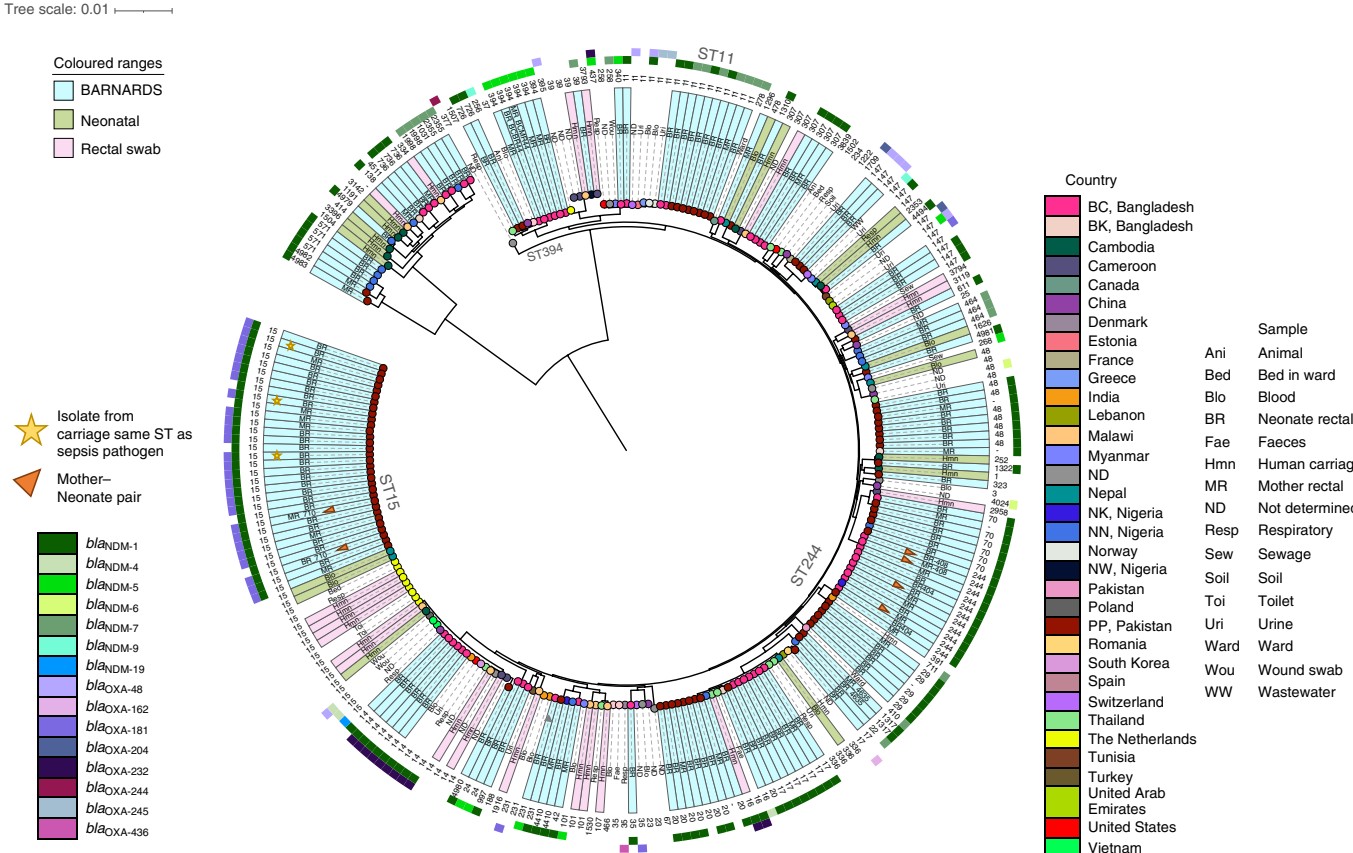

**Fig. 5 | Core genome characterization of *K. pneumoniae* isolates.** The phylogenetic tree of 268 *K. pneumoniae* genomes, including 161 from BARNARDS and 107 from other studies[12–20], is shown, using Roary (v.3.12.0)[55] and FastTree (v.2.1.11)[56]. Isolates are coloured at the endpoint according to country and the outer ring abbreviation is labelled according to the sample source. STs for all isolates are shown in the text after the sample source. The additional two outer rings denote the presence of $bla_{NDM}$ and $bla_{OXA-48}$-like genes. Clades containing isolates from the present study are highlighted in teal, green clades indicate *K. pneumoniae* neonatal sepsis isolates from other studies and pink *K. pneumoniae* rectal carriage from different studies. Major STs are labelled around the phylogeny and isolates that belong to a mother–neonate pair are denoted by an orange triangle. Any carriage isolates sequenced in the present study that are genetically similar to the isolate recovered from the corresponding neonatal blood culture[10] are denoted by a yellow star. For site acronyms, see Methods.

that initial colonization occurred at birth through contact with the mother and/or the hospital environment (for example, CS), and WGS analysis shows evidence for both routes of neonatal gut colonization with AMR microbiota. These findings support the need for future studies assessing mother/non-clinical environment–neonate transmission of ARGs, to improve infection prevention control measures in LMICs and study the development of the gut microbiome and resistome.

We chose β-lactamase genes as markers given the WHO recommendations of first- and second-line treatments for neonatal BS: ampicillin and ceftazidime, respectively. We acknowledge that there are many ESBL and carbapenemase genes; our selection was based on clinical importance and epidemiology. We had limitations with the retrospective recovery of *E. coli*, *K. pneumoniae* and *E. cloacae* for WGS due to loss of the carbapenemase gene and this may be due in part to freeze–thaw effects resulting in plasmid loss. Antibiotic susceptibility testing was performed on a proportion of recoverable isolates. One colony per phenotypically similar bacterial colony was selected for confirmation of the presence of ARGs. We acknowledge the limitation of a microbial culture-based approach that may not have detected the presence of multiple strains of the same species from a single sample. We did not perform a longitudinal study collecting samples across different time points to assess acquisition/ loss of ARGs during the present study, or collect history of antibiotic exposure to understand the effects of antibiotic treatment

on the neonatal microbiota. It should be noted that approximately 40% of neonatal samples discussed in the present study originate from Bangladesh (1,117/2,931, Fig. 1) and a limitation of the present study is the under-representation of available neonatal samples from other LMICs. The sociodemographic data collected and used for statistical analysis was largely self-reported and we acknowledge that this approach may have been subject to social desirability and recall bias.

In summary, the present study highlighted the prevalence of the carriage of important β-lactamase genes among the microbiota of mothers and their neonates with either suspected or confirmed sepsis in LMICs. We demonstrated the presence of ARGs in the gut microbiota from the first hours of life that has seldom been reported. We showed that poorer WASH indicators, use of antibiotics and previous infection were probably associated with gut microbiota carriage of $bla_{CTX-M-15}$, $bla_{NDM}$ or $bla_{OXA-48}$-like genes. Furthermore, the carriage of these genes was most probably associated with neonatal sepsis and adverse birth outcomes. By performing WGS on *E. coli*, *K. pneumoniae* and *E. cloacae* complex isolates, we unveiled the major lineages present in the guts of mothers and neonates in LMIC sites and their AMR-related genetic machinery. WGS showed relatedness between isolates from mothers' and neonates' microbiota and between gut microbiota and those isolates causing neonatal sepsis, warranting future studies. These results contribute to AMR surveillance in previously unexplored settings and populations to

**Table 1 | Exploratory multivariable statistical analysis to understand associations between sociodemographic and clinical data and maternal and neonatal carriage of ARGs**

| A) Mothers' handwashing frequency | $bla_{CTX-M-15}$ | | $bla_{NDM}$ | | $bla_{OXA-48}$-like | |
|---|---|---|---|---|---|---|
| | UV | MV | UV | MV | UV | MV |
| | (OR; 95% CI; *P* value) | | (OR; 95% CI; *P* value) | | (OR; 95% CI; *P* value) | |
| Occasional | 1.00 | 1.00 | 1.00 | 1.00 | 1.00 | 1.00 |
| Frequent | **0.86; 0.79–0.93;<0.001** | **0.85; 0.78–0.93;<0.001** | 0.86; 0.72-1.03; 0.109 | **0.81; 0.67–0.98; 0.027** | **1.56; 1.17–2.09; 0.003** | **1.49; 1.10–2.02; 0.010** |
| B) Maternal infection in the 3 months before enrolment | $bla_{CTX-M-15}$ | | $bla_{NDM}$ | | $bla_{OXA-48}$-like | |
| | UV | MV | UV | MV | UV | MV |
| | (OR; 95% CI; *P* value) | | (OR; 95% CI; *P* value) | | (OR; 95% CI; *P* value) | |
| No | 1.00 | 1.00 | 1.00 | | 1.00 | |
| Yes | **1.28; 1.03–1.59; 0.029** | **1.26; 1.01–1.58; 0.040** | 1.22; 0.72–2.07; 0.466 | | 0.77; 0.24–2.47; 0.655 | |
| C) Maternal use of antibiotics in the 3 months before enrolment | $bla_{CTX-M-15}$ | | $bla_{NDM}$ | | $bla_{OXA-48}$-like | |
| | UV | MV | UV | MV | UV | MV |
| | (OR; 95% CI; *P* value) | | (OR; 95% CI; *P* value) | | (OR; 95% CI; *P* value) | |
| No | 1.00 | 1.00 | 1.00 | 1.00 | 1.00 | 1.00 |
| Yes | **1.39; 1.23-1.58;<0.001** | **1.36; 1.20-1.55;<0.001** | **1.79; 1.32-2.43;<0.001** | **1.71; 1.24–2.35; 0.001** | **1.83; 1.19–2.80; 0.006** | **1.69; 1.09-2.61; 0.018** |

Confounder variables Part A: education status of mother; household income equal to or greater than country average; residence water supply; access to soap; frequency of solid waste collection; access to wastewater network; type of residence. Confounder variables Part B: age of mother; maternal visit to hospital in the 12 months before enrolment; maternal attendance to private healthcare in the 3 months before enrolment; maternal visit to traditional healer in the 3 months before enrolment; household income equal to or greater than country average; education status of mother; type of toilet in residence; mothers' handwashing frequency; residence access to wastewater network; frequency of solid waste collection; residence water supply; mother immunocompromised; mother with diabetes. Confounder variables Part C: mother immunocompromised; mother with diabetes; maternal visit to hospital in the 12 months before enrolment; maternal attendance to private healthcare in the 3 months before enrolment; maternal visit to traditional healer in the 3 months before enrolment; household income equal to or greater than country average; education status of mother. Analyses were performed to understand: Part A: association between the mother's handwashing frequency (explanatory variable) and maternal carriage of ARGs (outcome) and controlled for the variables described in the tabl legend; Part B: association between maternal infection in the 3 months before enrolment in the study (explanatory variable) and maternal carriage of ARGs (outcome), and controlled for the variables shown in the table legend; and Part C: association between maternal usage of antibiotics in the 3 months before enrolment in the study (explanatory variable) and maternal carriage of ARGs (outcome), and controlling for the variables depicted in the table legend. The z-tests were used from multivariable logistic regression models and statistical tests were two sided. For all models, UV = univariate analysis (all data in Supplementary Table 6), MV = multivariable analysis.CI, confidence interval; OR, odds ratio. Mother handwashing frequency, MV = full multivariable model including type of residence; the results for the full model excluding type of residence are in Supplementary Table 7. Maternal infection in the 3 months before enrolment, MV; the results for full multivariable model are in Supplementary Table 7. Statistically significant *P* values are in bold.

inform national action plans on better infection prevention practices and to reduce the burden of AMRs in LMICs.

## Methods

**Settings, ethics, participants and study design.** In the present study, the term 'neonates' is used to include all neonates and infants (aged >28 to 60 d) enrolled. The BARNARDS network included: Bangladesh: Chittagong Maa-O-Shishu Hospital, Chattogram (BC) and Kumudini Women's Medical College, Mirzapur (BK); Ethiopia: St Paul's Hospital Millennium Medical College, Addis Ababa (ES); India: Division of Bacteriology, ICMR-National Institute of Cholera and Enteric Diseases Beliaghata and Institute of Post-Graduate and Medical Education & Research, Kolkata (IN); Nigeria: National Hospital Abuja (NN), Wuse District Hospital (NW) Abuja and Murtala Mohammad Specialist Hospital, Kano (NK); Pakistan: Pakistan Institute of Medical Sciences, Islamabad (PP) and Bhara Kahu Rural Health Centre, Bhara Kahu (PC); Rwanda: University Central Hospital of Kigali, Kigali (RU) and Kabgayi Hospital, Kabgayi (RK); and South Africa: Tygerberg Hospital, Cape Town (ZAT). Standard operating procedures were designed and adhered to throughout the network (https://www.ineosoxford. ox.ac.uk/research/barnards), and ethical approval was obtained from local ethics committees before the start of the study (Supplementary Table 2). The site abbreviation names were commonly used throughout this publication; however, the country name was used when the results were applicable to all sites within that country.

From November 2015 to November 2017, women in labour (preferably) or immediately post partum were recruited prospectively following their consent and their neonate(s) followed up for the first 60 d of life or until study withdrawal/neonatal death. For neonates lost to follow-up, the information available at the last follow-up point was considered. In addition, neonates who presented to clinical sites with clinically suspected sepsis in the first 60 d of life were recruited (with their mothers) on consent and followed up as described. Demographic and clinical data were collected on pretested study forms by trained researchers. The definitions for clinically suspected sepsis are detailed in https://www.ineosoxford.ac.uk/research/barnards.

BS was assigned to neonates with blood culture-positive sample(s), as described elsewhere[10].

Further details of the study design and sociodemographic and clinical characteristics of mothers and neonates are described elsewhere[41].

According to the established protocol, rectal samples were to be taken from all mothers on recruitment and from neonates aged ≥7 d up to 60 d with clinically suspected sepsis. However, during the course of the present study, rectal samples were taken from neonates with clinically suspected sepsis from 0 d of life onward and these samples were also characterized and included in the present study. For this, sterile swabs in Amies Transport Medium with charcoal (Liofilchem) were used as described in https://www.ineosoxford.ox.ac.uk/research/barnards. Swabs were maintained at 4 °C until transfer to Cardiff University (CU) under UN3733 regulations at room temperature.

**Ethics approval and consent to participate.** Ethical approval was obtained at each of the seven participating countries (Supplementary Table 2). Bangladesh: Ethical Review Committee, Bangladesh Institute of Child Health (BICH-ERC-4/3/2015); Ethiopia: Boston Children's Hospital (IRB-P00023058); India: Institutional Ethics Committee, National Institute of Cholera and Enteric Diseases and Institute of Post Graduate Medical Education and Research, IPGME&R Research Oversight Committee (A-I/2016-IEC and Inst/IEC/2016/508); Nigeria: Kano State Hospitals Management Board (8/10/1437AH), Health Research Ethics Committee (HREC) and National Hospital, Abuja (NHA/EC/017/2015); Pakistan: Shaheed Zulfiqar Ali Bhutto Medical University, Pakistan Institute of Medical Sciences (PIMS), Islamabad (ref. no. NA, signed letter from T. Hazir); Rwanda: Republic of Rwanda, National Ethics Committee (No342/RNEC/2015); and South Africa: Stellenbosch University and Tygerberg Hospital, Research projects, Western Cape Government (N15/07/063). All approval dates are listed in Supplementary Table 2. In local languages, research nurses provided mothers with study information and collected consent for mother and/or neonatal enrolment. Informed consent was obtained in writing unless this was not possible (due to literacy barriers) and oral consent was collected from the mothers by trained researchers. Oral consent was documented by the participant signing/marking the consent form.

**Gut microbiota characterization.** On arrival to CU, rectal swabs were stored at 4 °C until processing. Mothers' rectal samples were processed on a ratio of a minimum of 1:3 BS:NoBS-related sample per site[10] and all neonatal rectal swabs were processed. Swabs were streaked on three chromogenic agar medium plates (Liofilchem) supplemented with either vancomycin (10 mg l⁻¹), vancomycin and cefotaxime (VC, 10 mg l⁻¹ and 1 mg l⁻¹, respectively) and vancomycin and ertapenem (VE, 10 mg l⁻¹ and 2 mg l⁻¹, respectively) to select for cefotaxime-resistant GNBs (indicative of the presence of ESBL producers) and ertapenem-resistant GNBs (indicative of the presence of carbapenemase producers).

The GNB microbiota grown on VC and VE plates was scrutinized for the presence of $bla_{CTX-M-15}$ and of $bla_{NDM}$, $bla_{KPC}$ and $bla_{OXA-48}$-like genes, correspondingly, by PCR/multiplex-PCR using the Illustra PuReTaq Ready-To-Go PCR Beads (GE Healthcare) in a Gene Touch Thermal Cycler (Hangzhou Bioer Technology Co., Ltd). PCR conditions, primers (Eurofins) and control strains are described in Supplementary Table 3. Amplicons were subjected to electrophoresis in a 1% agarose (Sigma-Aldrich) gel at 300 V for 35 min in 1× Tris/borate/EDTA buffer containing 25 μl of ethidium bromide. All bacterial cultures were preserved in TS/72 beads (Technical Service Consultants) at −80 °C.

Phenotypically distinct bacterial colonies in VE plates from multiplex-PCR-positive samples were selected and pure cultures obtained by repeated isolation of individual colonies in the same medium. All isolates were subjected to multiplex-PCR and those with a positive result for any of the carbapenemase genes in the study were identified by MALDI-TOF MS (Bruker Daltonik GmbH) and preserved as mentioned before until further analysis. The workflow for sample collection and processing is shown in Extended Data Fig. 8. Due to the high prevalence of $bla_{CTX-M-15}$-positive isolates, we did not scrutinize samples for $bla_{CTX-M-15}$-positive isolates.

Indian samples were processed locally using the same methodology, except for bacterial isolate identification, which was done using Enterosystem 18R (Liofilchem) and the VITEK 2 Compact Automated System.

BSyn sample results were included in both BS and NoBS groups, because the same mother had neonates with different BS statuses. Hence, the results for each of the 68 samples were accounted for twice.

**AMR profiles.** Antibiotic susceptibility testing was performed using the disk diffusion method for a subset of isolates ($n = 584$) according to EUCAST v.9 guidelines (2019)[31], using appropriate control strains to test quality control. Antibiotics tested were tigecycline (TGC, 15 μg), fosfomycin (FOS, 200 μg), ciprofloxacin (CIP, 5 μg), levofloxacin (LVX, 5 μg), gentamicin (GEN, 10 μg), amikacin (AMK, 30 μg), nitrofurantoin (F, 100 μg), trimethoprim–sulfamethoxazole (SxT, 1.25/23.75 μg), ertapenem (ETP, 10 μg), amoxicillin (AML, 10 μg), amoxicillin–clavulanic acid (AMC, 20/10 μg), piperacillin–tazobactam (TZP, 30/6 μg), CTX (cefotaxime, 5 μg), ceftazidime (CAZ, 10 μg), cefepime (FEP, 30 μg), imipenem (IPM, 10 μg), meropenem (MEM, 10 μg) and aztreonam (ATM, 30 μg). Indian bacterial isolates were tested locally using the same methods. Supplementary Table 4 shows the antibiotics tested and the disk concentrations, control strains used and EUCAST v.9 breakpoint tables used for interpretation of results.

**Genomic analysis of *E. coli*, *E. cloacae* and *K. pneumoniae*.** *K. pneumoniae*, *E. coli* and *E. cloacae* isolates from rectal swabs from Bangladesh, Pakistan and Nigeria were selected for further characterization by WGS and bioinformatics analysis. Genomic (g)DNA extraction and Illumina WGS were performed as described[10].

Briefly, gDNA was extracted using the QIAmp DNA mini-kit (QIAGEN), with an additional RNase step, on the automated QIAcube platform (QIAGEN), and was quantified using the Qubit fluorometer 3.0. Genomic libraries were prepared using Nextera XT v.2 (Illumina), with a bead-based normalization, following the manufacturers' guidelines. A total of 48 isolates is multiplexed per sequencing run to provide a depth of coverage >15×. Paired-end WGS was performed on an Illumina MiSeq using the v.3 chemistry to generate fragment lengths up to 300 bp (600 cycles). For Oxford Nanopore Technology (ONT) sequencing, fresh gDNA was extracted as described above, concentrated using SPRI beads (Mag-Bind TotalPure, Omega) and libraries were generated using the 96-Rapid Barcoding Kit (SQK-RBK110.96; ONT). Sequencing was performed using MinION flow cells (R9.4 and R.10) for a running time of 72 h within MinKnow.

Bioinformatics analysis was performed using a high-performance computing cluster at CU (Advanced Research Computing at Cardiff (ARCCA)) and CLIMB[42]. Paired-end reads (fastq) were subjected to quality control checks before downstream analysis. Trimgalore (v.0.4.3)[43] was used to remove the Nextera adaptor sequences and low-quality bases. Reports before and after read trimming were generated using fastqc (v.0.11.2)[44] and collated using MultiQC (v.1.7)[45]. The mean read length and number of sequences provided on the MultiQC reports were used to determine sequencing coverage. Paired-end reads were assembled using the Shovill pipeline with associated dependencies. Final genome assembly metrics were generated using QUAST (v.2.1)[46]. Bacterial species were identified using BLASTn (https://blast.ncbi.nlm.nih.gov/Blast.cgi) (v.2.2.25)[47] (input: contigs) and PathogenWatch[48]. Multilocus sequence typing (MLST), antibiotic resistance and plasmid genomic profiles were characterized using ABRicate v.0.9.7 (ref. [49]) and

associated databases: NCBI[50] and PlasmidFinder[51]. The average nucleotide identity (ANI) was calculated using ChunLab's online ANI[52].

Previously undefined alleles and ST profiles were submitted to Enterobase, BIGSbd and PubMLST for assignment[53]. Genomes were annotated using Prokka (v.1.12)[54]. Isolate relatedness analysis was performed using Roary (v.3.12.0)[55] to create a core genome alignment and FastTree (v.3.12.0)[56] to generate maximum likelihood phylogenetic trees. Core phylogenetic trees were mid-rooted, visualized and annotated using iTOL (v.4)[57].

SNP analysis was performed on ST-specific clades using snippy (v.4.6.0)[58] (input paired-end fastq) with BWA and freebayes mapping the reads and calling variants. To maximize SNP calling, for each clade, a high-quality reference was generated[59] using long reads (ONT bioinformatics, see below; Supplementary Table 5 summarizes the genome metrics for each SNP reference genome). Snippy-core was used to concatenate SNPs and snp-sites[60] was used to extract SNPs. Gubbins (v.2.3.4)[61] was used to identify and remove recombination. IQ-tree (v.2.0) was used to generate a maximum likelihood SNP tree[62]. Snp-dists was used to generate a pairwise SNP matrix[63]. SNP trees were outgroup rooted where possible, or mid-point rooted and visualized with iTOL (v.4)[57].

ONT FAST5 reads were base called using Guppy v.5.0.11 and NVIDIA v.100 GPUs. Filtlong (v.0.2.0) was used to trim fastq (--min_length 1000 --keep_percent 90) and the reads were assembled with the corresponding short reads using Unicycler v.0.4.9 with default parameters. The number and length ($N50$) of long reads was determined using Nanoplot (v.1.19.0)[64]. Plasmid sequences were extracted using Bandage v.0.8.11 (ref. [65]) and assessed for similarity using PLSDB[66] and BLAST[47]. Plasmid analysis was performed for $n = 50$ isolates chosen based on short-read sequencing analysis (ARG carriage, ST and core genome phylogeny) and other metadata available (clinical site, sample type, date).

Illumina paired-end sequence reads were submitted to the European Nucleotide Archive (ENA) and given accession no. PRJEB39293. Hybrid genomes (Illumina and ONT) were submitted to the NCBI and given BioProject accession no. PRJNA767644.

*Global isolates for contextual analysis.* Approximately 100 isolates of *E. coli*, *K. pneumoniae*, and *E. cloacae* were included in phylogenetic analyses. Isolates were chosen from two searches. First, a literature search ascertained the availability of whole genomes from studies focusing on neonatal studies and/or rectal or intestinal carriage of ESBL/carbapenemase, primarily, but not exclusively in LMICs. Available clinical and other associated data, including country, sample, source and date, were collected, where available.

Second, to provide further context from additional countries and sources, including animal and environmental, between 50 and 80 genomes were chosen from the NCBI Assembly collection. On 5 March 2020 sequence data in fasta file format were downloaded from the NCBI's Assembly resource. For *E. coli*, 18,761 genomes were downloaded and 18,673 were further identified as *E. coli* using in-house bioinformatics analysis as described above. A total of 1,790 different STs were found within the *E. coli* collection. For *K. pneumoniae*, 8,663 genomes were downloaded and 8,660 were further identified as *K. pneumoniae* using in-house bioinformatics analysis. A total of 930 different STs were found within the *K. pneumoniae* collection. For *E. cloacae* complex, 1,960 genomes were downloaded and 1,886 were further identified as being *E. cloacae* complex, of which 398 different STs were found.

ABRicate v.0.9.7 was used to screen all genomes for ARGs. To assist choosing both $bla_{NDM/OXA-48}$-like-positive and -negative isolates, the dataset was divided according to the ARG output. Genomes were then chosen at random and the accession nos. used to obtain biosample information, including, where possible, country, sample, source and date of isolate.

**Statistical analysis.** A formal sample size was not calculated. Sites were asked to recruit all eligible mothers into the overarching study over a period of at least 12 months. All BRs were processed and MRs were processed at a ratio previously described.

Logistic regression models were fitted to maternal and living environment variables (maternal carriage) and healthcare settings, maternal and living environment variables (neonate carriage) to investigate associations with MR and BR β-lactamase gene carriage ($bla_{CTX-M-15}$, $bla_{NDM}$ and $bla_{OXA-48}$-like genes separately). These were also done for the neonate birth cohort only. Several multivariable analyses were also carried out to explore the association between these variables and maternal/infant carriage of ARGs. Explanatory variables were selected for inclusion in multivariable models on the basis of expert opinion and literature[67]. The variables included are detailed in Supplementary Methods.

We investigated the association between MR/BR β-lactamase gene carriage and neonatal BS by fitting logistic regression models.

To investigate the association between MR β-lactamase gene carriage and birth outcomes (delivery type, timing of birth, perinatal asphyxia, breech presentation and PPROM), multinomial (delivery type, SVD as the base outcome), timing of birth (on time as the base outcome), perinatal asphyxia (no as the base outcome)) and logistic regression (breech presentation and PPROM) models were fitted with birth outcomes as the outcome, and MR β-lactamase gene carriage as the explanatory variable. Conversely, to investigate the association between birth

outcomes and BR β-lactamase gene carriage, logistic regression models were fitted with BR β-lactamase gene carriage as the outcome and birth outcomes as the explanatory variable.

All models were adjusted for site as a fixed effect. For the association between β-lactamase gene carriage and neonatal BS, we also reported the associations without adjusting for site. Logistic regression models are reported as odds ratios (ORs), 95% confidence intervals (CIs) and $P$ values. Multinomial logistic regression models are reported as relative risk ratios, 95% CIs and $P$ values. $P$ values were adjusted for multiple testing using the Holm–Bonferroni method[68] on a per outcome/model basis (for example, associations with $bla_{CTX-M-15}$ in MRs were adjusted for separately to associations with $bla_{NDM}$ in MRs, and so on), with a familywise error rate (FWER) of 0.05. Furthermore, owing to the small percentage of missing data, no imputation of missing variables was performed. Given the large number of hypothesis tests reported in the present study, findings where $P_{adj} < 0.05$ are highlighted in the main text. However, findings from all analyses can be found in Table 1 and the accompanying data, and Supplementary Methods. All analyses used the $z$-test from a logistic regression model and all statistical tests were two sided. Statistical analyses were conducted using Stata v.16.1.

Extended Data Fig. 7 was edited to reflect significant values with coloured dots using Adobe Illustrator v.25.0.1.

**Reporting summary.** Further information on research design is available in the Nature Research Reporting Summary linked to this article.

## Data availability
Sequence reads have been submitted to the European Nucleotide Archive under accession no. PRJEB39293. Individual accession nos. and additional genomics data can be accessed in the Supplementary Methods and in the Source data provided with this paper. Hybrid assemblies (Illumina and ONT) have been submitted to the NCBI under the BioProject accession no. PRJNA767644. Databases used within this study: VFDB, http://www.mgc.ac.cn/VFs/download.htm; NCBI, https://github.com/tseemann/abricate/tree/master/db/ncbi; Resfinder, https://github.com/tseemann/abricate/tree/master/db/resfinder; Plasmidfinder, https://bitbucket.org/genomicepidemiology/plasmidfinder/src/master; mlst, https://github.com/tseemann/mlst/tree/master/db/pubmlst; PLSBD, https://ccb-microbe.cs.uni-saarland.de/plsdb.

## Code availability
Programs were used with default parameters unless specified. The code used is available upon request.

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

## Acknowledgements

We thank all participants and their families. This work was supported by a combination of two research awards (nos. OPP1119772 and OP1191522) from the Bill & Melinda Gates Foundation. We thank Liofilchem for their continued support in the distribution of their microbiology products to enable standardization of standard operating procedures across the clinical sites. We thank J. Parkhill for advice and guidance regarding the phylogenetic analyses. We thank Wales Gene Park and ARCCA for their continued bioinformatics support and infrastructure availability. Bioinformatics analysis was largely undertaken using the supercomputing facilities at CU operated by ARCCA on behalf of the Cardiff Supercomputing Facility and the HPC Wales and Supercomputing Wales projects. The latter is part funded by the European Regional Development Fund via the Welsh Government. We thank the team of curators for the databases hosted on PubMLST https://pubmlst.org/databases. We thank the curators of the Institut Pasteur MLST and whole-genome MLST databases for curating the *Klebsiella* spp. data and making them publicly available at http://bigsdb.pasteur.fr. We thank M. Islam for providing access to the clinical sites and epidemiology data in Bangladesh. We would like to acknowledge R. Kamran, the microbiologist from PIMS, Pakistan, who sadly passed away in 2018. We thank the team within the Specialist Antimicrobial Chemotherapy Unit, University Hospital Wales, Public Health Wales for their support for MALDI-TOF MS of bacterial isolates. We thank A. Reis (iBiMED, University of Aveior) for help in producing Fig. 2c and Extended Data Fig. 1.

## Author contributions

M.J.C. and K.S. designed, guided the study and analysis, and wrote the manuscript. K.S., E.P. and I.B. performed the WGS experiments. M.J.C., K.S., K. Thomson, E.P., J.M., C.D., C.A., P.H., H.S., A.F., M.N., T.H., S.N., A.M., A.R. and L.R. performed the microbiology experiments. M.J.C. and B.H. optimized the PCR reaction. K.S., M.J.C. and R.A. performed the bioinformatics analysis. R.M., D.G., K.H., G.C., C.D. and T.R.W. designed and delivered the epidemiological aspects of the study. J.W. performed analysis to guide sample selection and processing. D.G. performed statistical analysis. G.C., D.B., S.S., G.M., S.B., P.C., S.H., A.S., S.M., K.I., F.M., S.U., L.A., C.E., A.H.Y., A.A., A.S.M., R.Z., H.S., A.M., N.S., M.H.J., S.A., J.B.M., A.R., L.G., S.M., A.N.H.B., A.W. and L.M. assisted in collecting rectal swab samples and transporting to the United Kingdom. G.C., S.B., K.I., R.Z., J.B.M. and S.M. facilitated the epidemiology data collection at the clinical sites. T.R.W., M.J.C., R.M., G.C., S.B., K.I., R.Z., J.B.M. and S.M. designed the BARNARDS study.

## Competing interests

The authors declare no competing interests.

## Additional information

**Extended data** Extended data are available for this paper at https://doi.org/10.1038/s41564-022-01184-y.

**Correspondence and requests for materials** should be addressed to M. J. Carvalho.

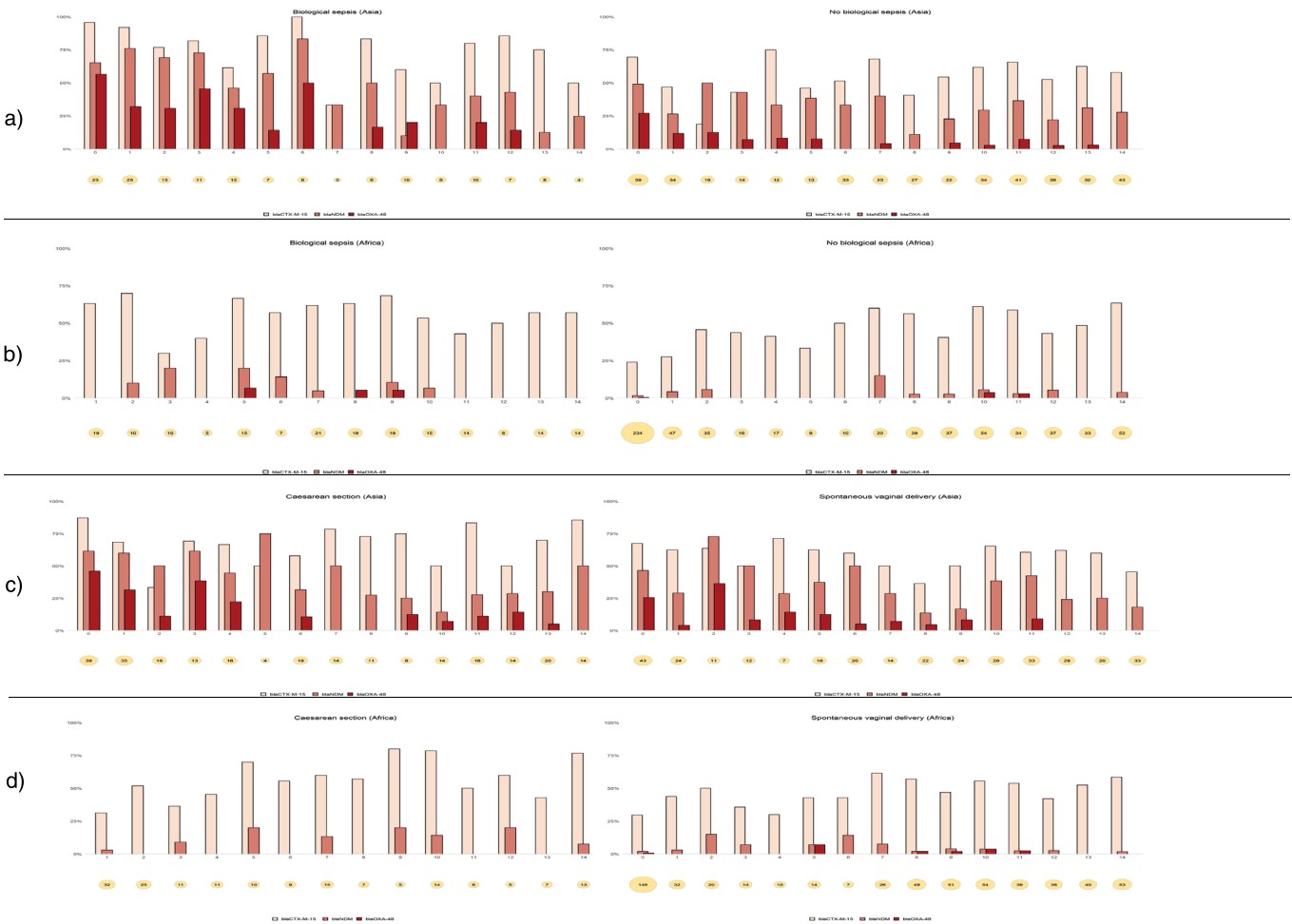

**Extended Data Fig. 1 | Carriage of *bla*$_{CTX-M-15}$, *bla*$_{NDM}$ and *bla*$_{OXA-48}$-like genes among neonates' rectal swabs by age of neonates at sample collection.**
Carriage of *bla*$_{CTX-M-15}$, *bla*$_{NDM}$ and *bla*$_{OXA-48}$-like genes among neonates' rectal swabs against age of neonates at sample collection per biological sepsis status in a) Asia and b) Africa and per delivery type in c) Asia and d) Africa. Prevalences of each antibiotic resistance gene (ARG) are plotted. The total number of samples collected per day for each type of delivery is shown in the circles below the graphs. From day 0, ARGs were detected in the neonates' faecal microbiota independently of biological sepsis status or delivery type. Plots were done in R studio using packages tidyr (v1.2.0), ggpubr (v0.4.3), gridExtra (v2.3), and egg (v0.4.3).

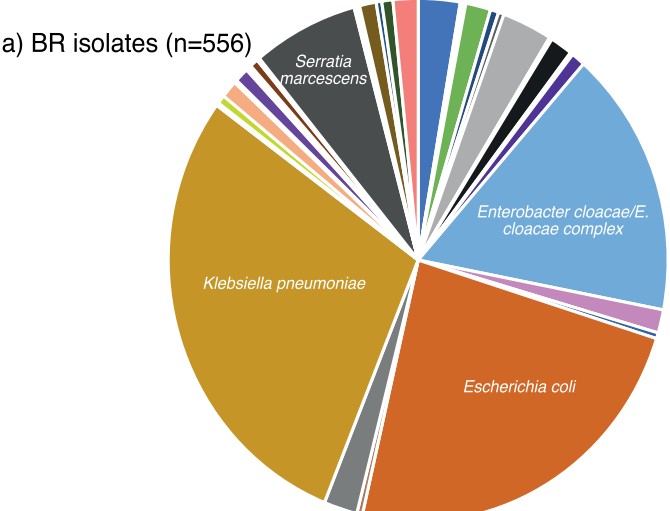

a) BR isolates (n=556)

*Serratia marcescens*

*Enterobacter cloacae/E. cloacae complex*

*Escherichia coli*

*Klebsiella pneumoniae*

- *Acinetobacter baumannii/A. baumannii cplx, n=15*
- *Acinetobacter bereziniae, n=1*
- *Acinetobacter junii, n=1*
- *Acinetobacter nosocomialis, n=9*
- *Acinetobacter pittii, n=3*
- *Aeromonas caviae, n=2*
- *Citrobacter freundii, n=18*
- *Citrobacter koseri, n=1*
- *Citrobacter sedlakii, n=8*
- *Citrobacter sp., n=1*
- *Enterobacter asburiae, n=5*
- *Enterobacter cloacae/E. cloacae cplx, n=92*
- *Enterobacter kobei, n=8*
- *Enterobacter sp., n=2*
- *Escherichia coli, n=132*
- *Klebsiella aerogenes, n=2*
- *Klebsiella oxytoca, n=12*
- *Klebsiella pneumoniae, n=161*
- *Klebsiella variicola, n=1*
- *Proteus mirabilis, n=3*
- *Pseudomonas aeruginosa, n=6*
- *Pseudomonas citronellolis, n=1*
- *Pseudomonas mendocina, n=5*
- *Pseudomonas sp., n=1*
- *Pseudomonas stutzeri, n=1*
- *Raoultella ornithinolytica, n=3*
- *Raoultella sp., n=1*
- *Serratia marcescens, n=38*
- *Serratia sp., n=1*
- *Serratia ureilytica, n=1*
- *Shewanella putrefaciens, n=6*
- *Sphingomonas paucimobilis, n=2*
- *Stenotrophomonas maltophilia, n=4*
- *Unidentified, n=9*

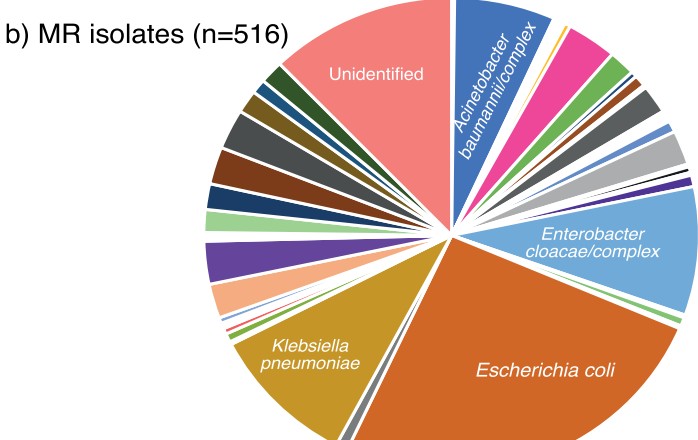

b) MR isolates (n=516)

Unidentified

*Acinetobacter baumannii/complex*

*Enterobacter cloacae/complex*

*Escherichia coli*

*Klebsiella pneumoniae*

- *Achromobacter xylosoxidans, n=1*
- *Acinetobacter baumannii/A. baumannii cplx, n=34*
- *Acinetobacter baylyi, n=1*
- *Acinetobacter bereziniae, n=1*
- *Acinetobacter haemolyticus, n=1*
- *Acinetobacter johnsonnii, n=1*
- *Acinetobacter junii, n=2*
- *Acinetobacter lwoffii, n=17*
- *Acinetobacter nosocomialis, n=9*
- *Acinetobacter pittii, n=2*
- *Acinetobacter sp., n=4*
- *Acinetobacter ursingii, n=1*
- *Aeromonas caviae, n=10*
- *Aeromonas eucrenophila, n=1*
- *Aeromonas salmonicida, n=1*
- *Alcaligenes faecalis, n=1*
- *Bordetella hinzii, n=1*
- *Brevundimonas sp., n=4*
- *Citrobacter freundii, n=12*
- *Citrobacter koseri, n=1*
- *Citrobacter sedlakii, n=2*
- *Delftia acidovorans, n=1*
- *Enterobacter asburiae, n=4*
- *Enterobacter cloacae/E. cloacae cplx, n=45*
- *Enterobacter kobei, n=1*
- *Enterobacter ludwigii, n=3*
- *Enterobacter sp., n=1*
- *Escherichia coli, n=132*
- *Klebsiella oxytoca, n=4*
- *Klebsiella pneumoniae, n=50*
- *Klebsiella sp., n=1*
- *Kluyvera cryocrescens, n=3*
- *Leclercia adecarboxylata, n=2*
- *Moraxella sp., n=1*
- *Pantoea sp., n=1*
- *Providencia rettgeri, n=2*
- *Pseudomonas aeruginosa, n=12*
- *Pseudomonas mendocina, n=15*
- *Pseudomonas monteilii, n=1*
- *Pseudomonas oleovorans, n=1*
- *Pseudomonas plecoglossicida, n=1*
- *Pseudomonas putida, n=8*
- *Pseudomonas stutzeri, n=9*
- *Raoultella ornithinolytica, n=13*
- *Serratia marcescens, n=14*
- *Shewanella putrefaciens, n=8*
- *Sphingomonas paucimobilis, n=5*
- *Stenotrophomonas maltophilia, n=8*
- *Unidentified, n=63*

**Extended Data Fig. 2 | Species diversity of the isolates recovered from rectal samples.** Diversity of carbapenemase positive Gram-negative bacterial isolates collected from a) neonates' and b) mothers' rectal swabs. The number of isolates collected per species is shown.

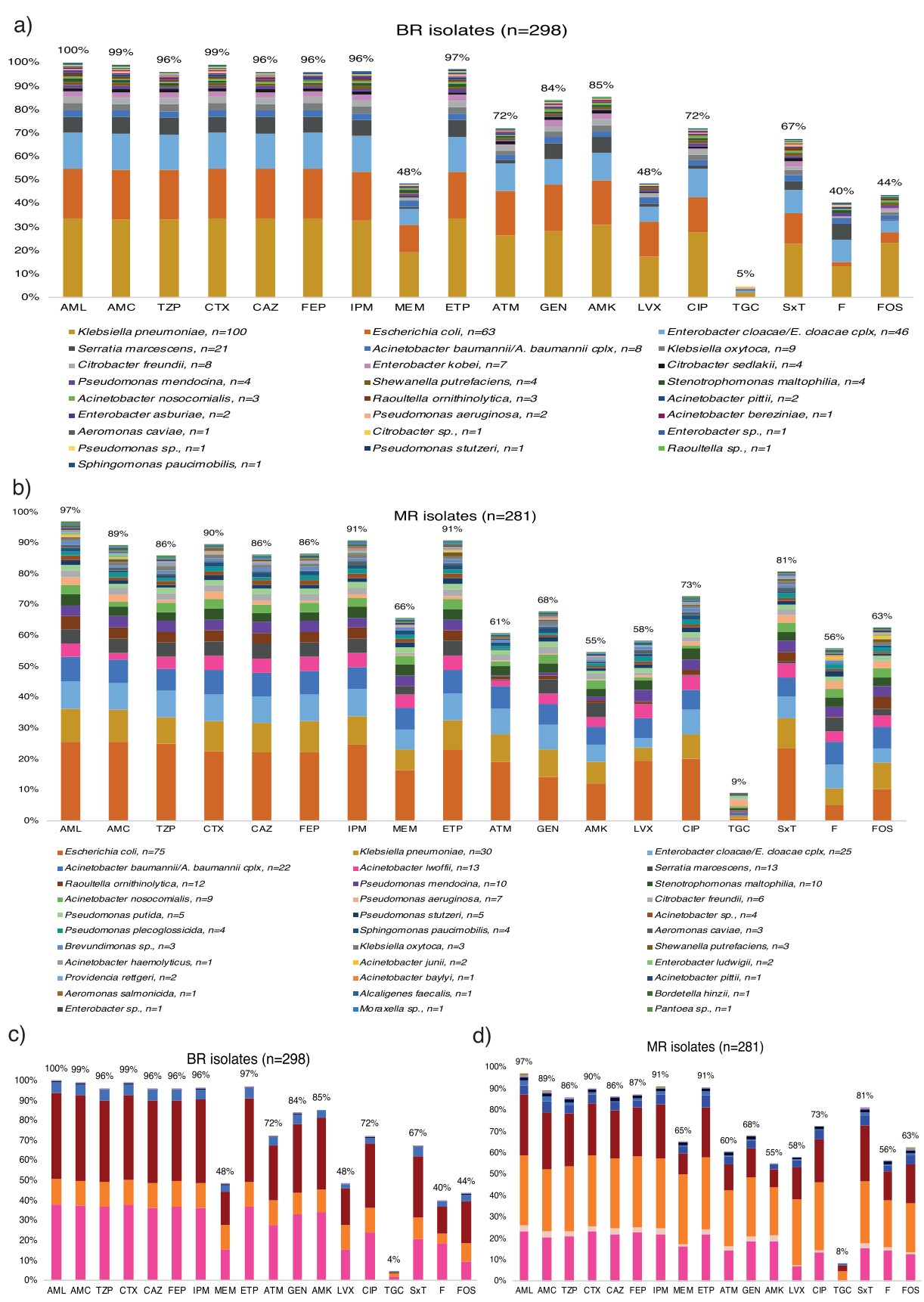

Extended Data Fig. 3 | See next page for caption.

**Extended Data Fig. 3 | Antimicrobial resistance profiles of the isolates recovered from rectal samples.** Antimicrobial resistance (AMR) profiles of the carbapenemase positive Gram-negative bacterial isolates collected from a) neonates' and b) mothers' rectal swabs per species. AMR profiles distributed by site for c) neonates' and d) mothers' isolates distributed by site are also shown. The number of isolates tested for antibiotic susceptibility per species (a, b) and per site (c, d) is shown in the legends. The overall percentage of resistant isolates to each antimicrobial tested is shown at the top of the corresponding bar. For antibiotics' acronyms see methods.

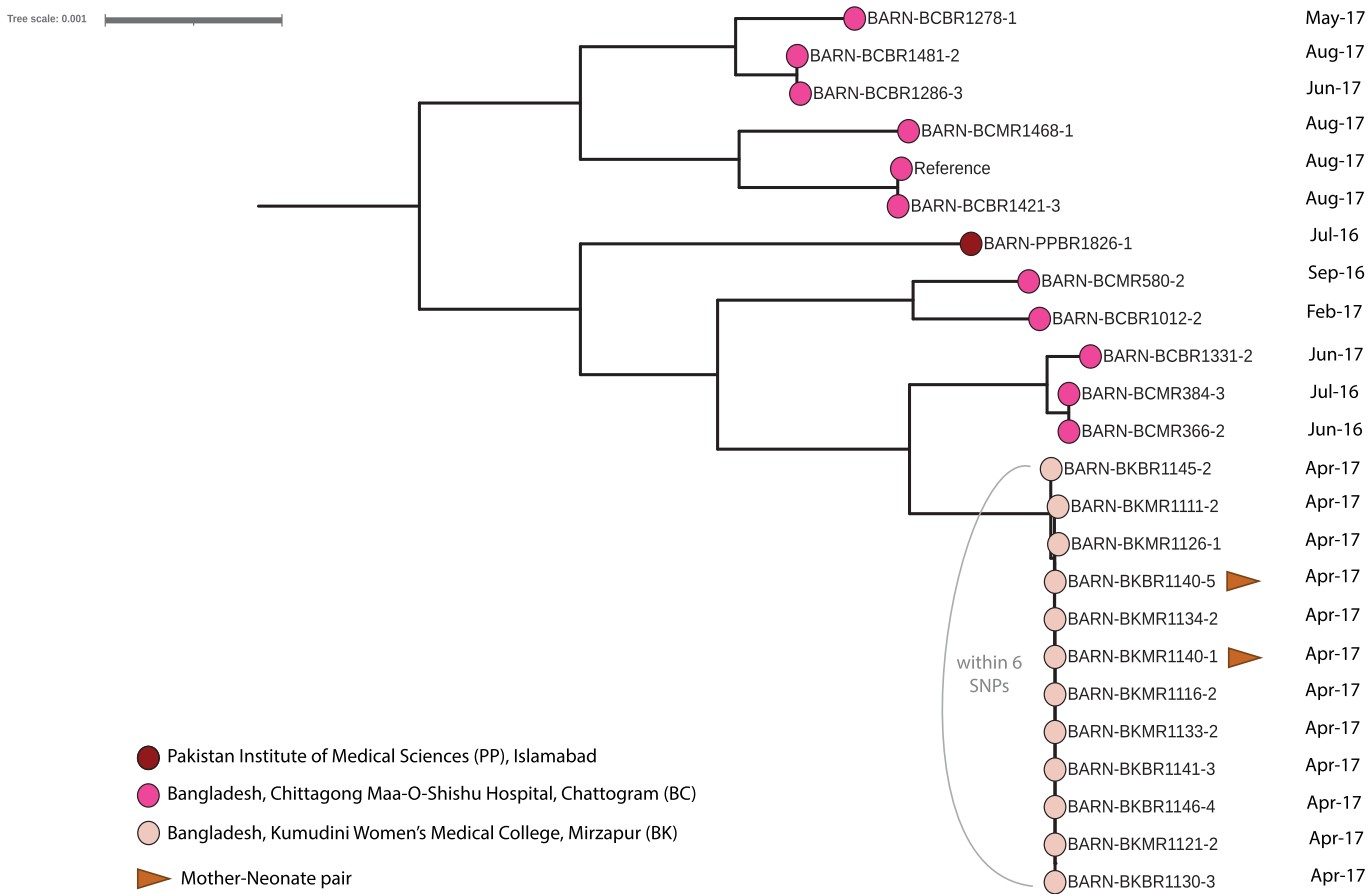

**Extended Data Fig. 4 | Whole genome single-nucleotide polymorphisms analysis of ST405 *E. coli* isolates.** Whole genome SNP analysis of ST405 *E. coli* (n=24 isolates) using snippy v4.6.0 on paired end fastq, gubbins v2.3.4 to remove recombination and IQ-tree v2.0 to construct the phylogeny. iTOL v4 was used to visualise the phylogenetic tree. BCBR/BCMR, neonate (BR)/mother (MR) rectal isolates from Bangladesh, Chittagong Maa-O-Shishu Hospital, Chattogram (BC). BKBR/BKMR, neonate (BR)/mother (MR) rectal isolates from Bangladesh, Kumudini Women's Medical College, Mirzapur (BK). PPBR, neonate (BR) rectal isolates from Pakistan Institute of Medical Sciences (PP), Islamabad. Reference, BC-MR1421-3. Dates of rectal swab collection are shown.

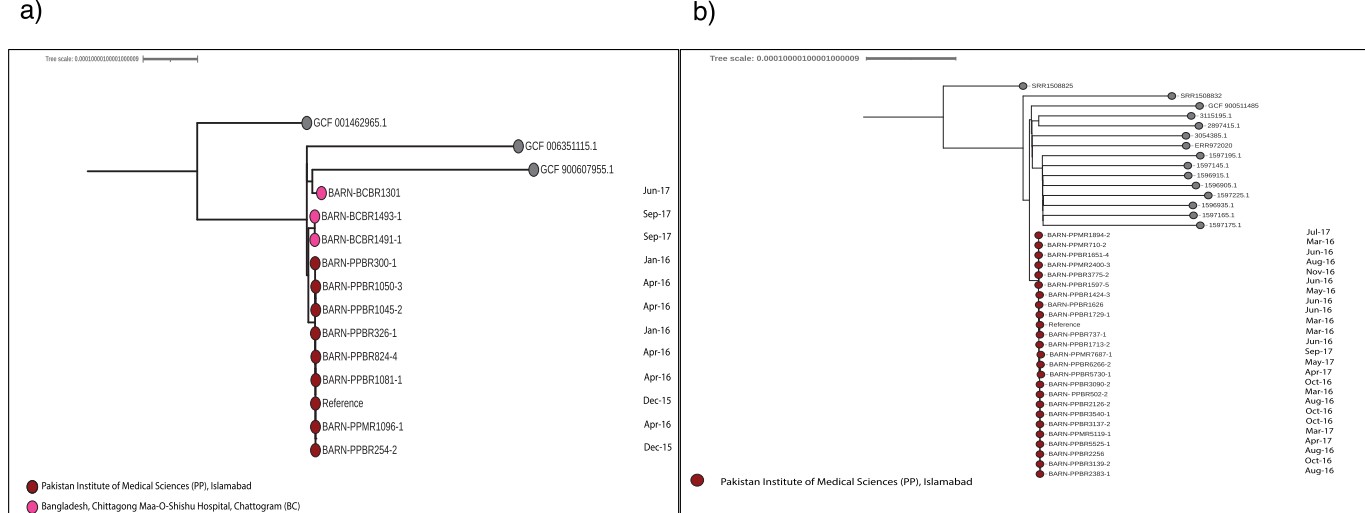

**Extended Data Fig. 5 | Whole genome single-nucleotide polymorphisms analysis of ST11 and ST15 *K. pneumoniae* isolates.** a) SNP analysis of ST11 *Klebsiella pneumoniae* (n = 12 isolates) collected during this study and *K. pneumoniae* GCF_001462965, GCF_006351115 and GCF_900607955 (see SourceData for *Enterobacter* sp. trees) using snippy v4.6.0 on paired end fastq, gubbins v2.3.4 to remove recombination and IQ-tree v2.0 to construct the phylogeny. iTOL v4 was used to visualise the phylogenetic tree. BCBR/BCMR, neonate (BR)/mother (MR) rectal isolates from Bangladesh, Chittagong Maa-O-Shishu Hospital, Chattogram (BC). BKBR/BKMR, neonate (BR)/mother (MR) rectal isolates from Bangladesh, Kumudini Women's Medical College, Mirzapur (BK). PPBR/PPMR, neonate (BR)/mother (MR) rectal isolates from Pakistan Institute of Medical Sciences (PP), Islamabad. Reference, PP-BR254-2. Dates of rectal swab collection are shown. b) SNP analysis of ST15 *K. pneumoniae* (n = 25 isolates; BARN) collected during this study and other *K. pneumoniae* isolates' genome assemblies available in GeneBank (see SourceData for *Klebsiella pneumoniae* trees) using snippy v4.6.0 on paired end fastq, gubbins v2.3.4 to remove recombination and IQ-tree v2.0 to construct the phylogeny. iTOL v4 was used to visualise the phylogenetic tree. PPBR/PPMR, neonate (BR)/mother (MR) rectal isolates from Pakistan Institute of Medical Sciences (PP), Islamabad. Reference, PP-BR737-1. Dates of rectal swab collection are shown.

Tree scale: 0.001

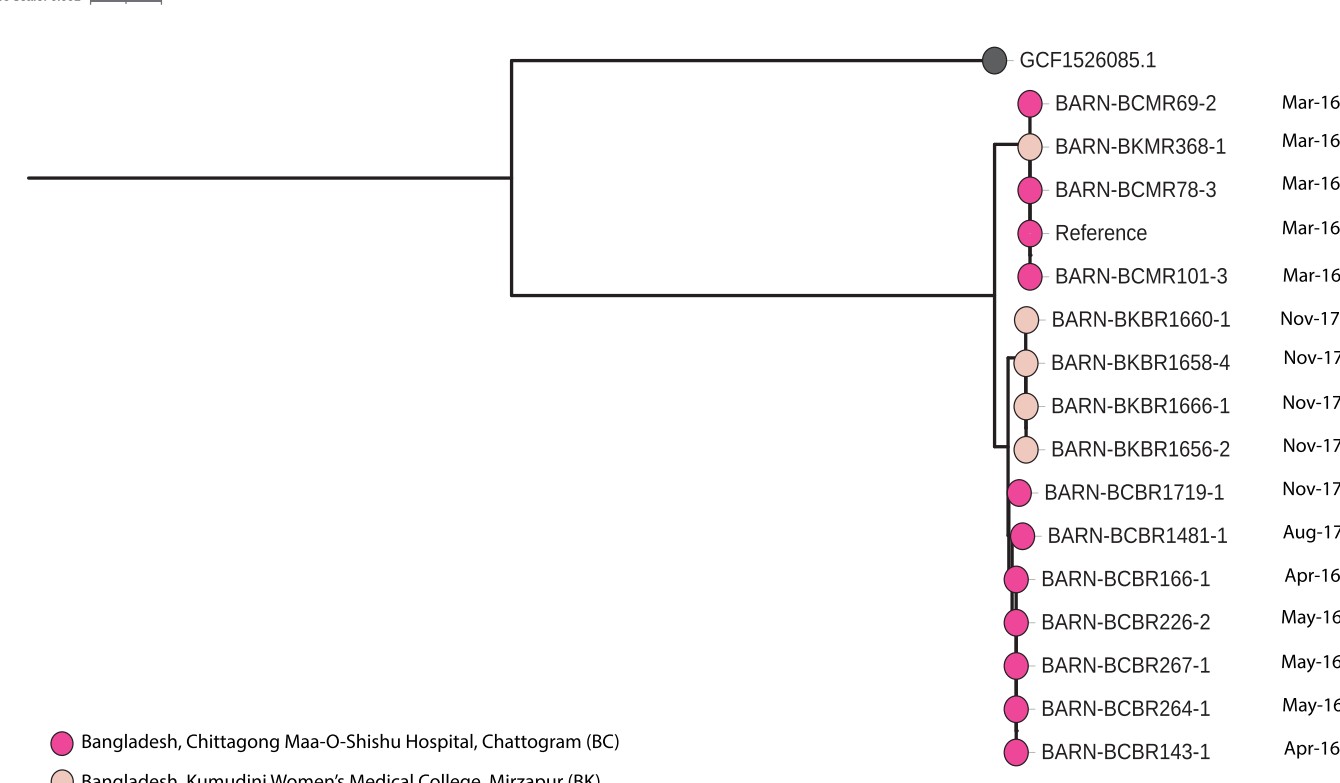

**Extended Data Fig. 6 | Whole genome single-nucleotide polymorphisms analysis of ST418 *E. hormaechei* isolates.** SNP analysis of ST418 *E. hormaechei* (n = 16) collected during this study and *Enterobacter hormaechei* subsp. *xiangfangensis* GCF_1526085 using snippy v4.6.0 on paired end fastq, gubbins v2.3.4 to remove recombination and IQ-tree v2.0 to construct the phylogeny. iTOL v4 was used to visualise the phylogenetic tree. BCBR/BCMR, neonate (BR)/mother (MR) rectal isolates from Bangladesh, Chittagong Maa-O-Shishu Hospital, Chattogram (BC). BKBR/BKMR, neonate (BR)/mother (MR) rectal isolates from Bangladesh, Kumudini Women's Medical College, Mirzapur (BK). Reference, BC-MR78-3. Dates of rectal swab collection are shown.

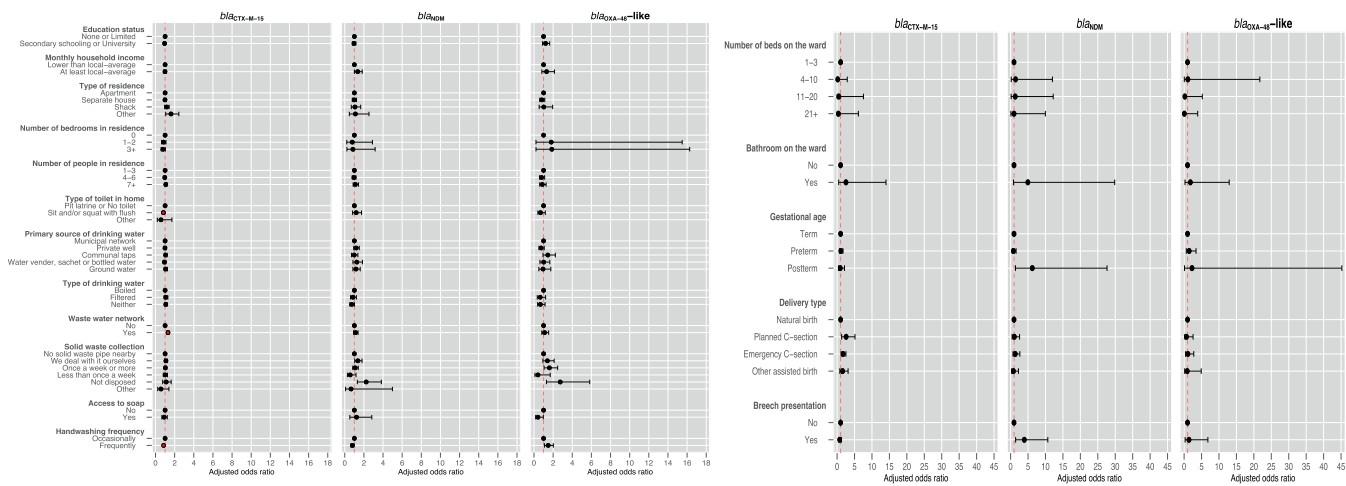

**Extended Data Fig. 7 | Exploratory multivariable statistical analysis to identify associations between socio-demographic and clinical data and maternal and neonatal carriage of ARGs.** Forest plots representing exploratory multivariable statistical analysis to identify associations between socio-demographic and clinical data and maternal and neonatal carriage of ARGs. Bars represent ranges of odds ratio. Multiplicity-adjusted p-values that remained statistically significant at the 5% level are coloured in red. Z tests were used from multivariable logistic regression models and statistical tests were two-sided. a) Two multivariable models (MV) were performed to understand the association between WASH (water, sanitation and hygiene) related variables and maternal carriage of ARGs using the explanatory variables shown. MV2 results are displayed; In MV1, type of toilet in home did not gave an association with carriage of any of the ARGs. b) Two MV were performed to understand the association between birth healthcare environment features and carriage of ARGs among neonates from the birth cohort using the explanatory variables shown. MV1 results are shown. Perinatal asphyxia is not plotted given the wide confidence intervals.

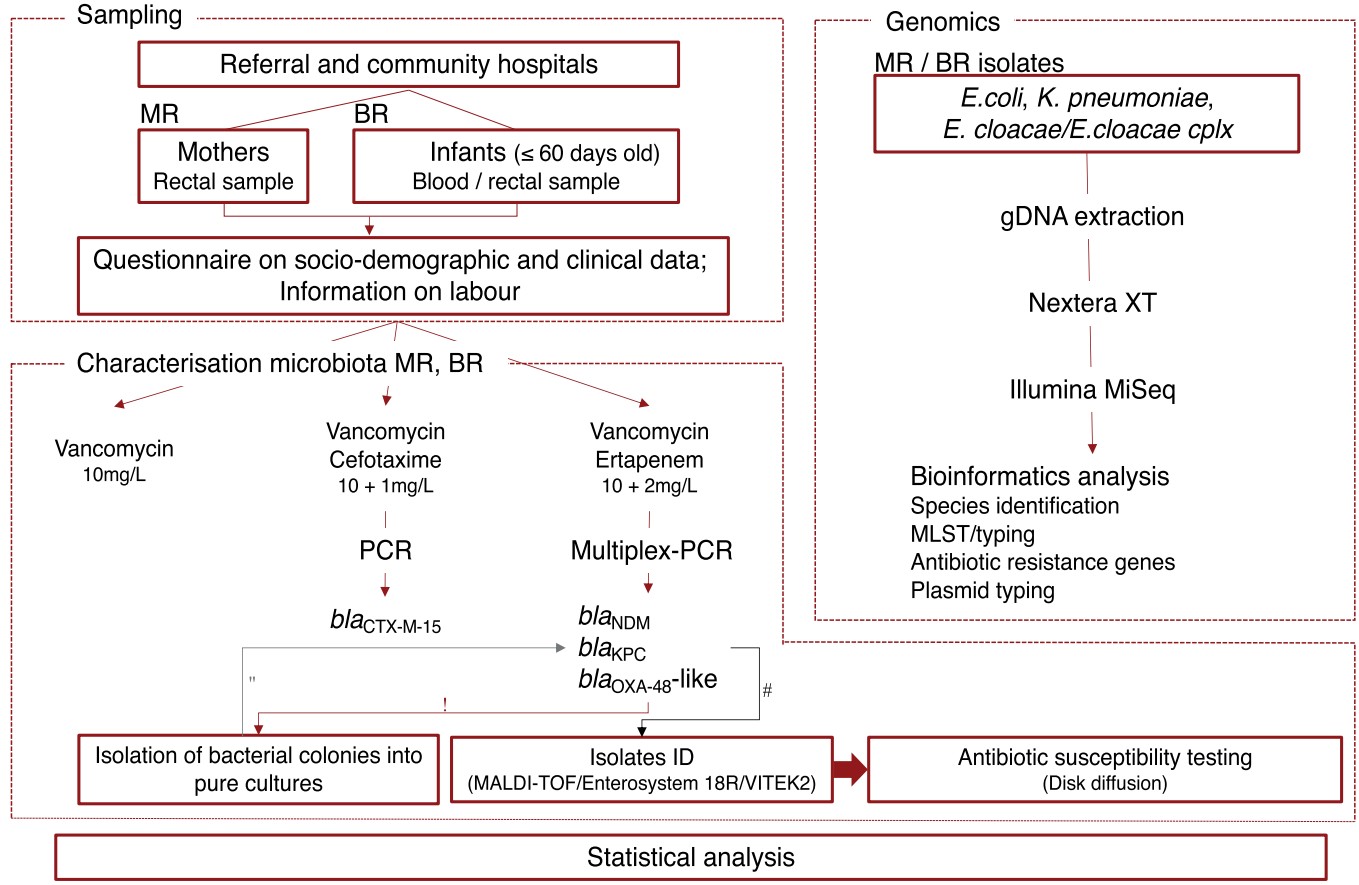

**Extended Data Fig. 8 | BARNARDS workflow for collection of samples and data in local sites and samples and isolates characterisation in Cardiff University.** Indian samples characterisation was performed locally.

# Reporting Summary

Nature Research wishes to improve the reproducibility of the work that we publish. This form provides structure for consistency and transparency in reporting. For further information on Nature Research policies, see our Editorial Policies and the Editorial Policy Checklist.

## Statistics

For all statistical analyses, confirm that the following items are present in the figure legend, table legend, main text, or Methods section.

| n/a | Confirmed | |
|---|---|---|
| ☐ | ☒ | The exact sample size (*n*) for each experimental group/condition, given as a discrete number and unit of measurement |
| ☐ | ☒ | A statement on whether measurements were taken from distinct samples or whether the same sample was measured repeatedly |
| ☐ | ☒ | The statistical test(s) used AND whether they are one- or two-sided<br>*Only common tests should be described solely by name; describe more complex techniques in the Methods section.* |
| ☐ | ☒ | A description of all covariates tested |
| ☐ | ☒ | A description of any assumptions or corrections, such as tests of normality and adjustment for multiple comparisons |
| ☐ | ☒ | A full description of the statistical parameters including central tendency (e.g. means) or other basic estimates (e.g. regression coefficient) AND variation (e.g. standard deviation) or associated estimates of uncertainty (e.g. confidence intervals) |
| ☐ | ☒ | For null hypothesis testing, the test statistic (e.g. *F*, *t*, *r*) with confidence intervals, effect sizes, degrees of freedom and *P* value noted<br>*Give P values as exact values whenever suitable.* |
| ☒ | ☐ | For Bayesian analysis, information on the choice of priors and Markov chain Monte Carlo settings |
| ☒ | ☐ | For hierarchical and complex designs, identification of the appropriate level for tests and full reporting of outcomes |
| ☒ | ☐ | Estimates of effect sizes (e.g. Cohen's *d*, Pearson's *r*), indicating how they were calculated |

*Our web collection on statistics for biologists contains articles on many of the points above.*

## Software and code

Policy information about availability of computer code

| | |
|---|---|
| Data collection | No software was used in the data collection. At the low middle income countries, research nurses completed questionnaires with the women approaching labor. These questionnaires were either transcribed onto paper, due to availability of resources/infrastructure, i.e. Internet access, and later uploaded into Bristol Online survey (BOS) or directly entered into BOS using a tablet device provided by the project. |
| Data analysis | CLIMB (v1.0)<br>Trimgalore (v0.4.3)<br>fastqc (v0.11.2)<br>MultiQC (v1.7)<br>Shovill and associated dependencies (v0.9.0)<br>quast (v.2.1)<br>Blast nt (https://blast.ncbi.nlm.nih.gov/Blast.cgi) (v2.2.25)<br>PathogenWatch (v.3.13.10; https://pathogen.watch)<br>ABRicate (v0.9.7) and associated pipelines: NCBI and PlasmidFinder<br>Enterobase<br>BIGSbd (v1.25.1)<br>PubMLST<br>Prokka (v1.12)<br>Roary (v3.12.0)<br>FastTree (v3.12.0)<br>iTOL (v4)<br>snippy (v4.6.0)<br>Gubbins (v2.3.4)<br>IQ-tree (v2.0)<br>Guppy (v5.0.11) |

Filtlong (v0.2.0)
Unicycler (v0.4.9)
Nanoplot (v1.19.0)
Bandage (v0.8.1)
PLSBD
ChunLab's online ANI calculator
Stata (v16.1)
R studio using packages tidyr (v1.2.0), ggpubr (v0.4.3), gridExtra (v2.3), and egg (v0.4.3)

For manuscripts utilizing custom algorithms or software that are central to the research but not yet described in published literature, software must be made available to editors and reviewers. We strongly encourage code deposition in a community repository (e.g. GitHub). See the Nature Research guidelines for submitting code & software for further information.

## Data

Policy information about availability of data

All manuscripts must include a data availability statement. This statement should provide the following information, where applicable:
- Accession codes, unique identifiers, or web links for publicly available datasets
- A list of figures that have associated raw data
- A description of any restrictions on data availability

Sequences reads have been submitted to the European Nucleotide Archive (ENA) under the project number PRJEB39293. Individual accession numbers and additional genomics data can be accessed in the Supplementary Material Methods and Source Data for Fig. 3-5. Hybrid assemblies (Illumina and ONT) have been submitted to NCBI under the BioProject number PRJNA767644.

Databases used within this study:
VFDB: http://www.mgc.ac.cn/VFs/download.htm
NCBI: https://github.com/tseemann/abricate/tree/master/db/ncbi
Resfinder: https://github.com/tseemann/abricate/tree/master/db/resfinder
Plasmidfinder: https://bitbucket.org/genomicepidemiology/plasmidfinder/src/master
mlst: https://github.com/tseemann/mlst/tree/master/db/pubmlst
PLSBD: https://ccb-microbe.cs.uni-saarland.de/plsdb/

# Field-specific reporting

Please select the one below that is the best fit for your research. If you are not sure, read the appropriate sections before making your selection.

☒ Life sciences ☐ Behavioural & social sciences ☐ Ecological, evolutionary & environmental sciences

For a reference copy of the document with all sections, see nature.com/documents/nr-reporting-summary-flat.pdf

# Life sciences study design

All studies must disclose on these points even when the disclosure is negative.

| | |
|---|---|
| Sample size | The sampling method was purposive and a formal sample size calculation was not conducted. Based on previous studies led by PI Professor Timothy Walsh (unpublished studies/awaiting publication), BARNARDS anticipated the enrollment level between 500-2000 neonates per clinical site for the duration of the study (depending on geographical location i.e. smaller rural site would have a smaller catchment area). |
| Data exclusions | The following exclusion criteria was pre-defined: the sepsis case infant/mother sampling pair was excluded in the case of a still born. Following this, data was retrospectively excluded based on the following criteria: <br> - Incomplete questionnaire; missing multiple data points in the epidemiological dataset <br> - Mother asked for infant withdrawal <br> - Error/substantial inconsistencies in the questionnaire - laboratory sampling match up |
| Replication | No replicas were used in this study, as one sample was taken per enrolled subject. |
| Randomization | All women approaching labour and their neonates and women who recently were in labour and whose neonates (≤60 days old) were clinically diagnosed with sepsis, were enrolled onto the study following consent. This was an observational study with no experimental and control groups, hence randomization was not relevant to the study. |
| Blinding | Blinding was not relevant for the study as this was an observational study with no randomization used. In each site, all samples were sequentially coded. |

# Reporting for specific materials, systems and methods

We require information from authors about some types of materials, experimental systems and methods used in many studies. Here, indicate whether each material, system or method listed is relevant to your study. If you are not sure if a list item applies to your research, read the appropriate section before selecting a response.

## Materials & experimental systems

| n/a | Involved in the study |
|---|---|
| ☒ ☐ | Antibodies |
| ☒ ☐ | Eukaryotic cell lines |
| ☒ ☐ | Palaeontology and archaeology |
| ☒ ☐ | Animals and other organisms |
| ☐ ☒ | Human research participants |
| ☒ ☐ | Clinical data |
| ☒ ☐ | Dual use research of concern |

## Methods

| n/a | Involved in the study |
|---|---|
| ☒ ☐ | ChIP-seq |
| ☒ ☐ | Flow cytometry |
| ☒ ☐ | MRI-based neuroimaging |

# Human research participants

Policy information about studies involving human research participants

**Population characteristics**

BARNARDS was a multi-site international prospective observational study including two recruitment pathways:
i.) Birth-Cohort: All mothers in labour admitted to clinical-sites were recruited prospectively and their infant(s) followed up until 60-days old or death.
ii.) Infant Admissions (IA): Infant(s) admitted to clinical-sites showing signs of suspected sepsis in the first 60-days of life until 60- days old or death.

For this study, isolates recovered from mothers' and neonates' rectal samples were included irrespective of cohort pathway. General population characteristics of the mothers' (outside of the scope of this manuscript): <10% previously had stillbirth, approx. 25% were first time mothers', 75% were aged between 21-35 years old.
Infants' presenting with sepsis were followed up for 60 days of life. Around 46% of neonates were 14 days old or less. Onset of sepsis was recorded, early onset (EOS) <72h, and late onset (LOS) >72h. Other population characteristics can be found in: Milton, R. et al. Neonatal sepsis and mortality in low-income and middle-income countries from a facility-based birth cohort: an international multisite prospective observational study. The Lancet Global Health 10, e661–e672 (2022).
Enrolled participants were not genotyped.

**Recruitment**

BARNARDS recruited from 12 clinical sites from Rwanda, Bangladesh, Ethiopia, Nigeria, Pakistan, India and South Africa. Where possible, large public hospitals were chosen. Recruitment took place between Nov 12, 2015, and Feb 1, 2018.
All women approaching labour and their neonates, and women who recently were in labour and whose neonates (≤60 days old) were clinically diagnosed with sepsis, were enrolled onto the study following consent. Consent was collected by trained research staff and using local languages. Neonates were then enrolled into the study.
Additionally, neonates not born within the clinical sites that were admitted with clinical signs of sepsis were also enrolled into the study following consent from the mother. The corresponding mothers were also enrolled into the study for the collection of samples and demographic data.
Neonatal follow-up was carried out at day 3, 7, 14, 28, and 60 by research nurses either face-to-face or by telephone.
Neonates remained in the study until 60 days old, withdrawal, or death.
This study incorporated two recruitment pathways to include both neonates born within the clinical sites, and also neonates in the larger catchment areas presenting to the hospital with signs of sepsis.
Women were approached and recruitment was totally dependent on their consent, so no selection bias was expected.

**Ethics oversight**

Site committees Named PI Reference(s) Approval date(s)
BC - Ethical Review Committee, Bangladesh Institute of Child Health Samir Kumar Saha BICH-ERC-4/3/2015 15/09/2015
BK - Ethical Review Committee, Bangladesh Institute of Child Health Samir Kumar Saha BICH-ERC-4/3/2015 15/09/2015
ES - Boston Children's Hospital Grace Chan IRB-P00023058 11/08/2016
IN - Institutional Ethics Committee, National Institute of Cholera and Enteric Diseases and Institue of Post Graduate Medical Education and Research, IPGME&R Research Oversight Committee Sulagna Basu A-I/2016-IEC and Inst/IEC/2016/508 17/11/2016 and 04/11/2016
NK - Kano State Hospitals Management Board Kenneth Iregbu 8/10/1437AH 13/07/2016
NN - Health Research Ethics Committee (HREC), National Hospital, Abuja Kenneth Iregbu NHA/EC/017/2015 27/04/2015 NW - Health Research Ethics Committee (HREC), National Hospital, Abuja Kenneth Iregbu NHA/EC/017/2015 27/04/2015 PC - Shaheed Zulfiqar Ali Bhutto Medical University, Pakistan Institute of Medical Sciences (PIMS) Islamabad Rabaab Zahra NA, signed letter from Prof. Tabish Hazir 27/05/2015
PP - Shaheed Zulfiqar Ali Bhutto Medical University, Pakistan Institute of Medical Sciences (PIMS) Islamabad Rabaab Zahra NA, signed letter from Prof. Tabish Hazir 27/05/2015
RK - Republic of Rwanda, National Ethics Committee Jean-Baptiste Mazarati No342/RNEC/2015 10/11/2015
RU - Republic of Rwanda, National Ethics Committee Jean-Baptiste Mazarati No342/RNEC/2015 10/11/2015
ZAT - Stellenbosch University and Tygerberg Hospital, Research projects, Western Cape Government Shaheen Mehtar N15/07/063 04/12/2015 and 02/02/2016

Ethical approval was obtained at each of the seven participating countries
Bangladesh: Ethical Review Committee, Bangladesh Institute of Child Health (BICH-ERC-4/3/2015), Ethiopia: Boston Children's Hospital (IRB-P00023058), India: Institutional Ethics Committee, National Institute of Cholera and Enteric Diseases and Institute of Post Graduate Medical Education and Research, IPGME&R Research Oversight Committee (A-I/2016-IEC and Inst/IEC/2016/508), Nigeria: Kano State Hospitals Management Board (8/10/1437AH), Health Research Ethics Committee (HREC) and National Hospital, Abuja (NHA/EC/017/2015), Pakistan: Shaheed Zulfiqar Ali Bhutto Medical University, Pakistan Institute of Medical Sciences (PIMS) Islamabad (Ref No NA, signed letter from Prof. Tabish Hazir). Rwanda: Republic of Rwanda, National Ethics Committee (No342/RNEC/2015), South Africa: Stellenbosch University and Tygerberg Hospital, Research

projects, Western Cape Government (N15/07/063). All approval dates are listed in the Supplementary Material Table 2. In local languages, research nurses provided mothers with study information and collected consent for mother and/or neonatal enrolment. Informed consent was obtained in writing unless this was not possible (due to literacy barriers), and oral consent was collected from the mothers by trained researchers. Oral consent was documented by the participant signing/marking the consent form.

Note that full information on the approval of the study protocol must also be provided in the manuscript.

