## [Peer Review File · Nature Microbiology]

Peer Review Information

Journal: Nature Microbiology

Manuscript Title: Antibiotic resistance genes in the gut microbiota of mothers and linked neonates with or without sepsis from low- and middle-income countries

Corresponding author name(s): Maria Carvalho

Reviewer Comments & Decisions:

Decision Letter, initial version:

Dear Dr. Carvalho,

Thank you for your patience while your manuscript "Rapid development of resistome in neonates in low- and middle-income countries: prevalence, risk factors and genomic analysis." was under peer-review at Nature Microbiology. It has now been seen by 4 referees, whose expertise and comments you will find at the end of this email. Although they find your work of some potential interest, they have raised a number of concerns that will need to be addressed before we can consider publication of the work in Nature Microbiology.

In particular, you will see that Referee #1 raises several concerns including the need for more detailed statistical analysis, improvement of data visualization, and overinterpretation of some of the conclusions. Referee #2 asks for more details on the logistic regression model and to improve the explanation of the samples included in the study. Referee #3 asks to improve the overall presentation of the data. Referee #4 points at the need to make clear what is the advance of your work compared to previous papers associated to the same dataset, and asks to remove the section linking ARGs to plasmids (as it is not performed with long-read sequencing; also pointed by referee #1).

Should further experimental data allow you to address these criticisms, particularly improving the statistical aspect of the work by describing the statistical plan you used and how you implemented it, we would be happy to look at a revised manuscript.

Please include a data availability statement as a separate section after Methods but before references, under the heading "Data Availability". This section should inform readers about the availability of the

2nature portfolio

data used to support the conclusions of your study. This information includes accession codes to public repositories (data banks for protein, DNA or RNA sequences, microarray, proteomics data etc...), references to source data published alongside the paper, unique identifiers such as URLs to data repository entries, or data set DOIs, and any other statement about data availability. At a minimum, you should include the following statement: "The data that support the findings of this study are available from the corresponding author upon request", mentioning any restrictions on availability. If DOIs are provided, we also strongly encourage including these in the Reference list (authors, title, publisher (repository name), identifier, year). For more guidance on how to write this section please see:

<http://www.nature.com/authors/policies/data/data-availability-statements-data-citations.pdf>

* If you have not done so already we suggest that you begin to revise your manuscript so that it conforms to our Article format instructions at <http://www.nature.com/nmicrobiol/info/final-submission>. Refer also to any guidelines provided in this letter.

When submitting the revised version of your manuscript, please pay close attention to our [href="https://www.nature.com/nature-research/editorial-policies/image-integrity">Digital Image Integrity Guidelines.](https://www.nature.com/nature-research/editorial-policies/image-integrity) and to the following points below:

{redacted}

2nature portfolio

Note: This url links to your confidential homepage and associated information about manuscripts you may have submitted or be reviewing for us. If you wish to forward this e-mail to co-authors, please delete this link to your homepage first.

Nature Microbiology is committed to improving transparency in authorship. As part of our efforts in this direction, we are now requesting that all authors identified as 'corresponding author' on published papers create and link their Open Researcher and Contributor Identifier (ORCID) with their account on the Manuscript Tracking System (MTS), prior to acceptance. This applies to primary research papers only. ORCID helps the scientific community achieve unambiguous attribution of all scholarly contributions. You can create and link your ORCID from the home page of the MTS by clicking on 'Modify my Springer Nature account'. For more information please visit www.springernature.com/orcid.

If you wish to submit a suitably revised manuscript we would hope to receive it within 6 months. If you cannot send it within this time, please let us know. We will be happy to consider your revision, even if a similar study has been accepted for publication at Nature Microbiology or published elsewhere (up to a maximum of 6 months).

Yours sincerely,

{redacted}

Reviewer Comments:

Reviewer #1 (Remarks to the Author):

In this work, Carvalho & Sands et al. characterize the distribution of clinically important extended spectrum β -lactamases (ESBLs) and carbapenemases in Gram-negative bacterial isolates selectively cultured from rectal swabs of mothers and their neonates. Authors leverage the BARNARDS network of 12 clinical sites across 7 low- and middle-income countries (LMICs) to collect 2,917 and 15,217 rectal swabs from neonates suspected of sepsis, and their mothers, respectively. Authors culture swabs on antibiotic-infused agar plates and perform multiplex-PCR of four ESBLs and carbapenemases on DNA of cultured isolates to show prevalence of beta-lactamase genes among rectal microbiota of mothers and neonates with suspected sepsis in LMICs. Using MALDI-TOF MS they determine the identity of most isolates grown on the ESBL-selective agar, and sequence the isolates they are able to resuscitate from the predominating taxa (*Enterobacter cloacae*, *Escherichia coli*, *Klebsiella pneumoniae*). The authors use computational tools to show relatedness of isolates as well as co-occurrence of globally impactful ARGs plasmid replicons. Lastly, they integrate their molecular dataset with extensive clinical and behavioral metadata to associate the spread of extended-spectrum resistance genes with poor sanitation, personal hygiene, clinical history and socioeconomic variables. Overall, the data presented by the authors complement an extensive body of literature characterizing the high abundance of clinically important resistance genes in LMICs with a new multinational study in

3nature portfolio

the context of suspected neonatal sepsis. In this work, authors describe elevated prevalence of certain ESBLs in neonates beyond what has previously been reported for LMICs, and identify key sociodemographic correlates associated with ARG spread. Though potentially interesting to a broader audience, methodological, semantic, and technical flaws limit the impact of this work:

Major Comments:

1) The resistome is commonly defined as 'the collection of resistance genes in a given environment' (Wright, *Nature Reviews Microbiology* 2007; Crofts et al. *Nature Reviews Microbiology* 2017). The authors use of this term in the title and throughout the manuscript is misleading as they characterize the abundance and spread of selected ESBLs and carbapenemases in a set of Gram-negative bacterial isolates selectively cultured from rectal swabs. These are not an appropriate approximation for the complete resistome, as they provide no information on the overall diversity or richness of resistance genes in the intestinal microbiome of the enrolled cohort. The authors should therefore modify their language at all relevant sections throughout the manuscript and avoid the term 'resistome' to describe the specific gene assays which they performed.

2) A large fraction of the manuscript is entirely descriptive in nature. While this is inherently not a problem, the level of detail provided makes the study's results hard to digest. Throughout the first half of the Results section the reader is lost in detail without necessary highlighting of key observations. The presentation of these results may be streamlined by the implementation of a statistical analysis comparing resistance gene distribution in different geographical areas and focusing description of the results of resistance genes significantly overrepresented in specific regions. It would also be interesting to assess whether specific resistance genes are taxonomically restricted (both on the species and sequence type level) within the whole-genome sequenced Enterobacteriaceae species. More detailed statistical analysis could help the authors to condense and streamline these result sections.

3) Overall, description of the statistical analysis used in this study lacks detail, which complicates evaluating the validity of the authors conclusions. Thus, statistical tests used are often not referenced and sample-sizes, test statistics, and P-values are not provided in text. This is especially true in the Results section associating the presence of resistance genes with clinical outcomes. Moreover, it is unclear whether the authors corrected the P-values of their risk factor evaluation for multiple hypothesis testing. Additional detail is required to provide clarity on methodology and validity of data interpretation throughout the manuscript.

4) The authors claim clonality of bacteria isolated from unrelated neonates within the same hospital system. Unfortunately, the employed methodology lacks the appropriate resolution to support these claims. Comparison of consensus sequences generated by genome-assemblers frequently generates spurious SNP calls. The authors should utilize their raw short-read data to call SNPs using an alignment-based approach. Moreover, the tool used by the authors (snpiphy) is not peer-reviewed and while it utilizes well-established tools, input parameter and quality filtering options are limited. The VCFtool-suite offers a user-friendly way to implement a less 'black-boxy' approach to calling SNPs.

Moreover, the authors should provide details on pairwise genome-coverage between isolates

4compared to identify SNPs. Low-coverage of the reference genome may indicate extensive difference of isolate accessory genomes disagreeing with the authors' assumption of isolate clonality.

5) The authors claim that key resistance genes are present on and mobilized among their isolates via widespread antibiotic-resistance plasmids. While this is a sensible hypothesis, the genomic evidence within this cohort in support of this remains unclear. Resistance gene cassettes frequently integrate into the bacterial chromosome and sole co-occurrence of specific replicon types (as determined via PlasmidFinder) and resistance genes does not provide sufficient evidence for plasmidic localization of those genes. Indeed, providing the necessary genomic evidence for this association is often impossible using the fragmented assemblies generate from short-read data. The authors should either provide detailed genomic evidence for resistance gene localization within the genome of their isolates through plasmid or long-read sequencing or remove the relevant claims from their manuscript.

6) The authors overinterpret the identified associations between presence of resistance genes and clinical outcomes (birth outcomes, infections), asserting causality where there are only associations. This is especially problematic where they attribute clinical outcomes to the presence of resistance genes. While some associations (e.g. hygiene and resistance gene load), may indeed be biologically meaningful, attributing birth complications and clinical interventions to the presence of resistance genes lacks any causal support. The authors cannot exclude the possibility that uncharacterized covariates (specifically socioeconomic or medical history seem likely) that also impact resistance gene burden may be the cause for poor health outcomes. The authors should remove these associative sections from their results and discussion or provide clear support for a causal link between resistance genes and health outcomes.

7) The data visualizations and their legends throughout the manuscript require additional attention. For example, data in Figure 1 and Supplementary Figure 3 may be better represented as grouped rather than stacked bar plots such that the length of bars are proportional to the data they represent. Rather than portray raw counts, authors may consider showing "Frequency of recovery of X gene in at least one isolate per sample" for standardization. The sample size (currently written above each bar) can be visually represented as a circle of increasing diameter or deepening color, though this is just a stylistic suggestion.

It is unclear what the bars are on the right side of the phylogenetic trees (green) depicted in Supplementary Figures 6-8, 10, 12, and 13. If they represent SNP distances in respect to the reference genome, the depicted values contradict topology of the phylogenetic tree. For example, the branch length of the phylogenetic tree in Supplementary Figure 13 suggests meaningful differences between the reference (BC-MR16-1) and isolate (BC-MR44-1), while the bar length indicates no SNPs between the two isolates. Similar inconsistencies can be found in all other figures. The authors need to clarify these inconsistencies and extend figure legends with additional information about the data that are depicted.

Minor Comments:

1) In lines 428-434, "Rectal samples" is a vague descriptor that can be misconstrued as a stool sample. This should be replaced with clear terminology (i.e. "Rectal swabs") at least in the Methods, if not throughout the manuscript.

52) Since the focus of the paper is on the ARG burden of neonates specifically (and given that ~84% of rectal "samples" assessed come from mothers), data in lines 92-94 and 98-103 (and Supplementary Table 3) can be written in-text to convey rectal sample counts corresponding to mother vs. neonate for each cohort (BS, NoBS, BSyn) as in Supplementary Table 2. It should also be noted in text that nearly 40% of neonate samples discussed in this work come from just 1 of the 7 LMICs (Bangladesh, 1,117/2,931, Supplementary Table 1, Supplementary Figure 1), and that samples from Ethiopia (ES), India (IN), and Rwanda each comprise ~2.5% of all neonate samples (71, 71, 74 / 2,931, respectively).

3) Is the gestational age for these neonates known? Can you be certain these data portray term neonates in LMICs? There is clear literature evidence showing that premature birth dramatically influences species richness and composition in the first 2 months of life, enriching for *Escherichia coli*, *Enterobacter cloacae*, and *Klebsiella*, all organisms investigated in this work. If this information was not recorded, it should be acknowledged in the Discussion as a limitation, similar to lines 329-334 and 376-379 on lack of antibiotic history of neonates.

4) Panels are not labeled within any figure. The legend of Supplementary Figure 3 reads A-B-A-B. The authors should add information on sequence type for all isolates to Figure 3, 4, 5 to facilitate data interpretation.

5) A significant value in Table 1B is not bolded (Maternal hand-washing blaCTX-M-15 UV).

6) In lines 59-60, "... often beta-lactams due to availability and cost, ..." would benefit from a citation.

7) Lines 128-132 are two sentences that sound similar but show different statistics, causing some confusion when grouped together. Rewriting these sentences for extra clarity would benefit readers.

8) In Lines 157-158, authors state >40% of BR isolates and >55% of MR isolates are resistant to all antibiotics besides tigecycline; however, it is unclear from Figure 2 whether each isolate in this range is pan-resistant or whether >40% and >55% of the BR and MR isolate cohorts are resistant to each antibiotic individually.

9) Line 166: The authors should add "grown on ESBL-selective agar" following "The majority of BR isolates".

10) Lines 428-431 (Methods) should be clarified to note that samples were also taken from neonates with no clinical diagnosis of sepsis from 0 days of life onward. As written, sampling appears to begin at 7 days of life, inconsistent with Figure 1 bar graphs and lines 88-90.

11) This manuscript would benefit from extensive copy-editing to resolve grammatical and sentence structure incongruities throughout the work. Selected examples include "is" (line 58), "0,2% BSyn" (line 95), "enrolment" (lines 264, 271),

Reviewer #2 (Remarks to the Author):

The manuscript from BARNARDS group describing carriage of blaCTX-M-15, blaOXA-48 and blaNDM genes among the rectal samples of mothers, and neonates with either suspected or confirmed BS. The study highlights the importance of WASH, and provides important findings based on WGS. The manuscript is well written. Limitations are clearly described. The findings are very important to the field. I have a few major and minor concerns as follows.

Major concern.

(1) It is unclear who were included in this sub-study; in terms of total no, reasons, inclusion and exclusion from the whole large BERNADRS study.

(1.1) Although there is the flow diagramme (Figure S1) – Flow diagramme should start from 36,348 neonates and 35,040 mothers (from line 91). It is unclear how many neonates and mothers were excluded from collecting BR and MR and due to which reasons. Were neonates without BS and mothers of neonates without BS randomly selected or consecutively selected during a period of time? In short, the flow diagram could start from all neonates and all mothers before moving to rectal samples (one sample per neonate/mother) and providing reasons for not including.

(1.2) Line 92, "Overall, 18,148 rectal samples were analysed: 1,925 from neonates with biological sepsis (BS) and mothers from BS cases; 16,155 were from non-BS (NoBS) cases and 68 were rectal samples from mothers with a multiple pregnancy, whose neonates had different sepsis outcomes (BSyn)." It might be better if it is clearer, for example, "Overall, 18,148 rectal samples were analysed: 626 from neonates with biological sepsis (BS), 1,299 mothers of neonates with BS; 2,305 neonates without BS; 13,850 mothers of neonates without BS; and 68 mothers with a multiple pregnancy, whose neonates had different sepsis outcomes (BSyn).", as from Table S2 (if I understand correctly). It's also unclear to me why the total number of mothers of neonates with BS included into the study was much higher than that of neonates with BS.

(1.3) Line 91, it would be good if this line is clear whether "36,468 neonates" were "36,468 neonates with clinically diagnosed sepsis". And whether "35,040 mothers" were "35,040 mothers of those neonates" This would be helpful for clarity.

(1.4) There were two points mentioning 'dyad' findings but it's unclear what are the denominators (i.e. total no of dyad samples evaluated). It would be better if something like "Of overall, XXX rectal samples from XX pairs of neonates and mothers of BS cases and XX pairs of neonates and mothers of non-BS cases were evaluated.

(2) Line 95-97. In my opinion, % by types of neonates and mothers with BS and without BS is more important. For example, "XX.X%, XX.X% and XX.X% of XXX neonates and XX.X%, XX.X% and XX.X% of XXXX mothers, carried blaCTX-M-15, blaNDM, blaKPC, and blaOXA-48-like genes, respectively."

(3) It is unclear to me whether 'neonates without BS' were different from 'neonates with BS'. It would be good if it's mentioned in the results shortly.

(3.1) For example, for the paragraph between the line 125-136; Was findings different between BS

7nature portfolio

and NoBS

(3.2) Supplementary Figure 3; would the figure be different if it's split for BS and NoBS

(4) How missing variables were treated in multivariable logistic regression? Whether any imputation method was used? All records with any missing data were dropped? As per STARD recommendations, the mechanism used to deal with missing data should be noted shortly in the method.

(5) Line 293-299; the association between ARG carriage and Cesarean section. Would this be associated with the timing of specimen collection of mother? This was because by the study design (line 415-416), I understand that MR samples might be collected while (a) mothers were in labour or (b) immediately post-partum. If a sensitivity analysis for this part was conducted by including mothers whose specimens were collected only while there were in labour, are the findings still present? This was because 'immediately post-partum' of C/S could have ATB prophylaxis during C/S and leading to higher ARG carriage as authors mentioned. However, this additional analysis in supplementary text could help readers understand whether 'ARG carriage prior to delivery' was associated with 'C/S' or not.

Minor concerns,

(1) If possible, Supplementary Table 6, 7, 9 and 12, frequency of each value of each variable should be presented. (e.g. Supplementary Table 14 is clearer) This would allow readers to understand the power and credibility of the logistic regression models (for crude OR and adjusted OR as well).

(2) Methods. Number of colonies per plated selected for evaluation are unclear. It was noted only "Phenotypically distinct bacterial colonies .. were selected" so it's unclear whether only 1 colony per phenotypically similar bacterial colonies was selected, and what the maximum no of colonies selected for the study was.

Reviewer #3 (Remarks to the Author):

The authors present the results of an ambitious and globally important study, investigating the prevalence and risk factors for a select number of gram neg AMR resistance genes in mothers and neonates from LMICs.

There is no doubt that the study is unique, and the breadth of samples collected form around the world important. The microbiology that has been undertaken on the same is quite basic to date (PCR detection of a small number of resistance genes, and culture and genomics on a subset of these), with more of the analysis based on an assessment of risk factors for ARG colonisation. This sample set would be ideal for a metagenomic approach to analysis, and presumably this is planned at some stage.

8nature portfolio

While the sample collection is admirable, there are some concerns with how the data is currently presented and this needs significant consideration before the study could be published. In particular, there are many sample / isolate / resistance genes numbers presented throughout the results, and it is very difficult to follow exactly what is being presented here. The multivariate analysis table is very long and needs to be re-structured, or broken up into different tables. It is very difficult to determine how the resistance rates and risk factors differ across countries/sites, and it would be worth considering presenting summary data by country.

The figures presenting huge lists of species names and resistance rates are an example of how difficult it is to decipher the data in the paper, and it would be worth re-considering an approach to presenting this information.

Specific comments:

1. The title includes the word neontaes, but the methods (line 399) mentions inclusion of infants up to 60 days in the study - this needs to be clarified
2. The abstract requires some more clarity, including some information on the context of the study population, and some measures of statistical "prediction" of risk
3. Line 375 - clarify the terminology regarding loss of viability of the carbapenemase gene

Reviewer #4 (Remarks to the Author):

This is an important large data set based on a well done research strategy and analysis plan (BARNARDS) of the incidence of resistance in neonates with and without sepsis as well as their mothers from multiple low and middle income countries in Africa and Asia. There is a high incidence of carbapenemases (although interesting that almost a complete absence of blaKPC) and a very high incidence of blaCTX-M across the countries. The authors then do an evaluation of epidemiologic risk factors for acquisition which demonstrates precedent across the resistance mechanisms maternal antibiotics in the 3 months prior to delivery was the only risk factor across the board for acquisition. Other items like frequent handwashing, access to wastewater network, type and complications from delivery were risk factors for some genes of drug resistance but not others. Would favor removal of plasmid attribution of genes to a specific plasmid as this cannot be successfully done normally without long read sequencing. As there are several manuscripts which have come from this data set it would be helpful for the authors to emphasize how this manuscript differs for the other work in the intro and potentially the discussion.

Minor:

Abstract Line 45-6: "...transmission of bacteria between neonates, neonates and mothers and within the same neonate" Not sure what is meant by transmission within the same neonate and could the authors clarify.

Figure 1 C is slightly confusing. Understand that the intent is to demonstrate percent over time as babies age but confusing about the percents. Were there not some patients who were tested and negative for any resistance genes (as that was what was understood from the text). This might be more helpful for this type of figure to show a # of the total without resistance as well as type and with

9resistance. For example on day zero Africa there are approximately 93/285 isolates who have resistance and thus 66% do not have one of these resistance genes and that might be nice to display graphically in the bar graphs over time.

Figure 2b—slightly confused by this figure. Not sure it is meaningful to demonstrate resistance to organisms where the intrinsic resistance has been defined (CLSI m100). May be more meaningful to take the isolates which have caused bacteremia and just do that subset of organisms. Could make the graph more readable. Antimicrobials typically defined in the legend

Line 185/362 (and others): The authors allude to the presence of bla_{NDM} on an IncX3 plasmid and did not see long read sequencing as a part of this study and thus would favor the authors clarifying or removing this finding as plasmid attribution of a gene to a plasmid cannot typically be done in type of complex isolates with short read sequencing

Methods: Line 517 Unclear about the choosing of 50-80 genomes and would it not be preferred to have selected at random across NCBI? Can the authors elaborate about this selection process?

Author Rebuttal to Initial comments

Reviewer #1 (Remarks to the Author):

In this work, Carvalho & Sands et al. characterize the distribution of clinically important extended spectrum β -lactamases (ESBLs) and carbapenemases in Gram-negative bacterial isolates selectively cultured from rectal swabs of mothers and their neonates. Authors leverage the BARNARDS network of 12 clinical sites across 7 low- and middle-income countries (LMICs) to collect 2,917 and 15,217 rectal swabs from neonates suspected of sepsis, and their mothers, respectively. Authors culture swabs on antibiotic-infused agar plates and perform multiplex-PCR of four ESBLs and carbapenemases on DNA of cultured isolates to show prevalence of beta-lactamase genes among rectal microbiota of mothers and neonates with suspected sepsis in LMICs. Using MALDI-TOF MS they determine the identity of most isolates grown on the ESBL-selective agar, and sequence the isolates they are able to resuscitate from the predominating taxa (*Enterobacter cloacae*, *Escherichia coli*, *Klebsiella pneumoniae*). The authors use computational tools to show relatedness of isolates as well as co-occurrence of globally impactful ARGs plasmid replicons. Lastly, they integrate their molecular dataset with extensive clinical and behavioral metadata to associate the spread of extended-spectrum resistance genes with poor sanitation, personal hygiene, clinical history and socioeconomic variables.

Overall, the data presented by the authors complement an extensive body of literature characterizing the high abundance of clinically important resistance genes in LMICs with a new multinational study in the context of suspected neonatal sepsis. In this work, authors describe elevated prevalence of certain ESBLs in neonates beyond what has previously been reported for LMICs, and identify key sociodemographic correlates associated with ARG spread. Though potentially interesting to a broader audience, methodological, semantic, and technical flaws limit the impact of this work:

10nature portfolio

Major Comments:

1) The resistome is commonly defined as ‘the collection of resistance genes in a given environment’ (Wright, Nature Reviews Microbiology 2007; Crofts et al. Nature Reviews Microbiology 2017). The authors use of this term in the title and throughout the manuscript is misleading as they characterize the abundance and spread of selected ESBLs and carbapenemases in a set of Gram-negative bacterial isolates selectively cultured from rectal swabs. These are not an appropriate approximation for the complete resistome, as they provide no information on the overall diversity or richness of resistance genes in the intestinal microbiome of the enrolled cohort. The authors should therefore modify their language at all relevant sections throughout the manuscript and avoid the term ‘resistome’ to describe the specific gene assays which they performed.

Response: We agree with the reviewer’s comments and have adjusted the language throughout the text and title accordingly.

2) A large fraction of the manuscript is entirely descriptive in nature. While this is inherently not a problem, the level of detail provided makes the study’s results hard to digest. Throughout the first half of the Results section the reader is lost in detail without necessary highlighting of key observations. The presentation of these results may be streamlined by the implementation of a statistical analysis comparing resistance gene distribution in different geographical areas and focusing description of the results of resistance genes significantly overrepresented in specific regions. It would also be interesting to assess whether specific resistance genes are taxonomically restricted (both on the species and sequence type level) within the whole-genome sequenced Enterobacteriaceae species. More detailed statistical analysis could help the authors to condense and streamline these result sections.

Response: In accordance with the reviewer’s suggestion, we have edited the text to clarify the key observations. Our first main figure (Figure 1, a global map) highlights the ARG distribution for each country/site, and in the main text we refer to the overall prevalence for each maternal and neonatal datasets, also considering the prevalence by geographical region. Primarily due to sample size restrictions for certain countries/sites (with positive ARG samples) we have not performed statistical analysis per country. Moreover, our site partners are working to publish their own descriptive datasets as

11part of our commitment to capacity building and we are supporting their ongoing work. All statistical analysis performed is however, available in the supplementary material and we have amended the methods section to better reflect the detail of analysis performed within this study.

We agree with the comment regarding the resistance genes and taxonomic data (both species and ST) and have edited the text in the results section (lines 183-256) to highlight the results in relation to ARG – species and ST. We selected *E. coli*, *E. cloacae* and *K. pneumoniae* for WGS based on these species being the dominant bacteria from both the maternal and neonatal samples (from our ESBL selective methodology; Figure 2). Supplementary Figure 1 highlights the proportion of carbapenemase positive isolates and clearly shows a higher prevalence in South Asia (for maternal rectal swabs we found 461 isolates from South Asia and 55 isolates from Africa, all denominators are described in Supplementary Figure 1). Additionally, we found 522 isolates from neonatal rectal samples from South Asia and 34 from Africa. The final section of Supplementary Figure 1 illustrates what proportion of each species (*E. coli*, *K. pneumoniae* and *E. cloacae*) were whole genome sequenced.

3) Overall, description of the statistical analysis used in this study lacks detail, which complicates evaluating the validity of the authors conclusions. Thus, statistical tests used are often not referenced and sample-sizes, test statistics, and P-values are not provided in text. This is especially true in the Results section associating the presence of resistance genes with clinical outcomes. Moreover, it is unclear whether the authors corrected the P-values of their risk factor evaluation for multiple hypothesis testing. Additional detail is required to provide clarity on methodology and validity of data interpretation throughout the manuscript.

Response: We have provided additional description of the statistical analysis used in the study in the methods section (lines 574–608). P-values were not adjusted for multiple testing as models used were hypothesis generating/exploratory in nature (lines 600-602). Due to length constraints we have not listed each statistical test and p-value (corrected) in the results section, however all data is accessible in Table 1 and in Supplementary Material (Supplementary Tables 11, 12, 14-19).

4) The authors claim clonality of bacteria isolated from unrelated neonates within the same hospital system. Unfortunately, the employed methodology lacks the appropriate resolution to support these claims. Comparison of consensus sequences generated by genome-assemblers frequently generates

12spurious SNP calls. The authors should utilize their raw short-read data to call SNPs using an alignment-based approach. Moreover, the tool used by the authors (snpiphy) is not peer-reviewed and while it utilizes well-established tools, input parameter and quality filtering options are limited. The VCFtool-suite offers a user-friendly way to implement a less 'black-boxy' approach to calling SNPs.

Moreover, the authors should provide details on pairwise genome-coverage between isolates compared to identify SNPs. Low-coverage of the reference genome may indicate extensive difference of isolate accessory genomes disagreeing with the authors' assumption of isolate clonality.

Response: We fully concur with the reviewer and have reanalysed all SNP data with fastq as input for read mapping and alignment against a reference. SNP analysis was performed using snippy to call the SNPs from the fastq (R1 and R2 files) and produce the alignment. Gubbins was used to account for recombination and snp-sites was used to extract SNP sites for phylogeny reconstruction. IQ-tree was used to generate a maximum likelihood tree and snp-dists was used to create a pairwise SNP matrix. Furthermore, to maximise the accuracy of SNP calling (as per SJ Bush, 2020 doi: [10.1093/gigascience/giaa007](https://doi.org/10.1093/gigascience/giaa007)), corresponding long reads were generated for a single isolate from each clade to produce a high quality and deep coverage reference genome. Please see the revised methods (lines 532-539). We have excluded some isolates from each SNP tree, dependent on coverage parameters picked up during mapping. We thank the reviewer for this comment as the repeat SNP analysis has, with increased QC checks, improved the quality of our analysis, and this can be reflected in the replaced SNP trees, where applicable (Supplementary Figures 5-8).

5) The authors claim that key resistance genes are present on and mobilized among their isolates via widespread antibiotic-resistance plasmids. While this is a sensible hypothesis, the genomic evidence within this cohort in support of this remains unclear. Resistance gene cassettes frequently integrate into the bacterial chromosome and sole co-occurrence of specific replicon types (as determined via PlasmidFinder) and resistance genes does not provide sufficient evidence for plasmidic localization of those genes. Indeed, providing the necessary genomic evidence for this association is often impossible using the fragmented assemblies generate from short-read data. The authors should either provide detailed genomic evidence for resistance gene localization within the genome of their isolates through plasmid or long-read sequencing or remove the relevant claims from their manuscript.

Response: Whilst in some cases it was possible for us to link the plasmid inc types to ARGs, we agree with the reviewers in that the use of short read sequencing often results in highly fragmented assemblies and limited information linking ARG-plasmids is available (purely dependent on the contig size). To further compliment short read sequencing analysis we have performed long read sequencing (MinION, ONT) on n=50 isolates, chosen based on short read sequencing analysis (ARG carriage, ST and core genome phylogeny) and other metadata available (clinical site, sample type, date). Data concerning the hybrid genome assemblies in relation to ARG and plasmid can be found in the supplementary material (Supplementary Table 6).

6) The authors overinterpret the identified associations between presence of resistance genes and clinical outcomes (birth outcomes, infections), asserting causality where there are only associations. This is especially problematic where they attribute clinical outcomes to the presence of resistance genes. While some associations (e.g. hygiene and resistance gene load), may indeed be biologically meaningful, attributing birth complications and clinical interventions to the presence of resistance genes lacks any causal support. The authors cannot exclude the possibility that uncharacterized covariates (specifically socioeconomic or medical history seem likely) that also impact resistance gene burden may be the cause for poor health outcomes. The authors should remove these associative sections from their results and discussion or provide clear support for a causal link between resistance genes and health outcomes.

Response: We do agree with the reviewer's comments and have edited the language used to avoid making definitive statements that the carriage of ARG is causal (lines 313-318 have been tempered accordingly). Furthermore, we have added an additional sentence to the discussion (lines 360-362) to emphasise the exploratory nature of the statistical analyses performed within this study and highlight that although our results may indicate associations, we acknowledge the limitations.

7) The data visualizations and their legends throughout the manuscript require additional attention. For example, data in Figure 1 and Supplementary Figure 3 may be better represented as grouped rather than stacked bar plots such that the length of bars are proportional to the data they represent. Rather than portray raw counts, authors may consider showing "Frequency of recovery of X gene in at least one isolate per sample" for standardization. The sample size (currently written above each bar) can be visually represented as a circle of increasing diameter or deepening color, though this is just a stylistic suggestion.

nature portfolio

Response: We agree with the reviewer and we have expanded all legends throughout the manuscript for clarity. For Figure 1c and Supplementary Figure 3 we have reproduced the data grouped rather than stacked bar plots as per the reviewer suggestions. We have presented the data as the frequency of the ARG per sample. We have replaced the sample size numeric with a circle of increasing diameter, however the original number may be easier to interpret for similar values.

It is unclear what the bars are on the right side of the phylogenetic trees (green) depicted in Supplementary Figures 6-8, 10, 12, and 13. If they represent SNP distances in respect to the reference genome, the depicted values contradict topology of the phylogenetic tree. For example, the branch length of the phylogenetic tree in Supplementary Figure 13 suggests meaningful differences between the reference (BC-MR16-1) and isolate (BC-MR44-1), while the bar length indicates no SNPs between the two isolates. Similar inconsistencies can be found in all other figures. The authors need to clarify these inconsistencies and extend figure legends with additional information about the data that are depicted.

Response: We have repeated all SNP analysis with a more accurate method utilising the raw sequence reads with a high quality close reference genome (generated with complementary long read sequencing). The previous SNP trees combined SNP pairwise distance with SNP distance per branch, and we agree this was unclear. During repeat analysis we have increased our QC threshold and isolates with a lower coverage were excluded (as this was detected more readily using the fastw compared to the genome). All figure legends have been expanded augmenting better interpretation of the data.

Minor Comments:

1) In lines 428-434, “Rectal samples” is a vague descriptor that can be misconstrued as a stool sample. This should be replaced with clear terminology (i.e. “Rectal swabs”) at least in the Methods, if not throughout the manuscript.

Response: We agree with the reviewer’s comments and have amended the text accordingly throughout to “rectal swab”.

2) Since the focus of the paper is on the ARG burden of neonates specifically (and given that ~84% of rectal “samples” assessed come from mothers), data in lines 92-94 and 98-103 (and Supplementary Table 3) can be written in-text to convey rectal sample counts corresponding to mother vs. neonate for each cohort (BS, NoBS, BSyn) as in Supplementary Table 2. It should also be noted in text that nearly 40% of neonate samples discussed in this work come from just 1 of the 7 LMICs (Bangladesh, 1,117/2,931, Supplementary Table 1, Supplementary Figure 1), and that samples from Ethiopia (ES), India (IN), and Rwanda each comprise ~2.5% of all neonate samples (71, 71, 74 / 2,931, respectively).

Response: We agree with the reviewer and have amended the first section of results. We described the findings for mothers’ and neonates’ rectal swabs outlining results for BS, NoBS and BSyn in the main text. Overall data per site and country is shown in Supplementary Table 1 and, in Supplementary Table 2, data per site/country for each BS, NoBS and BSyn cohort is available.

The number and species identification of isolates recovered per site/country overall are depicted in Supplementary Table 3 and in Supplementary Tables 4 and 5 these data are shown for BR (BS, NoBS) and MR (BS, NoBS and BSyn), respectively.

Of note, based on reviewer comments we have amended the article title, however we agree with your comment with the emphasis on the ARG burden of neonates. We have addressed in the limitations section that nearly 40% of neonate samples discussed in this work come from one of the seven LMICs and that there is an under representation of neonate samples from certain LMICs (lines 401-403).

3) Is the gestational age for these neonates known? Can you be certain these data portray term neonates in LMICs? There is clear literature evidence showing that premature birth dramatically influences species richness and composition in the first 2 months of life, enriching for *Escherichia coli*, *Enterobacter cloacae*, and *Klebsiella*, all organisms investigated in this work. If this information was not recorded, it should be acknowledged in the Discussion as a limitation, similar to lines 329-334 and 376-379 on lack of antibiotic history of neonates.

Response: Thank you for this comment. We agree that this evidence should be considered and we retrieved the gestational age information from our dataset. Among data available for 34,671 neonates, 76% were term neonates (37-41 weeks; n= 27,654), 14% were preterm (<37 weeks; n= 5119) and 5% were postterm (>41 weeks; n= 1898). Specifically, these 2,931 samples were from 2011 term (69%), 736 (25%) preterm and 147 (5%) postterm neonates; data was missing for 37 enrolled neonates (1%).

Accordingly, we believe the data reported in this work largely reflect microbiota diversity of term neonates in these sites. This has been added in lines 171-174.

4) Panels are not labeled within any figure. The legend of Supplementary Figure 3 reads A-B-A-B. The authors should add information on sequence type for all isolates to Figure 3, 4, 5 to facilitate data interpretation.

Response: We have clarified the labelling of panels within all figures. Thank you for edifying us regarding the typo with the legend of Supplementary Figure 3 - this has now been corrected. We have re-produced the core genome phylogenetic trees (Figures 3 - 5) with the addition of ST information for all isolates.

5) A significant value in Table 1B is not bolded (Maternal hand-washing blaCTX-M-15 UV).

Response: Thank you for noting this - Table 1 has been revised and amended accordingly.

To simplify the visualisation of the exploratory multivariable statistical analysis we performed to understand associations between socio-demographic and clinical data and maternal and neonatal carriage of ARGs, we have done forest plots for (A) association between WASH related variables and maternal carriage of ARGs and (E) association between birth healthcare environment features and carriage of ARGs among neonates from the birth cohort (added Figure 6). Presently, Table 1 contains (B now A) association between the mother's handwashing frequency and maternal carriage of ARGs; (C now B) association between maternal infection in the three-months prior to enrolment in the study and maternal carriage of ARGs; (D now C) association between maternal usage of antibiotics in the three-months prior to enrolment in the study and maternal carriage of ARGs.

6) In lines 59-60, "... often beta-lactams due to availability and cost, ..." would benefit from a citation.

nature portfolio

Response: We agree with the reviewer and have added a citation accordingly (line 60).

7) Lines 128-132 are two sentences that sound similar but show different statistics, causing some confusion when grouped together. Rewriting these sentences for extra clarity would benefit readers.

Response: The two sentences describe the data from the Asian continent and the African continent. We agree with the reviewer in that the similarity can cause confusion and have edited the text (lines 119-135). Also, we added another analysis to understand if the age of neonate vs carriage of the ARG in study was different between BS and NoBS samples.

8) In Lines 157-158, authors state >40% of BR isolates and >55% of MR isolates are resistant to all antibiotics besides tigecycline; however, it is unclear from Figure 2 whether each isolate in this range is pan-resistant or whether >40% and >55% of the BR and MR isolate cohorts are resistant to each antibiotic individually.

Response: We agree with the reviewer and we have expanded the text accordingly. The text now states: "Apart from tigecycline, resistance rates to each antibiotic tested were $\geq 40\%$ among BR isolates. For MR isolates, 9% were resistant to tigecycline and $\geq 55\%$ were resistant to each of the other antibiotics" (lines 166-168).

9) Line 166: The authors should add "grown on ESBL-selective agar" following "The majority of BR isolates".

Response: We agree with the reviewer and have amended the text accordingly (lines 178-181).

nature portfolio

10) Lines 428-431 (Methods) should be clarified to note that samples were also taken from neonates with no clinical diagnosis of sepsis from 0 days of life onward. As written, sampling appears to begin at 7 days of life, inconsistent with Figure 1 bar graphs and lines 88-90.

Response: We agree with the reviewer and have amended the text accordingly.

This is also stated in the results section (lines 91-93).

11) This manuscript would benefit from extensive copy-editing to resolve grammatical and sentence structure incongruencies throughout the work. Selected examples include “is” (line 58), “0,2% BSyn” (line 95), “enrolment” (lines 264, 271).

Response: We agree with the reviewer and have resolved grammatical errors and inconsistencies throughout the manuscript.Reviewer #2 (Remarks to the Author):

The manuscript from BARNARDS group describing carriage of blaCTX-M-15, blaOXA-48 and blaNDM genes among the rectal samples of mothers, and neonates with either suspected or confirmed BS. The study highlights the importance of WASH, and provides important findings based on WGS. The manuscript is well written. Limitations are clearly described. The findings are very important to the field. I have a few major and minor concerns as follows.

Major concern.

(1) It is unclear who were included in this sub-study; in terms of total no, reasons, inclusion and exclusion from the whole large BERNARDS study.

(1.1) Although there is the flow diagramme (Figure S1) – Flow diagramme should start from 36,348 neonates and 35,040 mothers (from line 91). It is unclear how many neonates and mothers were excluded from collecting BR and MR and due to which reasons. Were neonates without BS and mothers of neonates without BS randomly selected or consecutively selected during a period of time? In short, the flow diagram could start from all neonates and all mothers before moving to rectal samples (one sample per neonate/mother) and providing reasons for not including.

Response: According to the BARNARDS protocol all mothers in labour presenting in hospital agreeing to participate were enrolled together with their neonates. Rectal samples were collected from all mothers and from neonates ≥ 7 days old clinically diagnosed with sepsis. Additionally, neonates who presented to clinical sites with clinically suspected sepsis in the first 60 days of life were recruited (with their mothers) upon consent. From all sites there were BR from neonates < 7 days old and these samples were also included.

Given time and resources constraints, to aid the sample size for processing, we used the available data on sepsis incidence (published data Sands et al., 2021; citation added in methods section where sampling is described, lines 448-449) and processed three times the number of mother rectal samples in total. We selected samples across the duration of the sampling, referring to the local sepsis rates per month. For the neonate (BR) swabs, these were largely collected for ≥ 7 days old neonates, and therefore we received

20nature portfolio

fewer swabs in total and prioritised the processing of all (BR) rectal swabs. In Figure S1 we have included all denominators.

(1.2) Line 92, “Overall, 18,148 rectal samples were analysed: 1,925 from neonates with biological sepsis (BS) and mothers from BS cases; 16,155 were from non-BS (NoBS) cases and 68 were rectal samples from mothers with a multiple pregnancy, whose neonates had different sepsis outcomes (BSyn).” It might be better if it is clearer, for example, “Overall, 18,148 rectal samples were analysed: 626 from neonates with biological sepsis (BS), 1,299 mothers of neonates with BS; 2,305 neonates without BS; 13,850 mothers of neonates without BS; and 68 mothers with a multiple pregnancy, whose neonates had different sepsis outcomes (BSyn).”, as from Table S2 (if I understand correctly). It’s also unclear to me why the total number of mothers of neonates with BS included into the study was much higher than that of neonates with BS.

Response: We have edited the entire first section of the results for clarity based on all reviewers suggestions.

The total number of mothers of neonates with BS included into the study was much higher than that of neonates with BS because as per the BARNARDS protocol, BR were collected from all mothers and only from neonates ≥ 7 days old clinically diagnosed with sepsis. Even though, there were BR collected from < 7 days old neonates, not all neonates have been sampled.

(1.3) Line 91, it would be good if this line is clear whether “36,468 neonates” were “36,468 neonates with clinically diagnosed sepsis”. And whether “35,040 mothers” were “35,040 mothers of those neonates” This would be helpful for clarity.

Response: Thank you for this comment. For clarity, mothers presenting to the BARNARDS clinical sites in labour and their respective liveborn neonates were enrolled. Not all 36,348 neonates were diagnosed with clinical sepsis. Samples were only collected from neonates with clinical signs of sepsis. We have amended the text, lines 85-86.

21nature portfolio

(1.4) There were two points mentioning 'dyad' findings but it's unclear what are the denominators (i.e. total no of dyad samples evaluated). It would be better if something like "Of overall, XXX rectal samples from XX pairs of neonates and mothers of BS cases and XX pairs of neonates and mothers of non-BS cases were evaluated.

Response: We agree with the reviewer and have amended the text accordingly. The reference to 'dyad' has been reworded in lines 195, 256 and in figures' legends.

(2) Line 95-97. In my opinion, % by types of neonates and mothers with BS and without BS is more important. For example, "XX.X%, XX.X% and XX.X% of XXX neonates and XX.X%, XX.X% and XX.X% of XXXX mothers, carried blaCTX-M-15, blaNDM, blaKPC, and blaOXA-48-like genes, respectively."

Response: We agree with the reviewer and have amended the results, lines 94-97 and lines 105-107.

(3) It is unclear to me whether 'neonates without BS' were different from 'neonates with BS'. It would be good if it's mentioned in the results shortly.

(3.1) For example, for the paragraph between the line 125-136; Was findings different between BS and NoBS

(3.2) Supplementary Figure 3; would the figure be different if it's split for BS and NoBS

Response: Thank you for this suggestion. We have analysed carriage of ARG vs neonatal age per continent to understand if there were differences between BS and NoBS. As with delivery type, ARG were consistently found among BR, regardless of neonates developing BS or not, both in Asia and Africa.

22Among BR from Asian neonates developing BS, 80% (121/152), 54% (82/152) and 29% (44/152) carried *bla*_{CTX-M-15}, *bla*_{NDM}, and *bla*_{OXA-48}-like genes, respectively, while NoBS Asian BR carried 57% (250/441), 33% (145/441) and 7% (33/441) (Supplementary Figure 3a).

Among BR from African neonates developing BS, 58% (139/239), 5% (12/239) and 1% (3/239) carried *bla*_{CTX-M-15}, *bla*_{NDM}, and *bla*_{OXA-48}-like genes, respectively, whereas NoBS African BR carried 41% (274/674), 3% (21/674) and 1% (4/674) (Supplementary Figure 3b).

Thus, there were higher rates of ARG carriage in BS Asian BR during the first 14 days of life, which was also seen for African samples, although with substantially lower differences.

Similarly, among neonates born by C-section in Asia, the rates of ARG carriage during the first 14 days of life were higher. This was not seen in neonates from Africa where the carriage of ARG during the same time period was similar for SVD and C-section delivered babies (lines 119-139; Supplementary Figure 3).

(4) How missing variables were treated in multivariable logistic regression? Whether any imputation method was used? All records with any missing data were dropped? As per STARD recommendations, the mechanism used to deal with missing data should be noted shortly in the method.

Response: We concur with the reviewer's comments and have expanded our statistical methods sections to include these details, lines 601-607.

(5) Line 293-299; the association between ARG carriage and Cesarean section. Would this be associated with the timing of specimen collection of mother? This was because by the study design (line 415-416), I understand that MR samples might be collected while (a) mothers were in labour or (b) immediately post-partum. If a sensitivity analysis for this part was conducted by including mothers whose specimens were collected only while there were in labour, are the findings still present? This was because 'immediately post-partum' of C/S could have ATB prophylaxis during C/S and leading to higher ARG carriage as authors mentioned. However, this additional analysis in supplementary text could help readers understand whether 'ARG carriage prior to delivery' was associated with 'C/S' or not.

nature portfolio

Response: We agree with the reviewer's comment regarding association between ARG and C-section. It was advised (as explained in the protocols for BARNARDS) that all samples were collected pre-partum; however, we unfortunately do not have the detailed data (i.e. exact time of MR compared to time of CS) to perform such a sensitivity analysis. We have emphasised the exploratory nature of statistical analysis performed during this study (lines 360-362).

Minor concerns,

(1) If possible, Supplementary Table 6, 7, 9 and 12, frequency of each value of each variable should be presented. (e.g. Supplementary Table 14 is clearer) This would allow readers to understand the power and credibility of the logistic regression models (for crude OR and adjusted OR as well).

Response: We concur with the reviewer and have added Supplementary Table 10 with descriptive data to allow readers to understand the power behind the analyses performed in the study.

(2) Methods. Number of colonies per plated selected for evaluation are unclear. It was noted only "Phenotypically distinct bacterial colonies .. were selected" so it's unclear whether only 1 colony per phenotypically similar bacterial colonies was selected, and what the maximum no of colonies selected for the study was.

Response: Based on the quantity of phenotypically distinct colonies present (i.e. often up to 6 per agar plate), only one colony per phenotypically similar bacterial colonies was selected for purification for repeat PCR. Accordingly, this would often result in more than one colony being screened per swab. We do however acknowledge limitations with this approach, and we have added this to the limitations section for clarity (lines 395-398). We did not quantify the maximum number of colonies selected per VE plate (following a positive result for blaNDM/OXA-48-like) as this was beyond the remit of this study.

Reviewer #3 (Remarks to the Author):

The authors present the results of an ambitious and globally important study, investigating the prevalence and risk factors for a select number of gram neg AMR resistance genes in mothers and neonates from LMICs.

There is no doubt that the study is unique, and the breadth of samples collected from around the world is important. The microbiology that has been undertaken on the same is quite basic to date (PCR detection of a small number of resistance genes, and culture and genomics on a subset of these), with more of the analysis based on an assessment of risk factors for ARG colonisation. This sample set would be ideal for a metagenomic approach to analysis, and presumably this is planned at some stage.

While the sample collection is admirable, there are some concerns with how the data is currently presented and this needs significant consideration before the study could be published.

In particular, there are many sample / isolate / resistance genes numbers presented throughout the results, and it is very difficult to follow exactly what is being presented here. The multivariate analysis table is very long and needs to be re-structured, or broken up into different tables. It is very difficult to determine how the resistance rates and risk factors differ across countries/sites, and it would be worth considering presenting summary data by country.

Response: We concur with the reviewer in that Table 1 is very long and contains vast amounts of data and, thus, the table has been amended. To simplify the visualisation of the exploratory multivariable statistical analysis we performed to understand associations between socio-demographic and clinical data and maternal and neonatal carriage of ARGs, we have done forest plots for (A) association between WASH related variables and maternal carriage of ARGs and (E) association between birth healthcare environment features and carriage of ARGs among neonates from the birth cohort (added Figure 6) and the remaining analyses remained in Table 1.

Whilst we have presented the resistance rates per country (global maps in Figure 1), we have presented the multivariate analysis collectively. This is due to two main reasons - Firstly, for the majority of

25nature portfolio

variables the sample size was not high enough to allow for country level analysis. Secondly, a key aspect of the BARNARDS study was to provide capacity building and encourage local researchers to analyse country level data.

The figures presenting huge lists of species names and resistance rates are an example of how difficult it is to decipher the data in the paper, and it would be worth re-considering an approach to presenting this information.

Response: We concur with the reviewer and in accordance to responding to other comments, we have reproduced Figure 2 to convey data relating to species identity. However, we believe it is pertinent to acknowledge the wide diversity of bacterial species carrying carbapenemase ARG in both the mothers' and neonates' microbiota.

Specific comments:

1. The title includes the word neontaes, but the methods (line 399) mentions inclusion of infants up to 60 days in the study - this needs to be clarified

Response: We have now clarified this point. We have used the term neonate throughout this study (throughout all manuscripts stemming from BARNARDS; Sands et al., 2021, Thomson et al 2021). This has been described in the methods (line 432), however we have also added this to the abstract to allow the reader to understand this from the onset of the article.

2. The abstract requires some more clarity, including some information on the context of the study population, and some measures of statistical "prediction" of risk

26nature portfolio

Response: We concur with the reviewer, and although the abstract is less than 200 words, we have amended the current text to better reflect the context of the statistical analyses (associations) performed.

3. Line 375 - clarify the terminology regarding loss of viability of the carbapenemase gene

Response: To clarify, we included 'loss of viability of the carbapenemase gene', and this phrase was used to describe all bacterial isolates that did not recover/grow well on solid agar and/or the ARG was not detected from repeat PCR analysis prior to sequencing. We have simplified this in the text for clarity (lines 393-395).

Reviewer #4 (Remarks to the Author):

This is an important large data set based on a well done research strategy and analysis plan (BARNARDS) of the incidence of resistance in neonates with and without sepsis as well as their mothers from multiple low and middle income countries in Africa and Asia. There is a high incidence of carbapenemases (although interesting that almost a complete absence of blaKPC) and a very high incidence of blaCTX-M across the countries. The authors then do an evaluation of epidemiologic risk factors for acquisition which demonstrates precedent across the resistance mechanisms maternal antibiotics in the 3 months prior to delivery was the only risk factor across the board for acquisition. Other items like frequent handwashing, access to wastewater network, type and complications from delivery were risk factors for some genes of drug resistance but not others. Would favor removal of plasmid attribution of genes to a specific plasmid as this cannot be successfully done normally without long read sequencing. As there are several manuscripts which have come from this data set it would be helpful for the authors to emphasize how this manuscript differs for the other work in the intro and potentially the discussion.

Response: Whilst in some cases it was possible for us to link the plasmid inc types to ARGs, we agree with the reviewers in that the use of short read sequencing often results in highly fragmented assemblies and limited information linking ARG-plasmids is available (purely dependent on the contig size). To further compliment short read sequencing analysis we have performed long read sequencing (MinION, ONT) on n=50 isolates, chosen based on short read sequencing analysis (ARG carriage, ST and core genome phylogeny) and other metadata available (clinical site, sample type, date). Data concerning the hybrid genome assemblies in relation to ARG and plasmid can be found in the supplementary material (Supplementary Table 6).

Minor:

Abstract Line 45-6: "...transmission of bacteria between neonates, neonates and mothers and within the same neonate" Not sure what is meant by transmission within the same neonate and could the authors clarify.

Response: We have amended the sentence to clarify. We have edited the abstract in a few places to highlight key messages.

Figure 1 C is slightly confusing. Understand that the intent is to demonstrate percent over time as babies age but confusing about the percents. Were there not some patients who were tested and negative for any resistance genes (as that was what was understood from the text). This might be more helpful for this type of figure to show a # of the total without resistance as well as type and with resistance. For example on day zero Africa there are approximately 93/285 isolates who have resistance and thus 66% do not have one of these resistance genes and that might be nice to display graphically in the bar graphs over time.

Response: We agree with the reviewer and we have reproduced Figure 1c (and also Supplementary Figure 3) showing grouped bar graphs to clarify the key messages. There were neonates who were tested and negative for resistance genes and we believe Figure 1c is now clearer for interpretation of data.

Figure 2b—slightly confused by this figure. Not sure it is meaningful to demonstrate resistance to organisms where the intrinsic resistance has been defined (CLSI m100). May be more meaningful to take the isolates which have caused bacteremia and just do that subset of organisms. Could make the graph more readable. Antimicrobials typically defined in the legend

Response: We agree with the reviewer, Figure 2 is complex and attempts to convey a lot of data. To clarify, and in line with other comments, we have removed the presentation of the resistance data in Figure 2, which is now in Supplementary Figure 4.

All bacterial isolates analysed in this article were recovered from rectal swabs (carriage) and were not blood isolates. With this analysis, our aim was to highlight the carriage of antimicrobial resistant bacterial isolates within the mothers' and neonates' microbiota. In fact, our aim is to show the AMR profile of the neonates' and mother's gut microbiota which we were able to capture and not to report resistance rates of isolates, and, thus, show that the gut microbiota of community women and neonates presents a worrying AMR profile.

nature portfolio

Line 185/362 (and others): The authors allude to the presence of blaNDM on an IncX3 plasmid and did not see long read sequencing as a part of this study and thus would favor the authors clarifying or removing this finding as plasmid attribution of a gene to a plasmid cannot typically be done in type of complex isolates with short read sequencing.

Response: Whilst in some cases it was possible for us to link the plasmid inc types to ARGs, we agree with the reviewer's in that the use of short read sequencing often results in highly fragmented assemblies and limited information linking ARG-plasmids is available (purely dependent on the contig size). This aspect of the manuscript was picked up by >1 reviewer and therefore to further compliment short read sequencing analysis we have performed long read sequencing (MinION, ONT) on n=50 isolates, chosen based on short read sequencing analysis (ARG carriage, ST and core genome phylogeny) and other metadata available (clinical site, sample type, date). Data concerning the hybrid genome assemblies in relation to ARG and plasmid can be found in the supplementary material (Supplementary Table 6).

Methods: Line 517 Unclear about the choosing of 50-80 genomes and would it not been preferred to have selected at random across NBCI? Can the authors elaborate about this selection process?

Response: For clarification, we aimed to identify between 50-80 genomes at random, however due to the over representation of certain ST groups, we aimed to first narrow down a sub selection across each species, and then selected at random. Whilst we acknowledge that there is still some element of selection bias, we have included a fairly diverse representation across the species in attempt to place our isolates into a global context.

Decision Letter, first revision:

Dear Dr. Carvalho,

Thank you for your patience while your manuscript "Antibiotic resistance genes in the gut microbiota of mothers and neonates from low- and middle-income countries: prevalence, risk factors and genomics" was under peer-review at Nature Microbiology. It has now been seen by 4 referees, whose

30nature portfolio

expertise and comments you will find at the of this email. You will see from their comments below that while they find your work of interest, some important points are raised. We are very interested in the possibility of publishing your study in *Nature Microbiology*, but would like to consider your response to these concerns in the form of a revised manuscript before we make a final decision on publication.

In particular, you will see that while referees #2 and #3 do not raise any further comment, referee #1 asks you to improve the clarity and transparency of the statistical methodology used for the statistical associations, in order to improve the robustness of the study. Referee #4 also raise several concerns that need to be addressed, including the need to avoid shortening of terms to improve readability of the manuscript, to clarify some conclusions, to improve figures. In addition to these points, we have noticed that the website <https://barnards-group.com> is not functional. Please, address the referees comments. Also, supply the clear original statistical analysis plan and the corrected p-values, and adhere to the STROBE guidance to report observational studies.

If you have not done so already please begin to revise your manuscript so that it conforms to our Article format instructions at <http://www.nature.com/nmicrobiol/info/final-submission/>

The usual length limit for a *Nature Microbiology* Article is six display items (figures or tables) and 3,000 words. We have some flexibility, and can allow a revised manuscript at 3,500 words, but please consider this a firm upper limit. There is a trade-off of ~250 words per display item, so if you need more space, you could move a Figure or Table to Supplementary Information.

Some reduction could be achieved by focusing any introductory material and moving it to the start of your opening 'bold' paragraph, whose function is to outline the background to your work, describe in a sentence your new observations, and explain your main conclusions. The discussion should also be limited. Methods should be described in a separate section following the discussion, we do not place a word limit on Methods.

Nature Microbiology titles should give a sense of the main new findings of a manuscript, and should not contain punctuation. Please keep in mind that we strongly discourage active verbs in titles, and that they should ideally fit within 90 characters each (including spaces).

Please include a data availability statement as a separate section after Methods but before references,

31nature portfolio

under the heading "Data Availability". This section should inform readers about the availability of the data used to support the conclusions of your study. This information includes accession codes to public repositories (data banks for protein, DNA or RNA sequences, microarray, proteomics data etc...), references to source data published alongside the paper, unique identifiers such as URLs to data repository entries, or data set DOIs, and any other statement about data availability. At a minimum, you should include the following statement: "The data that support the findings of this study are available from the corresponding author upon request", mentioning any restrictions on availability. If DOIs are provided, we also strongly encourage including these in the Reference list (authors, title, publisher (repository name), identifier, year). For more guidance on how to write this section please see:

<http://www.nature.com/authors/policies/data/data-availability-statements-data-citations.pdf>

To improve the accessibility of your paper to readers from other research areas, please pay particular attention to the wording of the paper's opening bold paragraph, which serves both as an introduction and as a brief, non-technical summary in about 150 words. If, however, you require one or two extra sentences to explain your work clearly, please include them even if the paragraph is over-length as a result. The opening paragraph should not contain references. Because scientists from other sub-disciplines will be interested in your results and their implications, it is important to explain essential but specialised terms concisely. We suggest you show your summary paragraph to colleagues in other fields to uncover any problematic concepts.

If your paper is accepted for publication, we will edit your display items electronically so they conform to our house style and will reproduce clearly in print. If necessary, we will re-size figures to fit single or double column width. If your figures contain several parts, the parts should form a neat rectangle when assembled. Choosing the right electronic format at this stage will speed up the processing of your paper and give the best possible results in print. We would like the figures to be supplied as vector files - EPS, PDF, AI or postscript (PS) file formats (not raster or bitmap files), preferably generated with vector-graphics software (Adobe Illustrator for example). Please try to ensure that all figures are non-flattened and fully editable. All images should be at least 300 dpi resolution (when figures are scaled to approximately the size that they are to be printed at) and in RGB colour format. Please do not submit Jpeg or flattened TIFF files. Please see also 'Guidelines for Electronic Submission of Figures' at the end of this letter for further detail.

Figure legends must provide a brief description of the figure and the symbols used, within 350 words, including definitions of any error bars employed in the figures.

When submitting the revised version of your manuscript, please pay close attention to our [href="https://www.nature.com/nature-research/editorial-policies/image-integrity">Digital Image Integrity Guidelines](https://www.nature.com/nature-research/editorial-policies/image-integrity) and to the following points below:

32nature portfolio

Please include a statement before the acknowledgements naming the author to whom correspondence and requests for materials should be addressed.

Finally, we require authors to include a statement of their individual contributions to the paper -- such as experimental work, project planning, data analysis, etc. -- immediately after the acknowledgements. The statement should be short, and refer to authors by their initials. For details please see the Authorship section of our joint Editorial policies at http://www.nature.com/authors/editorial_policies/authorship.html

- * include a point-by-point response to any editorial suggestions and to our referees. Please include your response to the editorial suggestions in your cover letter, and please upload your response to the referees as a separate document.

- * ensure it complies with our format requirements for Letters as set out in our guide to authors at www.nature.com/nmicrobiol/info/gta/

- * state in a cover note the length of the text, methods and legends; the number of references; number and estimated final size of figures and tables

- * resubmit electronically if possible using the link below to access your home page:

{redacted}

*This url links to your confidential homepage and associated information about manuscripts you may have submitted or be reviewing for us. If you wish to forward this e-mail to co-authors, please delete this link to your homepage first.

Please ensure that all correspondence is marked with your Nature Microbiology reference number in the subject line.

Nature Microbiology is committed to improving transparency in authorship. As part of our efforts in this direction, we are now requesting that all authors identified as 'corresponding author' on published papers create and link their Open Researcher and Contributor Identifier (ORCID) with their account on the Manuscript Tracking System (MTS), prior to acceptance. This applies to primary research papers only. ORCID helps the scientific community achieve unambiguous attribution of all scholarly contributions. You can create and link your ORCID from the home page of the MTS by clicking on

33nature portfolio

'Modify my Springer Nature account'. For more information please visit www.springernature.com/orcid.

We hope to receive your revised paper within three weeks. If you cannot send it within this time, please let us know.

Yours sincerely,

{redacted}

Reviewer Expertise:

Referee #1: resistome

Referee #2: antimicrobial resistance in LMICs

Referee #3: Genomic epidemiology/public health

Referee #4: Enteric bacteria epidemiology

Reviewers Comments:

Reviewer #1 (Remarks to the Author):

The additional research by the authors in the interim period is appreciated and reflects a much-improved manuscript relative to the original submission. Responses to most major comments in the Rebuttal are satisfactory. Outstanding concerns are highlighted below.

Major comments:

The statistical analysis included in this revised version of the manuscript still requires clarification. The authors claim their analysis is exploratory in nature and therefore does not require p-value correction for multiple hypothesis testing. While the vast amount of data and metadata provided in this work is appreciated, this statement is invalid and unacceptable. Crucially, this work will be keenly discussed in the field and is likely to be well cited. While the authors intention may in fact be to generate hypotheses for follow-up work, as presented, this work runs an extremely high risk of being cited (inappropriately) as proof for associations between the presence of resistance genes and various health, birth outcomes, and hygiene measurements that are not currently statistically supported by their data. Because of the potential profound impact of this work in the field, robust statistical associations are of paramount importance. This includes correction of p-values, specifically in cases of exploratory, multi-associative analysis. Further, it remains unclear why the authors perform exploratory univariate analysis to investigate the association of metadata and ARG carriage (Supplementary Table 11) and subsequently, instead of testing significant univariate variables in the multivariate approach, rely on 'expert opinion and literature' (Lines 592-593) for variable inclusion in

34the multivariate approach. This seems really dangerous and misleading. Clarity and transparency of the statistical methodology is essential for this work to provide robust and informative associations for this vast dataset.

Minor revisions:

1. Lines 86-87: Numbers listed do not match the referenced Supplementary Figure 1 (35,040 vs. 35,016 and 36,348 vs. 36,285).
2. Line 95: An explanation would be appreciated as to why only 2,931 neonatal rectal swabs were analyzed if 36,285 neonates were enrolled, and how this sub-cohort of rectal swabs was chosen for analysis.
3. Lines 97-98 (repeated 106-107): The values in parentheses require revision. As an example, at first read the section "18.5% (5.0% BS, 13.4% NoBS)" seems to imply that just 5% of all BS rectal swabs and 13.4% of all NoBS rectal swabs harbor blaNDM. In reality, the values written by the authors reflect that 5% and 13.4% of all BR swabs are blaNDM-positive BS and blaNDM-positive NoBS, respectively. These are very different interpretations, as the former implies that prevalence of each carbapenemase or ESBL is three-fold lower in BS rectal swabs, which is not the case or intention as in reality this results from a roughly threefold difference in cohort size between BS (n=626) and NoBS (2305). Because of the marked differences in cohort size, the values in parentheses should be changed to reflect percentage of ARG carriage within BS rectal swabs and within NoBS rectal swabs, and this be explicitly stated.
4. Lines 120-123: Here the authors contrast the neonates' age at time of BR collection against carriage of ARGs, finding ARGs were consistently found among BR, regardless of delivery type or sepsis outcome. However, Figure 1C and Supplementary Figure 3 show a trend of steady decrease in prevalence of blaNDM (~50% \diamond 25%) and blaOXA48 (~30% \diamond 0%) from 0 days of life to 14 days of life (values in parentheses reflect Figure 1C but are consistent for Asian rectal swabs in Supplemental Figure 3); this should be commented on in this section.
5. Figure 1C: The sample size circles below each graph are duplicated across Asia and Africa. Requires revision.
6. Supplementary Figure 3: Sizes of the sample size circles should be consistent across panels. For example, Day of Life 9 in Biological sepsis (Asia) (n=10) has a much larger circle size than Day of Life 5 in No biological sepsis (Asia) (n=13). Standardized sizes should be reflected in Figure 1C as well.
7. Lines 298-300: One peril of uncorrected p-values described above is associations of this type seen here. What is the clinical relevance if occasional handwashing is associated with carriage of an ESBL and a carbapenemase, yet frequent handwashing is associated with carriage of a different carbapenemase? If these results hold after hypothesis correction, this finding would benefit from commentary to contextualize the association and clinical relevance.

Reviewer #2 (Remarks to the Author):

The authors have answered the questions and concerns clearly. The manuscript has been markedly improved. I have no additional comments.

35Reviewer #3 (Remarks to the Author):

The authors have undertaken a number of revisions to the manuscript in response to previous comments.

The paper is now easier to follow, and the presentation of the data more logical.

However, this still remains a complex paper, with many different aspects of the study included - I do wonder if the different analyses (detailed genomics versus epi investigations of risk factors for AMR carriage) would be better suited to different manuscripts.

Ultimately, this is important data on the prevalence of AMR in these settings.

This inclusion of additional information/comments highlighting limitations of the results is useful.

Reviewer #4 (Remarks to the Author):

Again a very valuable study/data set and the manuscript is improved but there are still several distracting issues with typos and difficulty reading and again although the authors addressed to some degree the work remains very descriptive and lacks additional analysis on source of ARGs in this valuable data set.

Please try and avoid all the shortening of terms as it makes an already dense manuscript even harder for a reader to follow (e.g. line 357-9 "We highlighted the carriage of ARGs in neonates from the very early hours after birth (day 0/first 24 hours of life), irrespectively of CS or SVD, or if developing BS or not, which may have been underpinned by antibiotherapy...". This is an issue throughout the manuscript and often the definitions are not even clear e.g.: "neonates had different sepsis outcomes (BSyn)." Line 106.

It is not that all abbreviations need to be removed but as written it is difficult to follow.

The reviewer would still encourage the authors to clarify some of the labeling in the figures so they can stand alone. For example in Fig 1. Define

Discussion

Some of the conclusions are unclear; for example the reviewer is not sure how the authors draw the conclusion that because they are on different plasmids it means it was nosocomial acquisition, please clarify. "Moreover, ARGs, and in particular blaNDM, were found in different plasmids (IncX3 in Bangladesh; IncN2 or IncA/C2 in Pakistan), emphasising the role of the hospital as a hotspot for acquisition and dissemination of MDR 390 pathogens harbouring ARGs."

Minor

Line 79 not clear how ii differs from iii, "(ii) maternal ARG carriage and birth traits, and (iii) birth traits and ARG's in the neonatal gut microbiota."

Line 101 define "PP" The highest prevalence was in PP, Pakistan

36Why do the authors think there is lower presence of resistance in neonates without sepsis? blaCTX-M-15, blaNDM, blaKPC, and blaOXA-48-like genes were detected in 56.1% (14.0% BS, 42.1% NoBS), 18.5% (5.0% BS, 13.4% NoBS) for CTX-M and NDM? This is likely statistically significant based on the numbers but might be helpful to include at least in the discussion. However, the data presented in supplemental Figure 3 does not appear to be as dramatically different.

As discussed in line 137-140 the antimicrobials given during sepsis would be anticipated to increase the AGR carriage if anything. It may be that the rectal swabs are taken at onset rather than later in the course.

The result as written above is in direct contrast to the conclusion statement line 354 "We speculate that because the majority of these neonates have been administered antibiotics, presenting with clinical sepsis, antibiotic selection pressure favoured resistant bacteria as described previously"

As the variation in resistance is so dramatic from one region to the next it will be important to make sure other deviations from the protocol in one or two locations are not accounting for the differences in the timing of resistance. For example, there is a line that all neonatal sepsis would be assessed after day 7 but a large portion of the data even for neonates with biologic sepsis seem to be prior to day 7. Was this accounted for with deviation from specific regions?

Seems like the paragraph 112-117 belongs in the discussion rather than results as you are comparing the results of this study to the known literature rather than presenting results from the study alone.

The genomics section "Genomic analysis of *E. coli*, *E. cloacae* and *K. pneumoniae* MR and BR isolates" remains very hard to follow. As this is such a valuable data set to look at local transmission versus maternal to infant transmission would ask that the authors put this in the context of the data with how many neonatal-maternal pairs could be sequenced and from that how often was there a match in strains versus the mentioned outbreak and look at the genomic comparisons of strains from a single location and then quantify the diversity again if possible. Not clear there is great value in placing in the context of other strains identified globally as direct transmission from these sources would need to be massively inferred and might ask that the relatedness of isolates from the same species and same hospital are compared versus maternal and infant isolate pairs compared as this can only be done with this type of data set which makes the work so valuable. It may be that the isolates to make these inferences are not available but putting some analysis and numbers to most likely source (i.e. nosocomial versus maternal) would be additive to the literature. This would include modifying or removing figures 3-5 and likely just looking at pairs or improved labeling for the supplementary figures 5-8. In the current form the ST trees are not additive as the labeling is not discernable.

Not sure the resistance profiles are very helpful if this data has been reported previously. If not would include.

Line 373 "...Potentially leading to neonatal BS" Would anticipate that that this statement needs to be modified. It would be helpful to explore the rate of organism transmission but likely a specific organism acquisition is not sufficient to cause neonatal sepsis (potentially Group B Streptococcus is an

37nature portfolio

exception or increases the likelihood although interesting if there was any signal in this data). Would be helpful to understand the source of MDRO in neonates as they should be coming from a sterile environment into the world via contaminated birth canal. Would anticipate more transmission and acquisition of flora from vaginal delivery which is natural and healthy versus acquiring from a nosocomial source. This would be an important issue for the authors to explore in the results and the discussion and if they cannot because of a limited matched data set it would be good to state this in the results/discussion. The lines 380-89 do address some of this but it would be more rigorous to provide the reader with the context of inferred frequency of nosocomial versus maternal rather than just mentioning a handful of cases without a denominator.

The plasmid analysis and which isolates were selected for long read sequencing is challenging. To claim co-location of blaNDM and plasmid Inc type long read sequencing would be needed for these isolate and thus could the authors clarify how this was decided. Line 553-4 "Plasmid analysis was performed for n=50 isolates chosen based on short read sequencing analysis (ARG carriage, ST and core genome phylogeny) and other metadata available (clinical site, sample type, date)."

If the legend and abbreviations can be addressed both Figure 1 and figure 6 are improvements in the revision.

Author Rebuttal, first revision:

Reviewer #1 (Remarks to the Author):

The additional research by the authors in the interim period is appreciated and reflects a much-improved manuscript relative to the original submission. Responses to most major comments in the Rebuttal are satisfactory. Outstanding concerns are highlighted below.

Major comments:

The statistical analysis included in this revised version of the manuscript still requires clarification. The authors claim their analysis is exploratory in nature and therefore does not require p-value correction for multiple hypothesis testing. While the vast amount of data and metadata provided in this work is appreciated, this statement is invalid and unacceptable. Crucially, this work will be keenly discussed in the field and is likely to be well cited. While the authors intention may in fact be to generate hypotheses for follow-up work, as presented, this work runs an extremely high risk of being cited (inappropriately) as proof for associations between the presence of resistance genes and various health, birth outcomes, and hygiene measurements that are not currently statistically supported by their data. Because of the potential profound impact of this work in the field,

38nature portfolio

robust statistical associations are of paramount importance. This includes correction of p-values, specifically in cases of exploratory, multi-associative analysis. Further, it remains unclear why the authors perform exploratory univariate analysis to investigate the association of metadata and ARG carriage (Supplementary Table 11) and subsequently, instead of testing significant univariate variables in the multivariate approach, rely on 'expert opinion and literature' (Lines 592-593) for variable inclusion in the multivariate approach. This seems really dangerous and misleading. Clarity and transparency of the statistical methodology is essential for this work to provide robust and informative associations for this vast dataset.

Author response: We thank reviewer 1 for their additional statistical comments. We have grouped these into two sections below (multiple hypothesis testing and variable selection for multivariable analysis) and have responded accordingly.

Multiple hypothesis testing:

We have now provided p-values corrected for multiple testing (where we have felt this appropriate) using the Holm-Bonferroni method to control the family-wise error rate at 0.05. When considering our adjustment approach, we decided to apply adjustments on a within outcome/model basis. For example, all univariable associations with carriage of $bla_{CTX-M-15}$ from the mother rectal samples will be adjusted for multiple testing separate to all univariable associations with carriage of bla_{NDM} from the mother rectal samples (Supplementary Table 11). Depending on the purpose of our multivariable analyses (MVA; see response to minor comment #7 for more detail on this), our adjustment approach differed. For the MVA related to WASH variables (Supplementary Table 12) and birth healthcare environment (Supplementary Table 17), we have adjusted these on a per-outcome basis as described above. For the remaining MVA, we were interested in the association between one exposure and the outcome, with the other variables included for confounder control only and thus having no meaningful interpretation on their own. We have not adjusted the exposure variable for multiple testing, and to avoid misinterpretation of these estimates (often referred to as the Table 2 fallacy), we have removed them from the tables and instead reported which variables were adjusted for in the MVA. Finally, models in Supplementary Table 16 (one p-value per outcome), and Supplementary Table 19 (one p-value per outcome) were not adjusted for

39nature portfolio

multiple testing. We have described this approach (in a more concise manner) in the manuscript and have updated the p-values within the text and Tables accordingly.

Variable selection for multivariable analysis:

As mentioned above, our MVA served two purposes. Some models (those related to WASH variables in Supplementary Table 12 and those related to birth healthcare environment in Supplementary Table 17), were descriptive in nature and aimed to fit a multivariable model using all variables pertaining to our phenomena of interest. The only exclusions applied were variables that had large amounts of missing data or variables that had a deterministic relationship with each other. In the latter situation, we refitted models swapping these variables out (hence the MV1 MV2 headings seen in some of the Supplementary Tables). The second purpose was confounding adjustment.

Basing variable selection on the group expert opinion and the vast literature used in our study, we selected variables among our long list of data which were relevant for the purposes above.

On behalf of the STRATOS initiative, Wallisch and colleagues (2022) in their Review of guidance papers on regression modelling in statistical series of medical journals (<https://doi.org/10.1371/journal.pone.0262918>), mention that the recent and rapid development of statistical methodologies has not been adequately reflected in many medical publications. They have found recommendations in favour of or against specific variable selection methods, the majority being in favour of taking advantage of background knowledge to select variables and considering that stepwise methods could be unstable, overfitting and, thus, not robust. The authors stated that it has been long known that univariable screening should be avoided because it has the potential to wrongly reject important variables and, in fact, this method for variable selection was never recommended in any of the reviewed series. Furthermore, "if sufficient background knowledge is available, pre-filtering or even the selection of variables should be based on this information rather than using data-driven methods on the entire data set". Furthermore, statistical associations in and of themselves cannot give us any indication that a variable is a confounder or any causal role a variable may be. These are always interpretations we place on model estimates, which only ever indicate correlation or association.nature portfolio

In this way, we respectfully disagree with the reviewer and believe we applied a contemporary, relevant, and recommended methodology for variable selection for the multivariable models we fitted.

Minor revisions:

1. Lines 86-87: Numbers listed do not match the referenced Supplementary Figure 1 (35,040 vs. 35,016 and 36,348 vs. 36,285).

Author response: Thank you for noticing this error and we apologise for it. We corrected the values in the text, Supplementary Figure 1 and Supplementary Tables 1 and 2.

Line 83: Overall, BARNARDS recruited 35,040 mothers and their respective neonates (n= 36,285; Supplementary Figure 1).

2. Line 95: An explanation would be appreciated as to why only 2,931 neonatal rectal swabs were analyzed if 36,285 neonates were enrolled, and how this sub-cohort of rectal swabs was chosen for analysis.

Author response: We thank the reviewer for raising this comment and we can offer a detailed explanation. During BARNARDS, 36,285 neonates were enrolled however neonatal rectal swabs were only collected upon clinical signs of sepsis and/or clinical discretion, therefore in total, during the study, 2,931 neonatal rectal swabs were collected.

41nature portfolio

We prioritised the analysis of all neonatal rectal swabs, and the data presented herein represents all neonatal rectal swabs.

3. Lines 97-98 (repeated 106-107): The values in parentheses require revision. As an example, at first read the section “18.5% (5.0% BS, 13.4% NoBS)” seems to imply that just 5% of all BS rectal swabs and 13.4% of all NoBS rectal swabs harbor bla_{NDM}. In reality, the values written by the authors reflect that 5% and 13.4% of all BR swabs are bla_{NDM}-positive BS and bla_{NDM}-positive NoBS, respectively. These are very different interpretations, as the former implies that prevalence of each carbapenemase or ESBL is three-fold lower in BS rectal swabs, which is not the case or intention as in reality this results from a roughly threefold difference in cohort size between BS (n=626) and NoBS (2305). Because of the marked differences in cohort size, the values in parentheses should be changed to **reflect percentage of ARG carriage within BS rectal swabs and within NoBS rectal swabs, and this be explicitly stated.**

Author response: We agree with the reviewer in that the way we presented these results may lead to confusion and have changed the text accordingly.

Lines 91-93: bla_{CTX-M-15}, bla_{NDM}, bla_{KPC}, and bla_{OXA-48}-like genes were detected in 56.1% (within 65% of BS and 54% of NoBS), 18.5% (within 24% of BS and 17% of NoBS), 0% and 4.1% (within 10% of BS and 2% of NoBS) of BR, respectively.nature portfolio

Lines 99-102: From these, 47.1% (detected within 54.4% BS, 46.4% NoBS, and 50% BSyn), 4.6% (detected within 6.93% BS, 4.40% NoBS, 1.47% BSyn), 0.05% (NoBS) and 1.6% (detected within 1.92% BS, 1.57% NoBS, 4.41% BSyn) carried *bla*_{CTX-M-15}, *bla*_{NDM}, *bla*_{KPC}, and *bla*_{OXA-48}-like, respectively.

4. Lines 120-123: Here the authors contrast the neonates' age at time of BR collection against carriage of ARGs, finding ARGs were consistently found among BR, regardless of delivery type or sepsis outcome. However, Figure 1C and Supplementary Figure 3 show a trend of steady decrease in prevalence of *bla*_{NDM} (~50% à 25%) and *bla*_{OXA48} (~30% à 0%) from 0 days of life to 14 days of life (values in parentheses reflect Figure 1C but are consistent for Asian rectal swabs in Supplemental Figure 3); this should be commented on in this section.

Author response: We agree with the reviewer and have amended the text accordingly to emphasise the steady decrease in prevalence of *bla*_{NDM} and *bla*_{OXA-48} among the Asian samples independently of type of delivery or sepsis outcome.

Lines 117 - 120: A steady decrease was observed for the prevalence of *bla*_{NDM} (53.7% to 27.7%) and *bla*_{OXA-48}-like genes (35.4% to 0%) among the Asian samples through the first 14 days of life (Figure 1c), independently of type of delivery or sepsis outcome (Supplementary Figures 3a, 3c).

5. Figure 1C: The sample size circles below each graph are duplicated across Asia and Africa. Requires revision.

Author response: We thank the reviewer for identifying this, and these have been revised accordingly.

436. Supplementary Figure 3: Sizes of the sample size circles should be consistent across panels. For example, Day of Life 9 in Biological sepsis (Asia) (n=10) has a much larger circle size than Day of Life 5 in No biological sepsis (Asia) (n=13). Standardized sizes should be reflected in Figure 1C as well.

Author response: We agree with the reviewer, and we have amended the circle sizes for consistency across Figure 1C and Supplementary Figure 3.

7. Lines 298-300: One peril of uncorrected p-values described above is associations of this type seen here. What is the clinical relevance if occasional handwashing is associated with carriage of an ESBL and a carbapenemase, yet frequent handwashing is associated with carriage of a different carbapenemase? If these results hold after hypothesis correction, this finding would benefit from commentary to contextualize the association and clinical relevance.

Author response: We thank the reviewer for raising this concern. After multiple hypothesis testing, the results confirm these associations (Table 1A; lines 231-235). We hypothesize that having a deficient hand hygiene, either if handwashing was described as occasional or frequent, could be a driver of ARG carriage and this may explain the results obtained.

Univariable and multivariable analysis suggests that there seems to be quite strong evidence that mother's handwashing frequency is associated with *bla*_{CTX-M-15} carriage in MR. This association is less strong for the other two ARGs.

nature portfolio

Reviewer #2 (Remarks to the Author):

The authors have answered the questions and concerns clearly. The manuscript has been markedly improved. I have no additional comments.

Author response: We thank reviewer 2 for all their previous suggestions and their kind words.

Reviewer #3 (Remarks to the Author):

The authors have undertaken a number of revisions to the manuscript in response to previous comments. The paper is now easier to follow, and the presentation of the data more logical.

However, this still remains a complex paper, with many different aspects of the study included - I do wonder if the different analyses (detailed genomics versus epi investigations of risk factors for AMR carriage) would be better suited to different manuscripts.

Ultimately, this is important data on the prevalence of AMR in these settings.

This inclusion of additional information/comments highlighting limitations of the results is useful.

Author response: We thank the reviewer for all previous suggestions. We do acknowledge that the current article remains a complex paper. Whilst it is a good suggestion to separate the data to separate manuscripts (i.e., genomics versus epi/risk factors), we do feel strongly that this data should be presented as one account. Providing a rich sociodemographic/epidemiological dataset amidst microbiology analysis allows the readership to understand the complexities of antimicrobial resistance research. Additionally, our combination of data within this article emphasises the need for inter-disciplinary research and analyses to expand understandings of carriage, infection, and bacterial transmission dynamics.

45Reviewer #4 (Remarks to the Author):

Again a very valuable study/data set and the manuscript is improved but there are still several distracting issues with typos and difficulty reading and again although the authors addressed to some degree the work remains very descriptive and lacks additional analysis on source of ARGs in this valuable data set.

Author response: We appreciate the reviewer raising this concern and do acknowledge the (intentional) descriptive nature of this study which aimed:

- To assess the carriage of clinically important ARGs in the normal gut microbiota of neonates and mothers from multiple clinical sites in seven low- and- middle income countries in the context of neonatal sepsis;
- To describe genomic traits of the most common Gram-negative isolates carrying the ARGs in study;
- To unravel associations between carriage of these ARGs and sociodemographic/ clinical/ epidemiological factors and, conversely,
- To understand if sociodemographic/ clinical/ epidemiological factors were associated with the carriage of ARGs.

This study was not designed to describe bacterial transmission dynamics, including the source of neonatal rectal bacterial isolates (mother or hospital environment). However, as we had access to genomic and epidemiological features which could reveal linkages between isolates collected from mothers, neonates and from blood samples collected from the same neonate (<https://doi.org/10.1038/s41564-021-00870-7>), we looked for genetic variations by single-nucleotide polymorphism analysis and results are shown in Supplementary Tables 7-9 and Supplementary Figures 5-8, which we have improved for clarification. There were n=24 ST405 *E. coli* from Bangladesh (13 BR, 11 MR) collected from June 2016 until May 2017, from which 12, including a mother-neonate pair were within 6 SNPs (Supplementary Table 7; Supplementary Figure 5); n=16 ST418 *E. hormaechei* from Bangladesh, among which 5 MR collected in March 2016 were closely related, and 11 BR collected between April 2016 and November 2017 formed a distinct cluster (Supplementary Table 8; Supplementary Figure 6); n=12 ST11 *K.*

46nature portfolio

pneumoniae (11 BR and 1 MR) collected between December 2015 and September 2017 formed two distinct clusters and one isolate clustered separately with an isolate previously reported in another study (Supplementary Table 9; Supplementary Figure 7); n=25 ST15 *K. pneumoniae* (20 BR and 5 MR) collected from March 2016 until September 2017 (Supplementary Table 8; Supplementary Figure 9).

Please try and avoid all the shortening of terms as it makes an already dense manuscript even harder for a reader to follow (e.g. line 357-9 "We highlighted the carriage of ARGs in neonates from the very early hours after birth (day 0/first 24 hours of life), irrespectively of CS or SVD, or if developing BS or not, which may have been underpinned by antibiotherapy...". This is an issue throughout the manuscript and often the definitions are not even clear e.g.: "neonates had different sepsis outcomes (BSyn)." Line 106.

It is not that all abbreviations need to be removed but as written it is difficult to follow.

The reviewer would still encourage the authors to clarify some of the labeling in the figures so they can stand alone. For example in Fig 1. Define

Author response: We appreciate there are many abbreviated terms within this manuscript, although we are limited by a tight word count. We have however rephrased sentences to avoid the inclusion of multiple abbreviations, where possible.

Example in Lines 273-276: We highlighted the carriage of ARGs in neonates from the very early hours after birth, irrespectively of delivery type or sepsis outcome, which may have been underpinned by antibiotherapy following acquisition from the mother and/or environment.

Also, we have used the full terms in the beginning of each section and then the abbreviation in the remaining part of the section.

47Example in Lines 261-265: We found that colonisation of the mothers' gut with *bla*_{CTX-M-15} or *bla*_{NDM} positive microbiota was associated with the development of biological sepsis in neonates and this may be due to mothers transmitting MDR pathogens to their neonates during or after labour and birth, potentially leading to neonatal BS. Neonates carrying *bla*_{CTX-M-15} or *bla*_{OXA-48}-like genes in their microbiota were more likely to have BS compared to non-carriers (Supplementary Table 19).

BSyn, refers to 68 mothers with a multiple pregnancy, whose neonates had different sepsis outcomes, wherein the mothers gave birth to two or more neonates (twins, triplets, quadruplets) and at least one developed biological sepsis whereas the other(s) did not.

We agree with the reviewer and have expanded the labelling in the figures' legends accordingly.

Discussion

Some of the conclusions are unclear; for example the reviewer is not sure how the authors draw the conclusion that because they are on different plasmids it means it was nosocomial acquisition, please clarify. "Moreover, ARGs, and in particular *bla*_{NDM}, were found in different plasmids (IncX3 in Bangladesh; IncN2 or IncA/C2 in Pakistan), emphasising the role of the hospital as a hotspot for acquisition and dissemination of MDR 390 pathogens harbouring ARGs."

Author response: We apologise for the confusion, and we have rephrased this sentence in the conclusion for clarity and to avoid the assumption of stating multiple acquisition events within the hospital environment.

nature portfolio

Lines 295-297: Moreover, ARGs, and in particular *bla*_{NDM}, were found in different plasmids (IncX3 in Bangladesh; IncN2 or IncA/C2 in Pakistan), emphasising a diverse dissemination of MDR pathogens harbouring ARGs.

Minor

Line 79 not clear how ii differs from iii, “(ii) maternal ARG carriage and birth traits, and (iii) birth traits and ARG’s in the neonatal gut microbiota.”

Author response: We thank the reviewer for raising this. (ii) examines if the carriage of ARG in the mothers’ microbiota (predictor) could be associated with birth traits (outcome) such as type of delivery, timing of birth, perinatal asphyxia, breech presentation and PPRM (Supplementary Table 15), and (iii) examines if these birth traits (predictor) could be associated with the carriage of these ARGs in the neonates rectal microbiota (outcome; Supplementary Table 18).

Line 101 define “PP” The highest prevalence was in PP, Pakistan

Author response: All site abbreviations are listed in the methods section, and we added this indication in the text and in the legend of Figure 1. We are limited by a tight word count.

Line 171: sites’ acronyms are detailed in methods

49nature portfolio

Why do the authors think there is lower presence of resistance in neonates without sepsis? blaCTX-M-15, “blaNDM, blaKPC, and blaOXA-48-like genes were detected in 56.1% (14.0% BS, 42.1% NoBS), 18.5% (5.0% BS, 13.4% NoBS)” for CTX-M and NDM? This is likely statistically significant based on the numbers but might be helpful to include at least in the discussion. However, the data presented in supplemental Figure 3 does not appear to be as dramatically different.

As discussed in line 137-140 the antimicrobials given during sepsis would be anticipated to increase the AGR carriage if anything. It may be that the rectal swabs are taken at onset rather than later in the course. The result as written above is in direct contrast to the conclusion statement line 354 “We speculate that because the majority of these neonates have been administered antibiotics, presenting with clinical sepsis, antibiotic selection pressure favoured resistant bacteria as described previously”.

Author response: Another comment made by another reviewer in relation to the way we presented these data made us realise it could lead to different interpretations:

“the section “18.5% (5.0% BS, 13.4% NoBS)” seems to imply that just 5% of all BS rectal swabs and 13.4% of all NoBS rectal swabs harbour. The values written by the authors reflect that 5% and 13.4% of all BR swabs are blaNDM-positive BS and blaNDM-positive NoBS, respectively. These are very different interpretations, as the former implies that prevalence of each carbapenemase or ESBL is three-fold lower in BS rectal swabs, which is not the case or intention as in reality this results from a roughly threefold difference in cohort size between BS (n=626) and NoBS (2305)”.

We agree with the reviewer in that the way we presented these results may lead to confusion and have changed the text accordingly to reflect percentage of ARG carriage within BS rectal swabs and within NoBS rectal swabs.

nature portfolio

Lines 91-93: *bla*_{CTX-M-15}, *bla*_{NDM}, *bla*_{KPC}, and *bla*_{OXA-48}-like genes were detected in 56.1% (within 65% of BS and 54% of NoBS), 18.5% (within 24% of BS and 17% of NoBS), 0% and 4.1% (within 10% of BS and 2% of NoBS) of BR, respectively.

Lines 99-102: From these, 47.1% (detected within 54.4% BS, 46.4% NoBS, and 50% BSyn), 4.6% (detected within 6.93% BS, 4.40% NoBS, 1.47% BSyn), 0.05% (NoBS) and 1.6% (detected within 1.92% BS, 1.57% NoBS, 4.41% BSyn) carried *bla*_{CTX-M-15}, *bla*_{NDM}, *bla*_{KPC}, and *bla*_{OXA-48}-like, respectively.

As the variation in resistance is so dramatic from one region to the next it will be important to make sure other deviations from the protocol in one or two locations are not accounting for the differences in the timing of resistance. For example, there is a line that all neonatal sepsis would be assessed after day 7 but a large portion of the data even for neonates with biologic sepsis seem to be prior to day 7. Was this accounted for with deviation from specific regions?

Author response: According to the established protocol, rectal samples were to be taken from neonates ≥ 7 up to 60 days old with clinically suspected sepsis. However, during the study, rectal samples were taken from neonates with clinically suspected sepsis from 0 days of life onward and these samples were also characterised and included herein. There were no other deviations from the protocol other than the timing of sample collection and most samples were taken ≥ 7 days from birth (see table below).

We did not take into consideration the timing of sample collection to report the prevalence of ARG among the different sites/countries/geographical regions. It is true that our analysis of neonate age vs carriage of ARG (Figure 1c) shows a decrease on the prevalence of ARG with time in South-Asian countries, although, the prevalence we see for each country is in accordance to what has been described, whenever we could find reliable data. Within country variation may be due to the idiosyncrasies of the sites as there were clinical sites in urban areas which

51nature portfolio

were district hospitals, and smaller hospitals in rural sites, with different IPC conditions. For example, the Nigerian sites in National Hospital Abuja (NN; referral hospital in the capital city) and Murtala Mohammad Specialist Hospital, Kano (NK; tertiary care hospital in a more rural setting) **in Nigeria.**

Commented [MC1]: This is the reposne I went over the top. Don't know what to say. We had thses samples and we studied them This is waht we have and is in agreement with what has been seen before...
Also, I didn't know what to say about Kano without literature.

Commented [MC2]: This is where I went over the top. i don't know what to say; it doest make sense to me. We had these samples and we went to look what was there. The data is consistent with waht is known, so what can we say. Also, I don;t known if what i said in relation to Kano i ok; I didn't know what to say without giving some literature...

	Samples taken days 0-6	Samples taken 7-60 days
BC-Bangladesh	103	728
BK-Bangladesh	35	210
ES-Ethiopia	45	21
IN-India	43	28
NK-Nigeria	53	618
NN-Nigeria	38	156
NW-Nigeria	1	15
PC-Pakistan	4	5
PP-Pakistan	94	26
RU-Rwanda	8	6
RK-Rwanda	17	35
ZAT-South Africa	323	63
Total	764	1911nature portfolio

Seems like the paragraph 112-117 belongs in the discussion rather than results as you are comparing the results of this study to the known literature rather than presenting results from the study alone.

Author response: We agree and we have restructured this section. This paragraph has been removed from the results due to the tight word count.

The genomics section “Genomic analysis of *E. coli*, *E. cloacae* and *K. pneumoniae* MR and BR isolates” remains very hard to follow. As this is such a valuable data set to look at local transmission versus maternal to infant transmission would ask that the authors put this in the context of the data with how many neonatal-maternal pairs could be sequenced and from that how often was there a match in strains versus the mentioned outbreak and look at the genomic comparisons of strains from a single location and then quantify the diversity again if possible. Not clear there is great value in placing in the context of other strains identified globally as direct transmission from these sources would need to be massively inferred and might ask that the relatedness of isolates from the same species and same hospital are compared versus maternal and infant isolate pairs compared as this can only be done with this type of data set which makes the work so valuable. It may be that the isolates to make these inferences are not available but putting some analysis and numbers to most likely source (i.e. nosocomial versus maternal) would be additive to the literature. This would include modifying or removing figures 3-5 and likely just looking at pairs or improved labelling for the supplementary figures 5-8. In the current form the ST trees are not additive as the labelling is not discernible.

Author response: We thank the reviewer for their comment and do appreciate that this section contains a lot of data. Our original aim and intention with this section was to characterise and describe the genomic diversity amongst the rectal isolates carrying the carbapenemase antibiotic resistance genes. In order to complete this, it was decided that the three most frequent bacterial species were selected to provide a detailed characterisation. A genomic global epidemiological comparison offers some context whilst emphasising the diversity amongst our

53nature portfolio

rectal isolate collections. Within this manuscript, it was not an aim to analyse any transmission networks between bacterial isolates recovered from the mothers and the neonates' Following bioinformatics analysis, we did report on such transmission networks, either mother-neonate and/or neonate-neonate. Matches between mother and neonate have also been annotated on the core genome phylogenetic trees (denoted with orange triangles). It is worth noting at this point that there is an additional BARNARDS dataset that incorporates bacterial transmission dynamics between the mother > neonate > hospital environment. The additional dataset is a large microbiological analysis (including sequencing) of bacteria recovered from hospital surfaces. This dataset will allow a deeper analysis of transmission dynamics between the multiple sources.

Not sure the resistance profiles are very helpful if this data has been reported previously. If not would include.

Author response: We thank for the reviewer for this comment however this data has not been reported in any previous BARNARDS articles. These isolates are recovered from rectal swabs and are not infection isolates (data presented here: <https://doi.org/10.1038/s41564-021-00870-7>).

Line 373 "...Potentially leading to neonatal BS" Would anticipate that that this statement needs to be modified. It would be helpful to explore the rate of organism transmission but likely a specific organism acquisition is not sufficient to cause neonatal sepsis (potentially Group B Streptococcus is an exception or increases the likelihood although interesting if there was any signal in this data). Would be helpful to understand the source of MDRO in neonates as they should be coming from a sterile environment into the world via contaminated birth canal. Would anticipate more transmission and acquisition of flora from vaginal delivery which is natural and healthy versus acquiring from a nosocomial source. This would be an important issue for the authors to explore in the results and the discussion and if they cannot because of a limited matched data set it would be good to state this in the results/discussion. The lines 380-89 do address some of this but it would be more rigorous to provide the reader

54nature portfolio

with the context of inferred frequency of nosocomial versus maternal rather than just mentioning a handful of cases without a denominator.

Author response: We thank the reviewer for this comment. Our matched dataset, as this was outside of the original scope for this study, is very limited and therefore we cannot fully convey this within the manuscript. We mention that our findings (genomic analysis, epidemiological data and statistical analysis) all point to the possibility of the hospital environment and the contact with mothers during birth being routes of neonates gut colonisation with ARGs. However, our data does not allow us to explore this further and it is outside of this study aims.

The plasmid analysis and which isolates were selected for long read sequencing is challenging. To claim co-location of bla_{NDM} and plasmid Inc type long read sequencing would be needed for these isolate and thus could the authors clarify how this was decided. Line 553-4 “Plasmid analysis was performed for n=50 isolates chosen based on short read sequencing analysis (ARG carriage, ST and core genome phylogeny) and other metadata available (clinical site, sample type, date).”

Author response: We thank the reviewer for raising this point. Ideally, and going forward it is becoming more feasible to perform long-read sequencing with a higher throughput which would largely resolve the current limitations faced in microbial AMR genomics. Unfortunately, this was not available for this study and therefore isolates for additional long-read sequencing were largely selected based on short read analysis, whereby we wanted to clarify and assemble the plasmids for certain isolates/ST clusters. For isolates that were closely related (as per core genome analysis) we would select representatives based on ST groups, bla_{NDM} variants, initial data available for short read plasmid *inc* typing, site and date. Supplementary Table 6 shows all details relating to this analysis.

We appreciate it is not possible to quantify co-location of ARG and plasmid Inc type for all isolates, and we have alluded to this within the manuscript with “often on an IncX plasmid”.

55nature portfolio

If the legend and abbreviations can be addressed both Figure 1 and figure 6 are improvements in the revision.

Author response: We have addressed the legend and abbreviations for Figure 1, Figure 6 and remaining figures where appropriate.

Decision Letter, second revision:

Dear Maria,

thank you for your patience and thank you for submitting your revised manuscript "Antibiotic resistance genes in the gut microbiota of mothers and neonates from low- and middle-income countries: prevalence, risk factors and genomics" (NMICROBIOL-21051153B). It has now been seen by the original referees and their comments are below. The reviewers find that the paper has improved in revision, and therefore we'll be happy in principle to publish it in Nature Microbiology, pending minor revisions to satisfy the referees' final requests and to comply with our editorial and formatting guidelines.

Thank you again for your interest in Nature Microbiology Please do not hesitate to contact me if you have any questions.

Sincerely,

{redacted}

--

Reviewer #1 (Remarks to the Author):

the authors have adequately addressed my concerns

Reviewer #4 (Remarks to the Author):

56nature portfolio

This is an important piece of work and manuscript. Would still favor the removal of all inference of plasmid co-location of all carbapenemase genes and blaCTX-M assignments to plasmids based on short read sequencing (this may have been done but it remains difficult to confirm). Inferring plasmid location of AMR genes from short read can be very misleading and do not think it is additive to the overall message and importance.

Decision Letter, final checks:

Dear Maria,

Thank you for your patience as we've prepared the guidelines for final submission of your Nature Microbiology manuscript, "Antibiotic resistance genes in the gut microbiota of mothers and neonates from low- and middle-income countries: prevalence, risk factors and genomics" (NMICROBIOL-21051153B). Please carefully follow the step-by-step instructions provided in the attached file, and add a response in each row of the table to indicate the changes that you have made. Please also check and comment on any additional marked-up edits we have proposed within the text. Ensuring that each point is addressed will help to ensure that your revised manuscript can be swiftly handed over to our production team.

In recognition of the time and expertise our reviewers provide to Nature Microbiology's editorial process, we would like to formally acknowledge their contribution to the external peer review of your manuscript entitled "Antibiotic resistance genes in the gut microbiota of mothers and neonates from low- and middle-income countries: prevalence, risk factors and genomics". For those reviewers who give their assent, we will be publishing their names alongside the published article.

Nature Microbiology offers a Transparent Peer Review option for new original research manuscripts submitted after December 1st, 2019. As part of this initiative, we encourage our authors to support increased transparency into the peer review process by agreeing to have the reviewer comments, author rebuttal letters, and editorial decision letters published as a Supplementary item. When you submit your final files please clearly state in your cover letter whether or not you would like to participate in this initiative. Please note that failure to state your preference will result in delays in accepting your manuscript for publication.

57nature portfolio

Cover suggestions

As you prepare your final files we encourage you to consider whether you have any images or illustrations that may be appropriate for use on the cover of Nature Microbiology.

Nature Microbiology has now transitioned to a unified Rights Collection system which will allow our Author Services team to quickly and easily collect the rights and permissions required to publish your work. Approximately 10 days after your paper is formally accepted, you will receive an email in providing you with a link to complete the grant of rights. If your paper is eligible for Open Access, our Author Services team will also be in touch regarding any additional information that may be required to arrange payment for your article.

Please note that Nature Microbiology is a Transformative Journal (TJ). Authors may publish their research with us through the traditional subscription access route or make their paper immediately open access through payment of an article-processing charge (APC). Authors will not be required to make a final decision about access to their article until it has been accepted. Find out more about Transformative Journals

Authors may need to take specific actions to achieve compliance with funder and institutional open access mandates. If your research is supported by a funder that requires immediate open access (e.g. according to Plan S principles) then you should select the gold OA route, and we will direct you to the compliant route where possible. For authors selecting the subscription publication route, the journal's standard licensing terms will need to be accepted, including self-archiving policies. Those licensing terms will supersede any other terms that the author or any third party may assert apply to any version of the manuscript.

For information regarding our different publishing models please see our Transformative Journals page. If you have any questions about costs, Open Access requirements, or our legal forms, please contact ASJournals@springernature.com.

Please use the following link for uploading these materials:
{redacted}

{redacted}
--

Reviewer #1:

Remarks to the Author:

the authors have adequately addressed my concerns

Reviewer #4:

Remarks to the Author:

This is an important piece of work and manuscript. Would still favor the removal of all inference of plasmid co-location of all carbapenemase genes and blaCTX-M assignments to plasmids based on short read sequencing (this may have been done but it remains difficult to confirm). Inferring plasmid location of AMR genes from short read can be very misleading and do not think it is additive to the overall message and importance.

Final Decision Letter:

Dear Maria,

I am pleased to accept your Article "Antibiotic resistance genes in the gut microbiota of mothers and linked neonates with or without sepsis from low- and middle-income countries" for publication in Nature Microbiology. Thank you for having chosen to submit your work to us and many congratulations.

Over the next few weeks, your paper will be copyedited to ensure that it conforms to Nature

59nature portfolio

Microbiology style. We look particularly carefully at the titles of all papers to ensure that they are relatively brief and understandable.

Acceptance of your manuscript is conditional on all authors' agreement with our publication policies (see <https://www.nature.com/nmicrobiol/editorial-policies>). In particular your manuscript must not be published elsewhere and there must be no announcement of the work to any media outlet until the publication date (the day on which it is uploaded onto our website).

Please note that *Nature Microbiology* is a Transformative Journal (TJ). Authors may publish their research with us through the traditional subscription access route or make their paper immediately open access through payment of an article-processing charge (APC). Authors will not be required to make a final decision about access to their article until it has been accepted. [Find out more about Transformative Journals](https://www.springernature.com/gp/open-research/transformative-journals)

Authors may need to take specific actions to achieve [compliance](https://www.springernature.com/gp/open-research/funding/policy-compliance-faqs) with funder and institutional open access mandates. If your research is supported by a funder that requires immediate open access (e.g. according to [Plan S principles](https://www.springernature.com/gp/open-research/plan-s-compliance)) then you should select the gold OA route, and we will direct you to the compliant route where possible. For authors selecting the subscription publication route, the journal's standard licensing terms will need to be accepted, including [self-archiving policies](https://www.nature.com/nature-portfolio/editorial-policies/self-archiving-and-license-to-publish). Those licensing terms will supersede any other terms that the author or any third party may assert apply to any version of the manuscript.

60nature portfolio

An online order form for reprints of your paper is available at https://www.nature.com/reprints/author-reprints.html. All co-authors, authors' institutions and authors' funding agencies can order reprints using the form appropriate to their geographical region.

As soon as your article is published, you will receive an automated email with your shareable link